# Solving Football by Exploiting Equilibrium Structure of 2p0s Differential Games with One-Sided Information

**Mukesh Ghimire**
Arizona State University
Tempe, AZ 85281, USA
mghimire@asu.edu

**Lei Zhang**
Purdue University
West Lafayette, IN 47907, USA
zhan5814@purdue.edu

**Zhe Xu**
Arizona State University
Tempe, AZ 85281, USA
xzhe1@asu.edu

**Yi Ren**[*]
Arizona State University
Tempe, AZ 85281, USA
yiren@asu.edu

## Abstract

For a two-player imperfect-information extensive-form game (IIEFG) with $K$ time steps and a player action space of size $U$, the game tree complexity is $U^{2K}$, causing existing IIEFG solvers to struggle with large or infinite $(U, K)$, e.g., differential games with continuous action spaces. To partially address this scalability challenge, we focus on an important class of 2p0s games where the informed player (P1) knows the payoff while the uninformed player (P2) only has a belief over the set of $I$ possible payoffs. Such games encompass a wide range of scenarios in sports, defense, cybersecurity, and finance. We prove that under mild conditions, P1's (resp. P2's) equilibrium strategy at any infostate concentrates on at most $I$ (resp. $I + 1$) action prototypes. When $I \ll U$, this equilibrium structure causes the game tree complexity to collapse to $I^K$ for P1 when P2 plays best responses, and $(I + 1)^K$ for P2 in a dual game where P1 plays best responses. We then show that exploiting this structure in model-free multiagent reinforcement learning and model predictive control leads to significant improvements in learning accuracy and efficiency from SOTA IIEFG solvers. Our demonstration solves a 22-player football game with continuous action spaces and $K = 10$ time steps, where the offense team needs to strategically conceal their play until a critical moment in order to exploit information advantage. Code is available here.

## 1 Introduction

The strength of game solvers has grown rapidly in the last decade, beating elite-level human players in Chess (Silver et al., 2017a), Go (Silver et al., 2017b), Poker (Brown & Sandholm, 2019; Brown et al., 2020b), Diplomacy (FAIR† et al., 2022), Stratego (Perolat et al., 2022), among others with increasing complexity. However, most of the existing solvers with proven convergence, either based on regret matching (Tammelin, 2014; Burch et al., 2014; Moravčík et al., 2017; Brown et al., 2020b; Lanctot et al., 2009) or gradient descent-ascent (McMahan, 2011; Perolat et al., 2021; Sokota et al., 2022; Cen et al., 2021; Vieillard et al., 2020), have computational complexities increasing with the size of the *finite* action set, and suffer from game-tree complexities growing exponentially with both the action size $U$ and the tree depth $K$. Real-world games, however, often have continuous actions and happen in continuous time and state spaces, making them *differential* in nature. Applying existing solvers to differential games would require game-specific insight or extra computational overhead for automated abstraction (Kroer & Sandholm, 2015; Hawkin et al., 2011; Brown & Sandholm, 2015).

In this paper, we address this scalability challenge for an important subset of 2p0s differential games where the informed player (P1) knows the payoff type while the uninformed player (P2) only holds a

---

[*]Corresponding author.

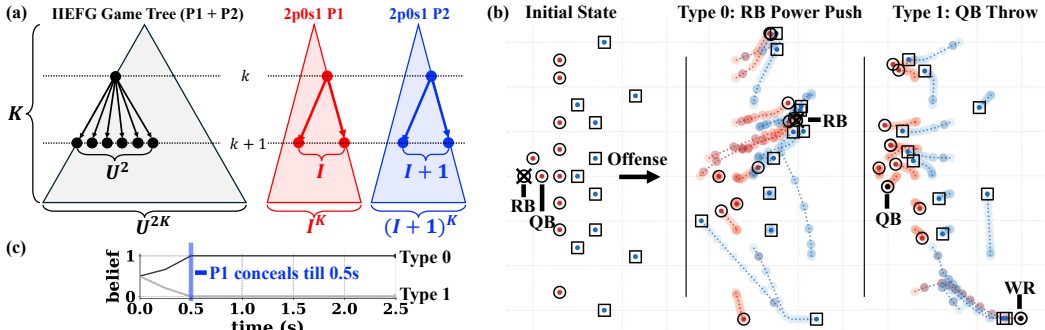

Figure 1: (a) IIEFG with $U$ actions per player per infostate and $K$ time steps has a game-tree complexity of $U^{2K}$. For 2p0s1 with $I$ payoff types, deterministic dynamics, and Isaacs' condition, we show that the NE is $I$-atomic for P1 and $(I+1)$-atomic for P2, leading to a game-tree complexity of $I^K$ for P1 in the primal game where P2 plays best responses and $(I+1)^K$ for P2 in the dual game where P1 plays best responses. (b) American Football with 22 players and continuous action spaces ($U = \infty$) with $K = 10$ time steps. P1 (red) attacks with two private game types ($I = 2$): Running back (RB) power-runs through the space created by blockers, and quarterback (QB) throws the ball to the leading wide receiver (WR). See animation. (c) At NE, P1 conceals type until 0.5 sec., similar to the reported 1.0 sec. Due to significant tree size reduction, the game can be solved in 30 minutes.

public belief $p_0 \in \Delta(I)$ over the finite set of $I$ possible types. At the beginning of the game, nature draws a game type according to $p_0$ and informs P1 about the type. As the game progresses, the public belief $p$ about the true game type is updated from $p_0$ based on the action sequence taken by P1 and its strategy via the Bayes' rule. P1's (resp. P2's) goal is to minimize (resp. maximize) the expected payoff over $p_0$. Due to the zero-sum nature, P1 may need to delay information release or manipulate P2's belief to take full advantage of information asymmetry. While restricted, such games represent a wide range of attack-defense scenarios including football set-pieces where the attacker has private information about which play is to be executed, and missile defense where multiple potential targets are concerned. The setting of one-sided information, i.e., P1 knows everything about P2, is necessary for P2 to derive defense strategies in risk-sensitive games. We call this focused set of games "**2p0s1**".

We claim the following contributions:

- We prove two unique Nash equilibrium (NE) structures for 2p0s1: (1) The equilibrium behavioral strategy for P1 (resp. P2) is $I$-atomic (resp. $(I+1)$-atomic) on their continuous action space, and (2) the equilibrium strategies for P1 and P2 can be computed via separated primal and dual reformulations of the game. Together, these structures collapse the game-tree complexity to at most $I^K$ for P1 and $(I+1)^K$ for P2. In comparison, solving the same game through the lens of IIEFG would have a game-tree complexity of $U^{2K}$, with a discretized action size of $U$ (Fig. 1a).

- We demonstrate how this structural knowledge can significantly accelerate game solving: (1) For value and policy approximation settings where the ground-truth NE is available, we achieve qualitative improvements on solution accuracy and efficiency from SOTA normal- and behavioral-form solvers (CFR+, MMD, Deep-CFR, JPSPG, PPO, R-NaD). (2) We further demonstrate the practical value of the equilibrium structure of 2p0s1 by solving an American football setting where the attacking team needs to strategically conceal their true intention between "RB power push" and "QB throw". While this IIEFG has a complexity of $10^{440}$ even with a coarse mesh of 10 values per action dimension, the atomic structure of its NE makes the game solvable in less than 30 minutes on a M1 Pro Macbook. See Fig. 1b,c.

## 2 RELATED WORK

**2p0s games with incomplete information.** Games where players have missing information only about the game types are called *incomplete-information*. These are a special case of imperfect-information games where nature only plays a chance move at the beginning Harsanyi (1967). The seminal work of Aumann et al. (1995) developed equilibrium strategies for a repeated game with one-sided incomplete information through the "Cav u" theorem, which reveals that belief-manipulating

behavioral strategies optimize value through value convexification. Building on top of this, De Meyer (1996) introduced a dual game reformulation where the behavioral strategy of P2 becomes Markov. This technique helped Cardaliaguet (2007); Ghimire et al. (2024) to establish the value existence proof for 2p0s differential games with incomplete information. Unlike repeated games where belief manipulation occurs only in the first round of the game, differential games may have multiple critical time-state-belief points where belief manipulation is required to achieve equilibrium, depending on the specifications of system dynamics, payoffs, and state constraints (Ghimire et al., 2024). Our paper builds on top of Cardaliaguet (2007); Ghimire et al. (2024), providing a thorough analysis of the convexification nature of behavioral strategies and, as a consequence, their atomic structure.

**IIEFGs.** IIEFGs are partially-observable stochastic games (POSGs) with finite horizons. Significant efforts have been taken to approximate equilibrium of large IIEFGs with finite $U$ (Koller & Megiddo, 1992; Billings et al., 2003; Gilpin & Sandholm, 2006; Gilpin et al., 2007; Sandholm, 2010; Brown & Sandholm, 2019; Brown et al., 2020a; Perolat et al., 2022; Schmid et al., 2023). Regret matching algorithms (Zinkevich et al., 2007; Lanctot et al., 2009; Abernethy et al., 2011; Brown et al., 2019; Tammelin, 2014; Johanson et al., 2012) have computational complexity $\mathcal{O}(U\varepsilon^{-2})$ to $\varepsilon$-Nash, while gradient-based solvers (McMahan, 2011; Perolat et al., 2021; Sokota et al., 2022) have $\mathcal{O}\left(\ln(U)\varepsilon^{-1}\ln\left(\varepsilon^{-1}\right)\right)$ to $\varepsilon$-QRE. All these complexities increase with $U$, and convergence guarantee only exists if the NE lies in the *interior* of the simplex $\Delta(U)$ (Perolat et al., 2021). Critically, this latter assumption does not hold for 2p0s1, as we explain in Sec. 4. For games with continuous action spaces, existing pseudo-gradient based approaches (Martin & Sandholm, 2023; 2024) lack convergence guarantee, and perform poorly in our case studies in both normal- and extensive-form settings.

**Multigrid for value approximation.** Since solving 2p0s1 essentially requires solving a Hamilton-Jacobi PDE, we briefly review multigrid methods for accelerating PDE solving(Trottenberg et al., 2000). In a typical V-cycle solver (Braess & Hackbusch, 1983), a fine-mesh solve is first performed briefly, the residual is then restricted to a coarser mesh where a correction is solved and prolongated back to the fine mesh. Essentially, multigrid uses coarse solves to reduce the low-frequency approximation errors in the PDE solution at low costs, leaving only high-frequency errors to be resolved through the fine mesh. Multigrid was successfully applied to solving Hamilton-Jacobi PDEs for optimal control and differential games (Han & Wan, 2013), although characterizing its computational complexity for such nonlinear PDEs is yet to be done. During value approximation, we extend multigrid to solving Hamilton-Jacobi PDEs underlying 2p0s1.

## 3 PROBLEM STATEMENT

We denote by $\Delta(I)$ the simplex in $\mathbb{R}^I$, $[I] := \{1, ..., I\}$, $a[i]$ the $i$th element of vector $a$, $\nabla_p V$ and $\partial_p V$ the respective gradient and subgradient of function $V$ with respect to $p$. $\|\cdot\|_2$ is the $l_2$-norm and $\|v\|_A := \sqrt{v^T A v}$. Consider a time-invariant dynamical system that defines the evolution of the joint state $x \in \mathcal{X} \subseteq \mathbb{R}^{d_x}$ of P1 and P2 with control inputs $u \in \mathcal{U} \subseteq \mathbb{R}^{d_u}$ and $v \in \mathcal{V} \subseteq \mathbb{R}^{d_v}$, respectively:

$$\dot{x}(t) = f(x(t), u, v). \tag{1}$$

The initial belief of players is set to nature's distribution about the game type. Denote by $\{\mathcal{H}_r^i\}^I$ the joint sets of $I$ behavioral strategies of P1, and $\mathcal{Z}_r$ the set of behavioral strategies of P2. The subscript $r$ indicates the random nature of behavioral strategies: At any infostate $(t, x, p) \in [0, T] \times \mathcal{X} \times \Delta(I)$ and with type $i$, P1 draws an action based on $\eta_i \in \mathcal{H}_r^i$, which is a probability measure over $\mathcal{U}$ parameterized by $(t, x, p)$. P2's strategy $\zeta \in \mathcal{Z}_r$ is a probability measure over $\mathcal{V}$ independent of $i$. Denote by $X_{t_1}^{t_0, x_0, \eta_i, \zeta}$ the random state arrived at $t_1 \in (t_0, T]$ from $(t_0, x_0)$ following $(\eta_i, \zeta)$ and the system dynamics in Eq. 1. With mild abuse of notation, let $(\eta_i(t), \zeta(t))$ denote the random controls at time $t$ induced by $(\eta_i, \zeta)$. P1 accumulates a running cost $l_i(x(t), \eta_i(t), \zeta(t))$ during the game and receives a terminal cost $g_i(X_T^{t_0, x_0, \eta_i, \zeta})$. Together, a type-$i$ P1 minimizes:

$$J_i(t_0, x_0; \eta_i, \zeta) := \mathbb{E}_{\eta_i, \zeta}\left[g_i\left(X_T^{t_0, x_0, \eta_i, \zeta}\right) + \int_{t_0}^T l_i(x(s), \eta_i(s), \zeta(s))ds\right],$$

while P2 maximizes $J(t_0, x_0, p_0; \{\eta_i\}, \zeta) = \mathbb{E}_{i \sim p_0}[J_i]$. We say the game has a value $V$ if and only if the upper value $V^+(t_0, x_0, p_0) = \inf_{\{\eta_i\}} \sup_\zeta J$ and the lower value $V^-(t_0, x_0, p_0) = \sup_\zeta \inf_{\{\eta_i\}} J$ are equal: $V = V^+ = V^-$. The game is proven to have a value under the following sufficient conditions (Cardaliaguet, 2007):

A1. $\mathcal{U} \subseteq \mathbb{R}^{d_u}$ and $\mathcal{V} \subseteq \mathbb{R}^{d_v}$ are compact and finite-dimensional.

A2. $f : \mathcal{X} \times \mathcal{U} \times \mathcal{V} \to \mathcal{X}$ is $C^1$ and has bounded value and first-order derivatives.

A3. $g_i : \mathcal{X} \to \mathbb{R}$ and $l_i : \mathcal{X} \times \mathcal{U} \times \mathcal{V} \to \mathbb{R}$ are bounded and Lipschitz continuous.

A4. Isaacs' condition holds for the Hamiltonian $H : \mathcal{X} \times \mathbb{R}^{d_x} \times \Delta(I) \to \mathbb{R}$:

$$
\begin{aligned}
H(x, \xi, p) &:= \min_{u \in \mathcal{U}} \max_{v \in \mathcal{V}} f(x, u, v)^\top \xi + \mathbb{E}_{i \sim p}[l_i(x, u, v)] \\
&= \max_{v \in \mathcal{V}} \min_{u \in \mathcal{U}} f(x, u, v)^\top \xi + \mathbb{E}_{i \sim p}[l_i(x, u, v)].
\end{aligned}
\tag{2}
$$

Isaacs' condition allows any complete-information versions of this game to have *pure* Nash equilibria, including *nonrevealing* games where neither player knows the actual game type.

A5. Both players have full knowledge about $f, \{g_i\}_{i=1}^I, \{l_i\}_{i=1}^I, p_0$. Control inputs and states are fully observable. Players have perfect recall.

Our goal is to compute a Nash equilibrium (NE) $(\{\eta_i^\dagger\}, \zeta^\dagger)$ that attains $V$, given the game $G = \{\mathcal{X}, (\mathcal{U}, \mathcal{V}), \{g_i\}, \{l_i\}, f, T\}$.

## 4 A PRIMAL-DUAL REFORMULATION OF THE GAME

In this section, we introduce discrete-time primal and dual reformulations of $G$, denoted by $G_\tau$ and $G_\tau^*$, respectively, for which dynamic programming principles (DP) exist. We show that P1's equilibrium behavioral strategy $\{\eta_{i,\tau}^\dagger\}$ in $G_\tau$ is $I$-atomic, i.e., $\eta_{i,\tau}^\dagger$ concentrates on at most $I$ actions in $\mathcal{U}$, and P2's strategy $\zeta_\tau^\dagger$ in $G_\tau^*$ is $(I+1)$-atomic. Then we show that $(\{\eta_{i,\tau}^\dagger\}, \zeta_\tau^\dagger)$ approaches the Nash equilibrium of the differential game $G$ as the time interval $\tau \to 0^+$.

**The primal game $G_\tau$.** $G_\tau$ is a discrete-time Stackelberg version of $G$ where P2 plays a pure best response *after* P1 announces its next action. Let the value of $G_\tau$ be $V_\tau$. $V_\tau$ satisfies the following DP for $(t, x, p) \in [0, T] \times \mathcal{X} \times \Delta(I)$:

$$
V_\tau(t, x, p) = \min_{\{\eta_i\} \in \{\mathcal{H}_r\}^I} \mathbb{E}_{i \sim p, u \sim \eta_i} \left[ \max_{v \in \mathcal{V}} V_\tau(t + \tau, x'(u, v), p'(u)) + \tau l_i(x, u, v) \right],
\tag{3}
$$

with a terminal boundary $V_\tau(T, x, p) = \sum_i p[i] g_i(x)$. For small enough $\tau$, $x'(u, v) = x + \tau f(x, u, v)$. $p'(u)$ is the Bayes update of the public belief after P1 announces $u$: $p'(u)[i] = \eta_i(u; t, x, p) p[i] / \bar{\eta}(u; t, x, p)$, where $\bar{\eta}(u; t, x, p) = \sum_{i \in [I]} \eta_i(u; t, x, p) p[i]$ is the marginal over $\mathcal{U}$ across types. Note that P2's equilibrium behavioral strategy cannot be derived from Eq. 3.

**The dual game $G_\tau^*$.** For P2's strategy, we need a separate DP where P2 announces its next action and P1 best responses to it. We do so by first introducing the convex conjugate $V^*$ of the value:

$$
\begin{aligned}
V^*(t_0, x_0, \hat{p}_0) &:= \max_p p^T \hat{p}_0 - V(t_0, x_0, p) = \max_p p^T \hat{p}_0 - \sup_{\zeta \in \mathcal{Z}_r} \inf_{\{\eta_i\} \in \{\mathcal{H}_r\}^I} \mathbb{E}_{\eta_i, \zeta, i} \left[ J_i \right] \\
&= \max_p \inf_{\zeta \in \mathcal{Z}_r} \sup_{\{\eta_i\} \in \{\mathcal{H}_r\}^I} p^T \hat{p}_0 - \mathbb{E}_{\eta_i, \zeta, i} \left[ J_i \right] = \inf_{\zeta \in \mathcal{Z}_r} \sup_{\eta \in \mathcal{H}} \max_{i \in \{1, \dots, I\}} \left\{ \hat{p}_0[i] - \mathbb{E}_\zeta \left[ J_i \right] \right\}.
\end{aligned}
\tag{4}
$$

Eq. 4 describes a dual game $G^*$ with complete information, where the strategy space of P1 becomes $\mathcal{H} \times [I]$, i.e., the game type is now chosen by P1 rather than the nature. We prove in App. A that P2's equilibrium in the dual game is also an equilibrium in the primal game if $\hat{p}_0 \in \partial_p V(t_0, x_0, p_0)$, and such $\hat{p}_0[i]$ represents the loss of type-$i$ P1 should it play best responses to P2's equilibrium strategy. Therefore $\hat{p}_0[i] - \mathbb{E}_\zeta[J_i]$ measures P2's risk, and P2's equilibrium strategy minimizes the worst-case risk across all game types. We now introduce a discrete-time version of the dual game $G_\tau^*$ where P1 plays a pure best response after P2 announces their action at each time step. Let the value of $G_\tau^*$ be $V_\tau^*$, the dual DP is:

$$
V_\tau^*(t, x, \hat{p}) = \min_{\zeta, \hat{p}'(\cdot)} \mathbb{E}_{v \sim \zeta} \left[ \max_{u \in \mathcal{U}} V_\tau^*(t + \tau, x'(u, v), \hat{p}'(v) - \tau l(x, u, v)) \right],
\tag{5}
$$

with a terminal boundary $V^*(T, x, \hat{p}) = \max_{i \in [I]} \{\hat{p}[i] - g_i(x)\}$. Here $\hat{p}'(\cdot) : \mathcal{V} \to \mathbb{R}^I$ is constrained by $\mathbb{E}_{v \sim \zeta}[\hat{p}'(v)] = \hat{p}$ (similar to the martingale nature of $p$ in $G_\tau$), and $l(u, v)[i] = l_i(u, v)$.

**Equilibrium strategies of $G_\tau$ and $G_\tau^*$ are atomic.** Our first theoretical result is Thm. 4.1, which states that P1's strategy that solves $G_\tau$ is $I$-atomic, and P2's strategy that solves $G_\tau^*$ is $(I+1)$-atomic:

**Theorem 4.1.** *The RHS of Eq. 3 can be reformulated as*

$$\min_{\{u^k\},\{\alpha_i^k\}} \max_{\{v^k\}} \sum_{k=1}^{I} \lambda^k \Big( V(t+\tau, x+\tau f(x, u^k, v^k), p^k) + \tau \mathbb{E}_{i \sim p^k}[l_i(x, u^k, v^k)] \Big)$$

$$s.t. \quad u^k \in \mathcal{U}, \ v^k \in \mathcal{V}, \alpha_i^k \in [0,1], \ \sum_{k=1}^{I} \alpha_i^k = 1, \ \lambda^k = \sum_{i=1}^{I} \alpha_i^k p[i], \ p^k[i] = \frac{\alpha_i^k p[i]}{\lambda^k}, \quad \forall i, k \in [I],$$

$$(\text{P}_1)$$

*i.e., $\eta_{i,\tau}^\dagger$ concentrates on actions $\{u^k\}_{k=1}^I$ for $i \in [I]$. The RHS of Eq. 5 can be reformulated as*

$$\min_{\{v^k\},\{\lambda^k\},\{\hat{p}^k\}} \max_{\{u^k\}} \sum_{k=1}^{I+1} \lambda^k \left( V^*(t+\tau, x+\tau f(x, u^k, v^k), \hat{p}^k - \tau l(x, u^k, v^k)) \right)$$

$$(\text{P}_2)$$

$$s.t. \quad u^k \in \mathcal{U}, \ v^k \in \mathcal{V}, \ \lambda^k \in [0,1], \ \sum_{k=1}^{I+1} \lambda^k \hat{p}^k = \hat{p}, \ \sum_{k=1}^{I+1} \lambda^k = 1, \quad k \in [I+1].$$

**Proof sketch.** (1) Using Isaacs' condition, we show that P2's best response in Eq. 3 is implicitly governed by P1's action $u$, and $u$ is in turn governed by the posterior belief $p'$. (2) With this insight, we can rewrite Eq. 3 as $V_\tau(t, x, p) = \min_\nu \int_{\Delta(I)} \tilde{V}_\tau(t, x, p') \nu(dp')$, where we control a pushforward density $\nu(dp')$ for the posterior belief to be $p'$. $\nu$ is subjected to $\int_{\Delta(I)} p' \nu(dp') = p$. For each $p'$, $(u(p'), v(p'))$ is found by solving $\tilde{V}(t, x, p') = \min_{u \in \mathcal{U}} \max_{v \in \mathcal{V}} V_\tau(t+\tau, x', p') + \mathbb{E}_{i \sim p'} l_i$. Here $\tilde{V}(t, x, p')$ is P1's loss should both players play pure strategies at $(t, x, p')$ within $[t, t+\tau)$, and its existence is guaranteed by Isaacs' condition. Since playing pure strategies does not change the public belief, we call $\tilde{V}_\tau$ the *non-revealing* value. (3) We can then show that $\min_\nu \int_{\Delta(I)} \tilde{V}_\tau(p') \nu(dp')$ is a convexification of $\tilde{V}_\tau$ in $\Delta(I)$. Since convexification requires at most $I$ vertices in $\Delta(I)$, $\nu$ is at most $I$-atomic. Since $\nu$ determines $\eta_{i,\tau}^\dagger$, the latter is also $I$-atomic. (4) A similar argument can be made for $G_\tau^*$, in which case the convexification is with respect to the dual variable $\hat{p}$. Since $\hat{p}$ is defined on $\mathbb{R}^I$ rather than constrained on $\Delta(I)$, $\zeta^\dagger$ is at most $(I+1)$-atomic. Proof in App. B.

**An example.** To support the proof sketch, we use Fig. 2 to illustrate why an equilibrium behavioral strategy is atomic. Here $I = 2$, thus the value is defined on $\Delta(2)$ for fixed $(t, x)$. To solve $\min_\nu \int_{\Delta(2)} \tilde{V}_\tau(p') \nu(dp')$, we scan the non-revealing value $\tilde{V}$ across $\Delta(2)$. One notices that if $\tilde{V}$ is not convex in $p$, it is always possible for P1 to achieve a lower loss by convexifying $\tilde{V}$ through a mixed strategy, leading to $\text{P}_1$. In the figure, P1 identifies $[\lambda^a, \lambda^b]^\top \in \Delta(2)$ and $\{p^a, p^b\}$ such that $\lambda^a p^a + \lambda^b p^b = p$. Solving the non-revealing pure NE $(u^k, v^k)$ for each $k \in \{a, b\}$, P1's mixed strategy that convexifies $\tilde{V}$ is to play action $u^k$ with probability $\alpha_i^k = p^k[i]\lambda^k/p[i]$ if P1 is type-$i$. By announcing this strategy, the

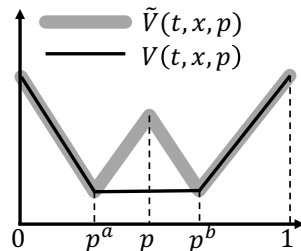

Figure 2: Value convexification causes NE to be atomic.

public belief shifts to $p^k$ via the Bayes' rule when P2 observes action $u^k$. As a result, P1 receives $V(p) = \lambda^a \tilde{V}(p^a) + \lambda^b \tilde{V}(p^b)$ on the convex hull of $\tilde{V}(p)$.

**Remarks.** (1) The atomic structure of the equilibrium strategies was first discovered for 2p0s repeated games with one-sided information (Aumann et al., 1995; De Meyer, 1996). Our new contribution is in explaining its presence in differential games and in demonstrating its significant utilities in improving the effectiveness of multiagent reinforcement learning and model predictive control schemes. (2) P1's actions in 2p0s1 simultaneously advance system states and achieve signaling. The deterministic dynamics allows precise belief control (e.g., $(p^a, p^b)$ in Fig. 2) for value convexification. For games with stochastic dynamics/observations, however, P1 will not be able to precisely control the belief, and the convex Bellman operators ($\text{P}_1$ and $\text{P}_2$) become lower bounds of the value. Finding tight upper-bounding operators that preserve atomic NEs is left for future work.

$(\{\eta_{i,\tau}^{\dagger}\}, \zeta_{\tau}^{\dagger})$ **approaches NE of** $G$. Next we present Thm. 4.2, which proves that $(\{\eta_{i,\tau}^{\dagger}\}, \zeta_{\tau}^{\dagger})$ computed from $P_1$ and $P_2$ approaches the equilibrium of $G$ when $\tau$ is sufficiently small. The theorem uses P1's loss in $G$ when they adapt $\{\eta_{i,\tau}^{\dagger}\}$ to $G$ by playing a constant action within each time interval. This loss is denoted by $V_1(t, x, \hat{p}) := \max_{\zeta \in \mathcal{Z}} J(t, x, p; \{\eta_{i,\tau}^{\dagger}\}, \zeta)$. Similarly, P2's loss in $G^*$ when they adapt $\zeta_{\tau}^{\dagger}$ to $G$ is denoted by $V_1^*(t, x, p) := \max_{\{\eta_i\} \in \{\mathcal{H}^i\}^I} J^*(t, x, \hat{p}; \{\eta_i\}, \zeta_{\tau}^{\dagger})$.

**Theorem 4.2.** *If A1-5 hold, then for any $\epsilon > 0$, there exists $\tau^* > 0$, such that for any $\tau \in (0, \tau^*]$ and the corresponding $(\{\eta_{i,\tau}^{\dagger}\}, \zeta_{\tau}^{\dagger})$, $V_1(t, x, p) - V(t, x, p) \in [0, \epsilon]$ for any $(t, x, p) \in [0, T] \times \mathcal{X} \times \Delta(I)$, and $V_1^*(t, x, \hat{p}) - V^*(t, x, \hat{p}) \in [0, \epsilon]$ for any $(t, x, \hat{p}) \in [0, T] \times \mathcal{X} \times \mathbb{R}^I$.*

**Proof sketch.** We first note that $V \leq V_1$ because P1's NE in $G_\tau$ is not necessarily a NE in $G$, i.e., P1 can get a better value in $G$ using $\{\eta_i^{\dagger}\}$ than $\{\eta_{i,\tau}^{\dagger}\}$. We also have, from Cardaliaguet (2009), the uniform convergence of $V_\tau \to V$ as $\tau \to 0^+$. Then, it remains to show that $V_1$ is "not far" from $V_\tau$. To do so, we first show that the per-stage value gap between $V_1$ and $V_\tau$ is within $\mathcal{O}(\tau^2)$, which accumulates over time into $\mathcal{O}(\tau)$: $V_1 \leq V_\tau + \mathcal{O}(\tau)$. Then as $\tau \to 0^+$, combined with the uniform convergence of $V_\tau$, we conclude that the gap $V_1 - V \in [0, \epsilon]$. This implies that the equilibrium $\{\eta_{i,\tau}^{\dagger}\}$ derived in $G_\tau$ is $\epsilon$-optimal in the original game $G$ as long as $\tau$ is small enough. Full proof in App. C. The same techniques can be applied to $G_\tau^*$.

With Thms. 4.1 and 4.2, and given that $G$ is differential ($\tau \to 0^+$), we now proved that NEs of $G$ are atomic, and can be approximated via Eqs. $P_1$ and $P_2$ with sufficiently small $\tau$.

# 5 INCORPORATING THE ATOMIC STRUCTURE INTO MARL AND CONTROL

We study the efficacy of atomic NEs when applied to (1) value approximation, (2) model-free MARL, and (3) control settings. For (1), we approximate value and strategies in the entire $[0, T] \times \mathcal{X} \times \Delta(I)$ through $P_1$ and $P_2$, and introduce multigrid to further reduce the compute. We compare with discrete-action solvers CFR+, MMD, CFR-BR, and continuous-action solver JPSPG. For (2) and (3), we solve NE strategies for a fixed initial $(t_0, x_0, p_0)$. We compare with discrete-action solvers MMD, PPO, and R-NaD. We call the proposed continuous-action mixed-strategy solvers CAMS, CAMS-DRL, and CAMS-MPC for value approximation, MARL, and control, respectively.

## 5.1 VALUE APPROXIMATION

**CAMS for** $G_\tau$. We discretize time as $\{k\tau\}_{k=0}^{K}$, $\tau = T/K$. Let $\mathcal{S} = \{(x, p)_i\}_{i \in [|\mathcal{S}|]}$ be a collocation set. We solve $P_1$ starting from $t = (K-1)\tau$ at all collocation points in $\mathcal{S}$. The resultant nonconvex-nonconcave minimax problems have size $(\mathcal{O}(I(I + d_u)), \mathcal{O}(Id_v))$ and are solved by DS-GDA (Zheng et al., 2023), which guarantees sublinear convergence on nonconvex-nonconcave problems. To generalize value and strategies across $\mathcal{X} \times \Delta(I)$, a value network is trained on the minimax solutions and used to formulate the next round of minimax at $t - \tau$. $G_\tau^*$ is solved similarly.

**Computational challenge.** From Thm. 4.2, large $K$ (small $\tau$) is required for strategies derived from $P_1$ and $P_2$ to be good approximations of the NE. Yet suppressing the value prediction error at $t = 0$ requires a computational complexity exponential in $K$. Specifically, let $\hat{V}_0(x, p) : \mathcal{X} \times \Delta(I) \to \mathbb{R}$ be the trained value networks at $t = 0$, we have the following result (see proof in App. D):

**Theorem 5.1.** *Given $K$, a minimax approximation error $\epsilon > 0$, a prediction error threshold $\delta > 0$, there exists $C \geq 1$, such that with a computational complexity of $\mathcal{O}(K^3 C^{2K} I^2 \epsilon^{-4} \delta^{-2})$, CAMS achieves*

$$\max_{(x,p) \in \mathcal{X} \times \Delta(I)} |\hat{V}_0(x, p) - V(0, x, p)| \leq \delta. \tag{6}$$

A similar result applies to the dual game. Zanette et al. (2019) discussed a linear value approximator that achieves $C = 1$. However, their method requires solving a linear program (LP) for every inference $\hat{V}_t(x, p)$ if $(x, p)$ does not belong to the training set $\mathcal{S}$. In our context, incorporating their method would require auto-differentiating through the LP solver for each descent and ascent steps in minimax, which turned out to be too expensive. To this end, we introduce a multigrid scheme to reduce the cost for games with a large $K$.

**Multigrid.** Since strategies at time $t$ are implicitly nonlinear functions of the value at $t + \tau$, the Hamilton-Jacobi PDEs underlying $P_1$ and $P_2$ are nonlinear. Our method extends the Full Approximation Scheme (FAS) for solving nonlinear PDEs (Trottenberg et al., 2000; Henson et al., 2003). A two-grid FAS has four steps: (1) Restrict the fine-grid approximation and its residual; (2) solve the coarse-grid problem using the fine-grid residual; (3) compute the coarse-grid correction; (4) prolong the coarse-grid correction to fine-grid and add the correction to fine-grid approximation. For conciseness, we focus on the primal problem. Let $\hat{V}_t^l$ be the value network for time $t$ on grid size (time interval) $l$. Let $\mathcal{R}^l$ be the restriction operator to a coarser grid with size $2l$: $\mathcal{R}^l(\hat{V}_t^l) = (\hat{V}_t^l + \hat{V}_{t+l}^l)/2$ is the value restriction from $l$ to $2l$. Similarly, we define the prolongation operators $\mathcal{P}^{2l}$ as $\mathcal{P}^{2l}(\hat{V}_t^{2l}) = \hat{V}_t^{2l}$ if $t \in \mathcal{T}^{2l}$ or $\hat{V}_{t+l}^{2l}$ otherwise, where $\mathcal{T}^{2l} := \{n \cdot 2l : n \in \mathbb{N}_0, n < T/2l\}$. Let $\mathbb{O}^l(t, x, p; \hat{V})$ solve $P_1$ at $(t, x, p)$ where $\hat{V}$ is the value at $t + l$, and outputs an approximation for $V(t, x, p)$. The dataset $\{(t, x^{(j)}, p^{(j)}, \mathbb{O}^l(t, x^{(j)}, p^{(j)}; \hat{V}_{t+l}^l))\}$ is used to train $\hat{V}_t^l(\cdot, \cdot)$. Let $r_t^l(x, p) = \hat{V}_t^l(x, p) - \mathbb{O}^l(t, x, p; \hat{V}_{t+l}^l)$ be the residual. To achieve $r_t^l(x, p) \approx 0$ for all $(t, x, p) \in \mathcal{T}^l \times \mathcal{X} \times \Delta(I)$, we restrict the fine grid approximations and residuals to the coarse grid and solve to determine the corrections. To do so, let $e_t^l(x, p)$ be the correction in grid $l$ at $(t, x, p)$. The coarse-grid problem is

$$\underbrace{\mathcal{R}^l r_t^l}_{\text{residual}} = \underbrace{\mathbb{O}^{2l}(t, x, p; \mathcal{R}^l \hat{V}_{t+2l}^l + e_{t+2l}^{2l}) - \left(\mathcal{R}^l \hat{V}_t^l + e_t^{2l}(x, p)\right)}_{\text{coarse-grid eq. w/ corrections}} - \underbrace{\left(\mathbb{O}^{2l}(\mathcal{R}^l \hat{V}_{t+2l}^l) - \mathcal{R}^l \hat{V}_t^l\right)}_{\text{coarse-grid eq. w/o corrections}}, \quad (7)$$

where $e_t^{2l}(x, p)$ is computed backward from $T - 2l$ using $e_T^{2l} = 0$:

$$e_t^{2l}(x, p) = \mathbb{O}^{2l}(t, x, p; \mathcal{R}^l \hat{V}_{t+2l}^l + e_{t+2l}^{2l}) - \mathbb{O}^{2l}(\mathcal{R}^l \hat{V}_{t+2l}^l) - \mathcal{R}^l r_t^l. \quad (8)$$

This correction ensures consistency: If $\hat{V}_t^l = V(t, \cdot, \cdot)$ for all $t \in \mathcal{T}^l$, $e_t^{2l}(\cdot, \cdot) = 0$ for all $t \in \mathcal{T}^{2l}$. The coarse grid corrections are prolonged to the fine grid to update the fine-grid value approximation. Note that from Eq. 8, computing the coarse correction in our case requires two separate minimax calls with similar loss formulations. We further accelerate the multigrid solver by warm-starting these minimax problems using the recorded minimax solution derived from the fine grid (during the residual computation).

## 5.2 MODEL-FREE MULTIAGENT REINFORCEMENT LEARNING

Recent studies (Rudolph et al., 2025; Sokota et al., 2022) showed that policy gradient methods for MARL, such as PPO and MMD, can effectively solve IIEFGs with properly tuned hyperparameters. We show that the atomic structure can be directly applied to this unified model-free framework with minimal code changes, while yielding significant solution improvement for 2p0s1. For CAMS-DRL, the policy network of P1 takes in the infostate $(t, x, p)$ and outputs $I$ logit vectors $\ell_k \in \mathbb{R}^I$ and $I$ action prototypes $u^k \in \mathcal{U}$ for $k \in [I]$. Each logit vector $\ell_k$ is transformed via softmax to define the behavioral strategy of type-$i$ P1, i.e., the probability $\eta_i(u^k; t, x, p)$ of choosing action $u^k$. The policy network for P2 takes in $(t, x, p)$ and outputs a single logit $\ell \in \mathbb{R}^{I+1}$ and action prototypes $\mu^k$ for $k \in [I + 1]$. We directly solve the NE of these policy models using PPO and MMD.

## 5.3 MODEL PREDICTIVE CONTROL

When dynamics $f$ is known and differentiable, and an initial state $(t_0, x_0, p_0)$ is given, we can formulate the primal (resp. dual) minimax objective as the sum over $I^K$ (resp. $(I + 1)^K$) game tree paths (Fig. 1a). CAMS-MPC builds the computational graph for the entire tree where the policy networks follow CAMS-DRL, and applies DS-GDA to this differentiable loss. This method is feasible thanks to the atomic structure of NEs and for small $I$. Since P1 in 2p0s1 games reveals their payoff mid-game, the game tree further collapses after information revelation. E.g., Hexner's game (see below) has a proven game-tree complexity of $I$, where P1 plays a fixed nonrevealing strategy before splitting to $I$ type-dependent revealing strategies. Due to this collapse, modeling strategies using neural networks turns out to be more effective for the convergence to NEs than using infostate-wise parameterization.

## 6 EMPIRICAL VALIDATION

**Hexner's game.** We introduce Hexner's game (Hexner, 1979; Ghimire et al., 2024) to compare CAMS variants with baselines on solution quality and computational cost. This game has an analytical NE. The dynamics is $\dot{x}_j = A_j x_j + B_j u_j$ for $j = [2]$, where $x_j \in \mathcal{X}_j$, $u_j \in \mathcal{U}_j$, and $A_j$ and $B_j$ are known matrices. The target state of P1 is $z\theta$ where $\theta$ is drawn with distribution $p_0$ from $\Theta$, $|\Theta| = I$, and $z \in \mathbb{R}^{d_x}$ is fixed and public. The expected payoff to P1 is:

$$J(\{\eta_i\}, \zeta) = \mathbb{E}_{i \sim p_0} \left[ \int_0^T \|\eta_i(t)\|_{R_1}^2 - \|\zeta(t)\|_{R_2}^2 dt + \|x_1(T) - z\theta_i\|_{K_1}^2 - \|x_2(T) - z\theta_i\|_{K_2}^2 \right],$$

where $R_1, R_2, K_1, K_2 \succ 0$ are control- and state-penalty matrices. The goal of P1 is to get closer to the target $z\theta$ than P2. To take information advantage, P1 needs to decide when to home-in to and reveal the target. **Analytical NE:** There exists a critical time $t_r := t_r(T, \{A_j\}, \{B_j\}, \{R_j\}, \{K_j\})$. If $t_r \in (0, T)$, P1 moves towards $\mathbb{E}[\theta]$ as if it does not know the actual target until $t_r$ when it fully reveals the target, i.e., value convexification happens at $t_r$. If $t_r \leq 0$, P1 homes towards the actual target at $t = 0$. P2's NE mirrors P1. See proof in App. F.1.

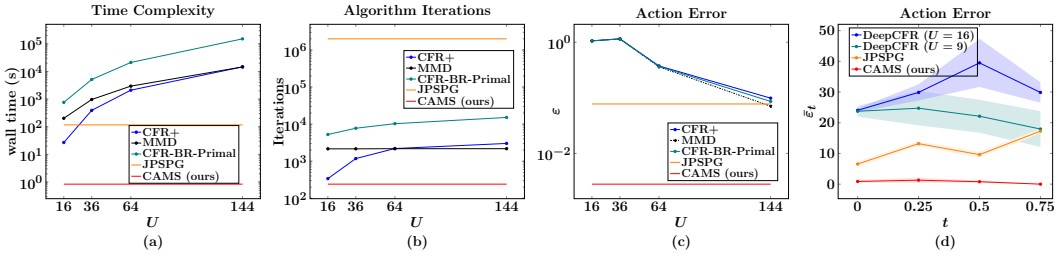

Figure 3: (a-c) Comparisons b/w CAMS, JPSPG, CFR+, MMD, CFR-BR-Primal on 1-stage Hexner's game. (d) Comparison b/w CAMS, JPSPG, and DeepCFR on 4-stage Hexner's w/ similar compute.

### 6.1 EFFECT OF ATOMIC NE ON VALUE APPROXIMATION

**Action-size-invariant convergence of CAMS.** We use a normal-form Hexner's game with $\tau = T$ and a fixed initial state $x_0 \in \mathcal{X}$ to demonstrate that baseline algorithms suffer from increasing discrete action sizes while CAMS does not. The baselines are SOTA normal-form solvers including CFR+ (Tammelin, 2014), MMD (Sokota et al., 2022), JPSPG (Martin & Sandholm, 2024), and a modified CFR-BR (Johanson et al., 2012) (dubbed CFR-BR-Primal), where we focus on converging P1's strategy and only compute P2's best responses. Among these, only JPSPG naturally handles continuous action spaces. All baselines except JPSPG are standard implementations in OpenSpiel (Lanctot et al., 2019). JPSPG implementation details are in App. G.4. The normal-form primal game has a trivial ground-truth strategy where P1 goes directly to its target. For visualization, we use $d_x = 4$ (position and velocity in 2D). For baselines except JPSPG, we use discrete action sets defined by 4 grid sizes so that $U = |\mathcal{U}_j| \in \{16, 36, 64, 144\}$. All algorithms terminate when a threshold of $NashConv(\pi_i) = \max_{\pi_i'} V_i(\pi_i') - V_i(\pi_i)$ is met. For conciseness, we only consider solving P1's strategy and thus use P1's NashConv. We set the threshold to $10^{-3}$ for baselines and use a more stringent threshold of $10^{-5}$ for CAMS. We then use DeepCFR and JPSPG as baselines for a 4-stage game where $T = 1$ and $\tau = 0.25$. DeepCFRs were run for 1000 CFR iterations (resp.

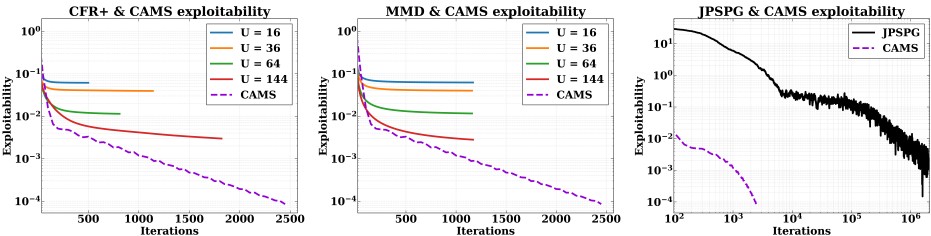

Figure 4: Exploitability comparison in the normal-form Hexner's game.

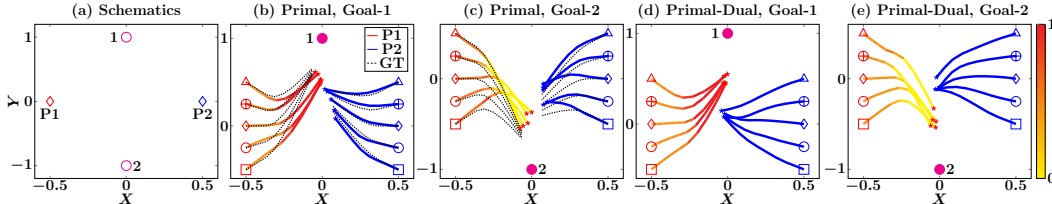

Figure 5: (a) Hexner's game schematics: one goal is selected out of two possible goals: Goal-1 and Goal-2, and communicated to P1. (b-e) Sample trajectories for the primal game (b-c) where P1 plays Nash and P2 plays best response, and primal-dual game (d-e) where both players play Nash. Dotted lines are ground-truth Nash. Color shades indicate evolution of public belief (Pr[Goal is 1]). Filled Magenta circle represents the true goal. Initial position pairs are marked with the same markers.

100) with 10 (resp. 5) traversals for $U = 9$ (resp. 16). The wall-time costs for game solving are 17 hours using CAMS (baseline), 24 hours for JPSPG, 29 hours ($U = 9$) and 34 hours ($U = 16$) using DeepCFR, all on an A100 GPU. More details on experiment settings can be found in App. G.3. Furthermore, JPSPG was run for $2 \cdot 10^8$ iterations, where each iteration consists of solving a game with a random initial state and type, and performing a strategy update.

Fig. 3 summarizes the comparisons with baselines. For the normal-form game, we compare both computational cost and the expected action error $\varepsilon$ from the ground-truth action of P1: $\varepsilon(x_0) :=$ $\mathbb{E}_{i \sim p_0} \left[ \sum_{k=1}^{|\mathcal{U}|} \alpha_{ki} ||u_k - u_i^*(x_0)||_2 \right]$, where $u_i^*(x_0)$ is the ground truth for type $i$ at $x_0$. In the 4-stage game, we compare the expected action errors at each time-step: $\bar{\varepsilon}_t := \mathbb{E}_{x_t \sim \pi} \left[ \varepsilon(x_t) \right]$, where $\pi$ is the strategy learned by the respective learning method. For each strategy, we estimate $\{\bar{\varepsilon}_t\}_{t=1}^4$ by generating 100 trajectories with initial states uniformly sampled from $\mathcal{X}$. In terms of computational cost, all baselines (except JPSPG) have complexity and wall-time costs increasing with $U$, while CAMS is invariant to $U$. We also compare exploitability of CFR+, MMD, and JPSPG against CAMS for the normal-form Hexner's game in Fig. 4. For a fair comparison, best responses are computed in the continuous action spaces.

With similar or less compute, CAMS achieves significantly better strategies than DeepCFR and JPSPG in the 4-stage game. Sample trajectories for the 4-stage game are visualized in App. G. Fig. 5 compares the ground-truth vs. approximated NEs for 10-stage games with different initial states. While approximation errors exist, CAMS successfully learns the target-concealing behavior of P1. Averaging over 50 trajectories derived from CAMS, P1 conceals the target until $t_r = 0.60s \pm 0.06s$ (compared to the ground-truth $t_r = 0.5s$).

**CAMS scalability with multigrid.** We solve Hexner's games using CAMS with and without multi-grid to demonstrate the scalability of our approach and the effect of multigrid. We report the runtime on one H100 GPU for 4-, 10-, and 16-stage games in Tab. 1. We run the 2-level multigrid on the 4- and 10-stage games, and 4-level multigrid on the 16-stage game (see Alg. 1 for pseudo code). We report result-ing trajectories in App. E.

Table 1: Runtime comparison of CAMS with and without multigrid

| # time steps | no multigrid | multigrid ↓ |
|---|---|---|
| 4 | 9.3 hrs | **2.3** hrs |
| 10 | 27.6 hrs | **10.9** hrs |
| 16 | 46.2 hrs | **17.8** hrs |

## 6.2 EFFECT OF ATOMIC NE ON MODEL-FREE MARL

For model-free MARL, we compare CAMS-DRL with MMD, PPO and R-NaD with discrete actions of size 100 formed by pairing each of the ten linearly spaced $x$-direction acceleration between $(-1, 1)$ and $y$-direction acceleration between $(-4, 4)$. We test the policy convergence in the normal-form game by comparing P1's learned policy every 8192 steps (corresponds to 1 iteration in Fig. 6(a)) against the ground truth policy. MMD, PPO, and R-NaD are implemented as in Rudolph et al. (2025). Results in Fig. 6 (a-b), which uses the same comparison metric as outlined in Sec. 6.1, show that CAMS-DRL approximates NEs accurately, while PPO and MMD fail. R-NaD, on the other hand, shows better performance than MMD and PPO with little to no hyperparameter tuning (see App. H). This contradicts with observations in Rudolph et al. (2025). We conjecture that PG methods such as PPO and MMD have difficulty converging to non-interior solutions, which is the case in 2p0s1 due to

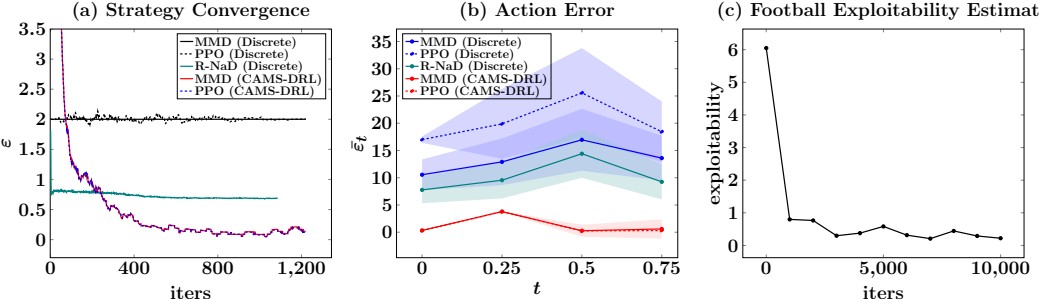

Figure 6: Comparisons b/w CAMS-DRL and standard PG methods on (a) 1-stage and (b) 4-stage games. (c) Exploitability estimate of the strategies in the Football game at different iterations.

the atomic structure of NEs. We performed the same comparison for a 4-stage game, with a discrete action size of 100 composed by pairing linearly spaced $x$- and $y$-direction accelerations, all between $(-12, 12)$. Results in Fig. 6b show significant improvement in solution accuracy when the atomic structure is exploited.

### 6.3 EFFECT OF ATOMIC NE ON MPC

**Hexner's game.** With known differentiable dynamics and $I = 2$, a 10-stage Hexner's (primal) game has at most $2^{10}$ rollouts, making exact policy gradient computation feasible. Using DS-GDA, the convergence to the ground-truth NE takes 5 minutes on a M1 Pro Macbook. See results in App. I.

**Football game.** The independence of the game-tree complexity from the action space allows us to solve games beyond what IIEFG solvers can afford. Here we model an 11-vs-11 American Football play as a short, two-team game in a 2D plane. The horizontal axis represents "downfield" progress and the vertical axis is sideline-to-sideline. Each player follows double-integrator kinematics and chooses acceleration at discrete time steps. Physical contact is captured by a smooth "merge" weight that grows as opponents approach, softly blending their velocities and attenuating their ability to apply new acceleration, so tackles emerge continuously rather than through hard impulses. The offense has two types ($I = 2$): an "RB power push" that prefers the RB to advance straight upfield, and a "QB throw" that rewards whichever offensive player gets furthest downfield. With loose bounds on velocity and acceleration, the soft tackle dynamics is control-affine, allowing Isaacs' condition to hold. Together with differentiable rollouts, the game settings satisfy A1-5, leading to atomic NE. With $K = 10$ and known differentiable dynamics, it is feasible to solve P1's (and then P2's) strategy by applying DS-GDA directly to the minimax objective constituting all $I^K$ $((I + 1)^K)$ rollouts. See App. J for detailed game settings. The convergence takes 30 minutes on a M1 Pro Macbook. The resulting plays are summarized in Fig. 1b,c and animated in the github repo. Qualitatively, the results resemble real football tactics: the offense either tries to push through the defense (aka inside zone play), or goes out wide by faking a move (aka waggle play), while the defense, in response, either close-in on the player with the ball possession or stay back to guard. Importantly, our results show that the offense conceal their play selection for 0.5 seconds, which is comparable to coaching analyses that estimate roughly a one-second window before the play becomes clear (Grabowski, 2020). Quantitatively, we estimate the exploitability of the strategies and plot its convergence in Fig. 6c. We also provide a comparison of resulting trajectories in App. J.1 when each player switches to their respective BR policy to show that the final policies are minimally exploitable.

### 7 CONCLUSION

Unlike general IIEFGs where NE strategies are distributions over the entire action space, we showed that 2p0s1 games enjoy an atomic NE structure when P1 can precisely control the public belief, leading to an exponentially reduced game-tree complexity. We demonstrated the utilities of this NE structure in solving games with continuous action spaces in model-free and model-based modes, in terms of computational cost and solution quality. Our methods enable fast approximation of deceptive and counter-deception *team* strategies, with potential applications to sports, missile/drone defense, and risk-sensitive robotics applications, tailored for specific team dynamics, action spaces, and task specifications.

ETHICS STATEMENT

**LLM Usage.** Large language models (LLMs) were sparingly used for (1) polishing the writing, (2) code generation for visualization, and (3) some proof steps for Thm. 4.1 and Thm. 4.2. All codes and proofs are verified by authors.

**Impact.** This work examines strategic behavior in sequential decision-making and how deceptive tactics can arise in autonomous agents. The findings have dual-use implications; all experiments are simulation-only and involve no human subjects or personal data.

REPRODUCIBILITY STATEMENT

The paper provides all the relevant details needed to setup the experiments and reproduce the results. These artifacts include: 1) value approximation algorithm and the necessary assumptions to achieve the theoretical guarantees, 2) detailed description of the game setups in the appendix, and, 3) fully executable code with adequate documentation shared via a github repository.

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

APPENDIX

# A CONNECTION BETWEEN PRIMAL AND DUAL GAMES

Here we use the infinitely-repeated game setting to explain the connection between the primal and dual games and the interpretation of the dual variable $\hat{p}$. Please see Theorem 2.2 in De Meyer (1996) and the extension to differential games in Cardaliaguet (2007).

**Game setting.** In an infinitely-repeated normal-form game with one-sided information, there is a set of $I$ possible payoff tables $\{A_i\}_{i=1}^I$. A table is initially drawn from $p_0$ and shown to P1, while P2 only knows $p_0$. At stage $t \in [T]$, the players draw actions from their strategies $\eta_i$ and $\zeta$, respectively, under the current belief $p_t$. The actions are public, which trigger belief updates, yet the actual payoffs are hidden (or otherwise P2 may infer the true payoff easily). At the terminal $T$, an average payoff is received by P1 from P2. This game is proven to have a value (Aumann et al., 1995) for finite and infinite $T$. Here we only consider the latter case where the value is defined as

$$V(p_0) = \max_{\{\eta_i\}_{i=1}^I} \min_{\zeta} \lim_{T \to \infty} \frac{1}{T} \sum_{t=1}^T \mathbb{E}_{i \sim p_0}[\eta_i^\top A_i \zeta]. \tag{9}$$

Notice that this is a special case of the differential games we are interested in, where the incomplete information is on the running loss, the action spaces are discrete, and the dynamics is identity.

**Primal-dual properties.** Let the primal game be $G(p)$ for $p \in \Delta(I)$, the dual game be $G^*(\hat{p})$ for $\hat{p} \in \mathbb{R}^I$, and let $\{\eta_i\}_{i=1}^I$ be the set of strategies for P1 and $\zeta$ the strategy for P2. $\eta_i \in \Delta(d_u)$ and $\zeta \in \Delta(d_v)$. We note that P1's strategy $\{\eta_i\}_{i=1}^I$ can also be together represented in terms of $\pi := \{\pi_{ij}\}^{I,d_u}$ such that $\sum_j^{d_u} \pi_{ij} = p[i]$ and $\eta_i[j] = \pi_{ij}/p[i]$, i.e., nature's distribution is the marginal of $\pi$ and P1's strategy the conditional of $\pi$. Let $G_{\eta\zeta}^i$ be the payoff to P1 of type $i$ for strategy profile $(\eta, \zeta)$. From De Meyer (1996) we have the following results connecting $G(p)$ and $G^*(\hat{p})$:

1. If $\pi$ is Nash for P1 in $G(p)$ and $\hat{p} \in \partial V(p)$, then $\{\eta_i\}_{i=1}^I$ is also Nash for P1 in $G^*(\hat{p})$.

2. If $\pi$ is Nash for P1 in $G^*(\hat{p})$ and $p$ is induced by $\pi$, then $p \in \partial V^*(\hat{p})$ and $\pi$ is Nash for P1 in $G(p)$.

3. If $\zeta$ is Nash for P2 in $G^*(\hat{p})$ and $p \in \partial V^*(\hat{p})$, then $\zeta$ is also Nash for P2 in $G(p)$.

4. If $\zeta$ is Nash for $G(p)$, and let $\hat{p}^i := \max_{\eta \in \Delta(d_u)} G_{\eta\zeta}^i$ and $\hat{p} := [\hat{p}^1, ..., \hat{p}^I]^T$, then $p \in \partial V^*(\hat{p})$ and $\zeta$ is also Nash for P2 in $G^*(\hat{p})$.

From the last two properties we have: If $\zeta$ is Nash for $G(p)$ and $G^*(\hat{p})$, then $\hat{p} = \max_{\eta \in \Delta(d_u)} G_{\eta\zeta}^i$, i.e., $\hat{p}[i]$ is the payoff of type $i$ if P1 plays a best response for that type to P2's Nash.

# B PROOF OF THEOREM 4.1

**Theorem 4.1** (A splitting reformulation of the primal and dual DPs). The RHS of Eq. 3 can be reformulated as

$$\min_{\{u^k\}, \{\alpha_{ki}\}} \max_{\{v^k\}} \sum_{k=1}^I \lambda^k \left( V(t+\tau, x^k, p^k) + \tau \mathbb{E}_{i \sim p^k}[l_i(x, u^k, v^k)] \right)$$

$$\text{s.t.} \quad u^k \in \mathcal{U}, \quad x^k = x + \tau f(x, u^k, v^k), \quad v^k \in \mathcal{V},$$

$$\alpha_{ki} \in [0, 1], \quad \sum_{k=1}^I \alpha_{ki} = 1, \quad \lambda^k = \sum_{i=1}^I \alpha_{ki} p[i], \tag{$P_1$}$$

$$p^k[i] = \frac{\alpha_{ki} p[i]}{\lambda^k}, \quad \forall i, k \in [I].$$

And the RHS of Eq. 5 can be reformulated as

$$\min_{\{v^k\},\{\lambda^k\},\{\hat{p}^k\}} \max_{\{u^k\}} \sum_{k=1}^{I+1} \lambda^k \left( V^*(t+\tau, x^k, \hat{p}^k - \tau l(x, u^k, v^k)) \right)$$

$$\text{s.t.} \quad u^k \in \mathcal{U}, \quad v^k \in \mathcal{V}, \quad x^k = x + \tau f(x, u^k, v^k), \tag{P_2}$$

$$\lambda^k \in [0,1], \sum_{k=1}^{I+1} \lambda^k \hat{p}^k = \hat{p}, \sum_{k=1}^{I+1} \lambda^k = 1, \ k \in [I+1].$$

*Proof.* Recall that the primal DP is:

$$V_\tau(t_0, x_0, p) = \min_{\{\eta_i\}} \mathbb{E}_{i \sim p, \ u \sim \eta^i} \left[ \max_{v \in \mathcal{V}} \{V_\tau(t_0 + \tau, x'(u, v), p'(u)) + \tau l_i(x, u, v)\} \right]$$

$$= \min_{\{\eta_i\}} \int_{\mathcal{U}} \bar{\eta}(u) \max_{v \in \mathcal{V}} \{V_\tau(t_0 + \tau, x'(u, v), p'(u)) + \tau \mathbb{E}_{i \sim p'(u)}[l_i(x, u, v)]\} \, du \tag{10}$$

$$= \min_{\{\eta_i\}} \int_{\mathcal{U}} \bar{\eta}(u) a(u, p'(u)) du,$$

where the last equality uses Isaacs' condition to reduce $v$ as an implicit function of $u$:

$$a(u, p'(u)) = \max_{v \in \mathcal{V}} V_\tau(t_0 + \tau, x'(u, v), p'(u)) + \tau \mathbb{E}_{i \sim p'(u)}[l_i(x, u, v)]. \tag{11}$$

Now we introduce a pushforward measure $\nu$ on $\Delta(I)$ for any $E \subset \Delta(I)$: $\nu(E) = \int_{\{u:p'(u) \in E\}} \bar{\eta}(u) \, du$. Let $\eta_{p'}$ be the conditional measure on $\mathcal{U}$ for each $p'$. Then we have

$$\min_{\{\eta_i\}} \int_{\mathcal{U}} \bar{\eta}(u) a(u, p'(u)) du = \min_\nu \int_{\Delta(I)} \min_{\eta_{p'}} \left[ \int_{p'(u)=p'} a(u, p') \, \eta_{p'}(du) \right] \nu(dp')$$

$$= \min_\nu \int_{\Delta(I)} \min_{u \in \mathcal{U}} a(u, p') \nu(dp')$$

$$= \min_\nu \int_{\Delta(I)} \tilde{a}(p') \nu(dp').$$

This leads to the following reformulation of $V_\tau$:

$$V_\tau(t_0, x_0, p) = \min_\nu \int_{\Delta(I)} \tilde{a}(p') \nu(dp')$$

$$\text{s.t.} \quad \mathbb{E}_\nu[p'] = p. \tag{12}$$

Now we show that the RHS of Eq. 12 computes the convexificiation of $\tilde{a}(p')$ at $p' = p$.

To do so, we first show $V_\tau$ is convex on $\Delta(I)$: Let two probability measures $\nu^1$ and $\nu^2$ be the solutions for $V_\tau(t_0, x_0, p^1)$ and $V_\tau(t_0, x_0, p^2)$, respectively. For any $\theta \in [0, 1]$ and $p^\theta = \theta p^1 + (1-\theta)p^2$, the mixture $\nu^\theta := \theta\nu^1 + (1-\theta)\nu^2$ satisfies $\int_{\Delta(I)} p'\nu(dp') = \theta \int_{\Delta(I)} p'\nu^1(dp') + (1-\theta) \int_{\Delta(I)} p'\nu^2(dp') = p^\theta$ and is a feasible solution to Eq. 12 for $p^\theta$. Therefore

$$V(p^\theta) \leq \int_{\Delta(I)} a(p')\nu^\theta(dp') = \theta V(p^1) + (1-\theta)V(p^2).$$

Then we show $V(p) \leq \tilde{a}(p)$ for any $p \in \Delta(I)$: The Dirac measure $\delta_p$ concentrated at $p$ satisfies $\int_{\Delta(I)} p'\delta_p(dp') = p$. Thus $\delta_p$ is a feasible solution to Eq. 12 and by definition $V(p) \leq \int_{\Delta(I)} \tilde{a}(p')\delta_p(dp') = \tilde{a}(p)$.

Lastly, we show $V$ is the largest convex minorant of $\tilde{a}$: Let $h$ be any convex function on $\Delta(I)$ such that $h(p) \leq \tilde{a}(p)$ for all $p$. Given $p \in \Delta(I)$, for any probability measure $\nu$ that satisfies $\int_{\Delta(I)} p'\nu(dp') = p$, we have

$$h(p) = h(\int_{\Delta(I)} p'\nu(dp')) \leq \int_{\Delta(I)} h(p')\nu(dp') \leq \int_{\Delta(I)} \tilde{a}(p')\nu(dp').$$

Since this inequality holds for arbitrary $\nu$, including the optimal ones that define $V(p)$ through Eq. 12, it follows that $h(p) \leq V(p)$. With these, $V(\cdot)$ in Eq. 12 is the convexification of $\tilde{a}(\cdot)$.

Since convexification in $\Delta(I)$ requires at most $I$ vertices, $\nu^*$ that solves Eq. 12 is $I$-atomic. We will denote by $\{p^k\}_{k \in [I]}$ the set of "splitting" points that have non-zero probability masses according to $\nu^*$, and let $\lambda^k := \nu^*(p^k)$. Using Isaacs' condition, $\arg\min_{u \in \mathcal{U}} a(u, p)$ is non-empty for any $p \in \Delta(I)$, and therefore each $p^k$ is associated with (at least) one action in $\arg\min_{u \in \mathcal{U}} a(u, p^k)$. As a result, $\{\eta_i\}$ is also concentrated on a common set of $I$ actions in $\mathcal{U}$. Specifically, denote this set by $\{u^k\}_{k \in [I]}$, we should have $\alpha_{ki} := \eta_i(u^k) = \lambda^k p^k[i]/p[i]$. Thus we reach $\mathrm{P}_1$. The same proof technique can be applied to the dual DP to derive $\mathrm{P}_2$. $\qquad\square$

## C    PROOF OF THEOREM 4.2

**Theorem 4.2.** For any $(t, x, p) \in [0, T] \times \mathcal{X} \times \Delta(I)$, if A1-5 hold, then for any $\epsilon > 0$, there exist $C > 0$, and $\tau \in (0, \tau^*]$, such that

$$0 \leq \max_{\zeta \in \mathcal{Z}} J(t, x, p; \{\eta_{i,\tau}^\dagger\}, \zeta) - V(t, x, p) \leq \epsilon \tag{13}$$

Similarly, for any $(t, x, \hat{p}) \in [0, T] \times \mathcal{X} \times \mathbb{R}^I$,

$$0 \leq \max_{\{\eta_i\} \in \{\mathcal{H}^i\}^I} J^*(t, x, \hat{p}; \{\eta_i\}, \zeta_\tau^\dagger) - V^*(t, x, \hat{p}) \leq \epsilon \tag{14}$$

### C.1    SETTINGS

We recall all necessary settings for the proof.

From Thm. 4.1, the Bellman backup for $G_\tau$ is $\mathrm{P}_1$, which can be written as an operator $T_\tau$:

$$T_\tau[V_\tau](t + \tau, x, p) := V_\tau(t, x, p) = \min_{\{\lambda^k\}, \{p^k\}} \sum_{k=1}^I \lambda^k \tilde{V}_\tau(t, x, p^k), \tag{15}$$

where $\tilde{V}_\tau(t, x, p^k) = \min_{u \in \mathcal{U}} \max_{v \in \mathcal{V}} V_\tau(t + \tau, x + \tau f(x, u, v), p^k) + \tau \mathbb{E}_{i \sim p^k}[l_i(x, u, v)]$ is the non-revealing value at $(t, x, p^k)$, and $\sum_{k=1}^I \lambda^k = 1$, $\sum_{k=1}^I \lambda^k p^k = p$, $p^k \in \Delta(I)$, $\forall k \in [I]$.

From Cardaliaguet (2007), the value $V$ of the original game $G$ is the unique viscosity solution to the following Hamilton-Jacobi equation

$$\nabla_t w + H(t, x, \nabla_x w) = 0, \tag{16}$$

with terminal boundary $w(T, x, p) = \sum_{i=1}^I p[i] g_i(x)$, and is convex in $p$.

### C.2    PRELIMINARY LEMMAS

The following lemmas will be used in the main proof.

**Lemma C.1** (Uniform Convergence (Thm 3.1 of Cardaliaguet (2009))**.** *Assuming (A1-A5) hold, the map $V_\tau$ converges uniformly to the value function $V$ on compact subsects of $[0, T] \times \mathcal{X} \times \Delta(I)$, in the following sense:*

$$\lim_{\tau \to 0^+,\ t_k \to t,\ x' \to x,\ p' \to p} V_\tau(t_k, x', p') = V(t, x, p) \quad \forall(t, x, p) \in [0, T] \times \mathcal{X} \times \Delta(I).$$

*Proof.* Proved in Cardaliaguet (2009) for both $V_\tau$ and $V_\tau^*$. $\qquad\square$

**Lemma C.2.** *Let $\bar{\mathcal{V}}(t_0) = \{v : [t_0, T] \to \mathcal{V}\}$ denote a set of open-loop controls for P2. Then, given $f$, a constant control $u \in \mathcal{U}$ on $[t_k, t_{k+1}]$, a time-varying control $v \in \mathcal{V}(t)$, there exists a $w$ such that*

$$w \in Cof(x, u, \mathcal{V}) \tag{17}$$

*where Co denotes the convex hull.*

*Proof.* Let

$$w := \frac{1}{\tau} \int_{t_k}^{t_{k+1}} f(x, u, v(s)) \, ds \tag{18}$$

We show that $w$ as defined above lies in the convex hull of the set $f(x, u, \mathcal{V})$.

Let $S := f(x, u, \mathcal{V}) = \{f(x, u, \bar{v}) : \bar{v} \in \mathcal{V}\}$. For any time $s$, since $v(s) \in \mathcal{V}$, we have:

$$f(x, u, v(s)) \in S \quad \text{for all } s \in [t_k, t_{k+1}] \tag{19}$$

**Case 1: when $v(\cdot)$ is piecewise-constant.** Assume first that $v(\cdot)$ is piecewise constant on the partition $[t_k, t_{k+1}]$:

$$[t_k, t_{k+1}] = \bigcup_{j=1}^{m} N_j, \quad |N_j| = \Delta t_j, \quad \sum_{j=1}^{m} \Delta t_j = \tau,$$

and $v(s) = v_j$ on $N_j$. Then

$$\int_{t_k}^{t_{k+1}} f(x, u, v(s)) ds = \sum_{j=1}^{m} \Delta t_j f(x, u, v_j).$$

Divide by $\tau$ to get

$$w = \frac{1}{\tau} \sum_{j=1}^{m} \Delta t_j f(x, u, v_j) = \sum_{j=1}^{m} q_j f(x, u, v_j), \quad \text{where } q_j = \frac{\Delta t_j}{\tau}.$$

$q_j \geq 0$ and $\sum_{j=1}^{m} q_j = 1$, so $w$ is a finite weighted average of points in $S$. Hence,

$$w \in \mathrm{Co}(S) = \mathrm{Co} f(x, u, \mathcal{V}) \tag{20}$$

**Case 2: general measurable case.** For a general measurable $v(\cdot)$, we can approximate it by piecewise-constant $v_n(\cdot)$ such that

$$f(x, u, v_n(\cdot)) \to f(x, u, v(\cdot)) \quad \text{in } L^1([t_k, t_{k+1}]$$

This is true because $f(x, u, \cdot)$ is bounded and measurable through $v(\cdot)$.

Define

$$w_n := \frac{1}{\tau} \int_{t_k}^{t_{k+1}} f(x, u, v_n(s) \, ds.$$

From Case 1, each $w_n \in \mathrm{Co}(S)$.

Moreover,

$$|w_n - w| = \left| \frac{1}{\tau} \int_{t_k}^{t_{k+1}} (f(x, u, v_n(s)) - f(x, u, v(s)) \, ds \right|$$

$$\leq \frac{1}{\tau} \int_{t_k}^{t_{k+1}} |f(x, u, v_n(s)) - f(x, u, v(s))| \, ds \to 0$$

So $w_n \to w$

Finally, $S$ is compact because $\mathcal{V}$ is compact and $f$ is continuous. Hence, $\mathrm{Co}(S)$ is compact and therefore closed. Thus, the limit of $w_n \in \mathrm{Co}(S)$ also belongs to $\mathrm{Co}(S)$:

$$w \in \mathrm{Co}(S) = \mathrm{Co} f(x, u, \mathcal{V}) \tag{21}$$

$\square$

**Lemma C.3** (Local Quadratic Error). *For any $k \in \{0, \ldots, L-1\}$, any $x$, a fixed control $u \in \mathcal{U}$, any control $v \in \bar{\mathcal{V}}(t_k)$, there exists some $w \in \mathrm{Co} f(x, u, \mathcal{V})$, such that*

$$|X_{t_{k+1}}^{t_k, x, u, v} - (x + \tau w)| \leq C_1 \tau^2 \tag{22}$$

*Proof.* We have, from the ODE,

$$X(t_{k+1}) = X_{t_{k+1}}^{t_k, x, u, v} = x + \int_{t_k}^{t_{k+1}} f(X(s), u, v(s)) \, ds. \tag{23}$$

From the definition (18) of $w$,

$$x + \tau w = x + \int_{t_k}^{t_{k+1}} f(x, u, v(s)) ds. \tag{24}$$

Subtract (24) from (23)

$$X(t_{k+1}) - (x + \tau w) = \int_{t_k}^{t_{k+1}} \Big( f(X(s), u, v(s)) - f(x, u, v(s)) \Big) \, ds$$

$$\Big| X(t_{k+1}) - (x + \tau w) \Big| = \Big| \int_{t_k}^{t_{k+1}} \Big( f(X(s), u, v(s)) - f(x, u, v(s)) \Big) \, ds \Big|$$

$$\leq \int_{t_k}^{t_{k+1}} \Big| f(X(s), u, v(s)) - f(x, u, v(s)) \Big| \, ds$$

From Lipschitz continuity of $f$, $|f(X(s), u, v(s) - f(x, u, v(s))| \leq L_f |X(s) - x|$,

$$\Big| X(t_{k+1}) - (x + \tau w) \Big| \leq \int_{t_k}^{t_{k+1}} L_f |X(s) - x| \, ds.$$

From boundedness of $f$: Let $\|f\|_\infty \leq F$. Then for $s \in [t_k, t_{k+1}]$

$$\Big| X(s) - x \Big| = \Big| \int_{t_k}^{s} f(X(r), u, v(r) \, dr \Big| \leq \int_{t_k}^{s} |f(\cdot)| \, dr \leq \int_{t_k}^{s} F dr = F(s - t_k),$$

which leads to

$$\Big| X(t_{k+1}) - (x + \tau w) \Big| \leq \int_{t_k}^{t_{k+1}} L_f \cdot F \cdot (s - t_k) \, ds = L_f \cdot F \int_0^\tau r \, dr = \frac{L_f \cdot F}{2} \tau^2$$

which yields equation (22) with $C_1 = \frac{L_f \cdot F}{2}$. $\qquad\qquad\square$

**Lemma C.4** (Grid Error). *There exists some constant $C > 0$ independent of $\tau$ such that, for any $k \in \{0, \cdots, L\}$, $x \in \mathbb{R}^{d_x}$, and $p \in \Delta(I)$, we have*

$$\max_{\zeta \in \mathcal{Z}} J(t_k, x, p; \{\eta_{i,\tau}^\dagger\}, \zeta) \leq V_\tau(t_k, x, p) + C(T - t_k)\tau \tag{25}$$

*Proof.* Let $\tau = T/L$, $\{\eta_{i,\tau}^\dagger\}$ be the random strategy associated with the feedback strategy $(u^k, \alpha_{ki})$ in the game $G_\tau$, and $\zeta \in \mathcal{Z}$. Then, for any $k \in \{0, \ldots, L\}$, $x \in \mathbb{R}^{d_x}$, and $p \in \Delta(I)$, claim that:

$$\max_{\zeta \in \mathcal{Z}} J(t_k, x, p; \{\eta_{i,\tau}^\dagger\}, \zeta) \leq V_\tau(t_k, x, p) + C(L - k)\tau^2 \tag{26}$$

We use backward induction to prove the equation (26). At $k = L$, the inequality holds trivially as there is no time left in the game and the error term goes to 0. Then, assume that (26) holds for $k + 1$:

$$\max_{\zeta \in \mathcal{Z}(t_{k+1})} J(t_{k+1}, x, p; \{\eta_{i,\tau}^\dagger\}, \zeta) \leq V_\tau(t_{k+1}, x, p) + C(L - (k+1))\tau^2 \tag{27}$$

Now, to prove for any $k$, let us analyze what happens on $[t_k, t_{k+1})$. From the feedback given by $(P_1)$, for a fixed $(x, p)$ we have:

$$X_T^{t_k, x, \{\eta_{i,\tau}^\dagger\}, \zeta} = X_T^{t_{k+1}, x^j, \{\eta_{i,\tau}^{j\dagger}\}, \zeta^j} \quad \text{with probability } \lambda^j p^j[i]/p[i],$$

where

- $x^j = X_{t_{k+1}}^{t_k, x, u^j, \zeta(u^j)}$,

- $\{\eta_{i,\tau}^{j\dagger}\}$ is the continuation of the random strategy from time $t_{k+1}$ associated with the same feedback but starting at $(t_{k+1}, x^j, p^j)$,

- $\zeta^j \in \mathcal{Z}(t_{k+1})$ is the continuation of the non-anticipative $\zeta$ after $u^j$ is chosen.

Then, using Lemma C.3, there is some $w^j \in \mathrm{Co} f(x, u^j, \mathcal{V})$ such that:

$$|x^j - (x + \tau w^j)| \leq C_1 \tau^2. \tag{28}$$

From the definition of $J$,

$$J(t_k, x, \{\eta_{i,\tau}^\dagger\}, \zeta, p)$$

$$= \sum_{i=1}^{I} p[i] \mathbb{E}_{\{\eta_{i,\tau}^\dagger\}} \left[ g_i\left( X_T^{t_k, x, \{\eta_{i,\tau}^\dagger\}, \zeta} \right) + \int_{t_k}^{T} l_i\left( X(s), \{\eta_{i,\tau}^\dagger\}(s), \zeta(s) \right) ds \right]$$

$$= \sum_{i=1}^{I} \sum_{j=1}^{I+1} p[i] \lambda^j p^j[i] / p[i] \mathbb{E}_{\{\eta_{i,\tau}^{j\dagger}\}} \left[ g_i\left( X_T^{t_{k+1}, x, \{\eta_{i,\tau}^{j\dagger}\}, \zeta^j} \right) + \int_{t_k}^{t_{k+1}} l_i(x(s), u^j, \zeta(u^j)) \, ds \right.$$

$$\left. + \int_{t_{k+1}}^{T} l_i\left( X(s), \{\eta_{i,\tau}^{j\dagger}\}(s), \zeta^j(s) \right) ds \right]$$

$$\leq \sum_{j=1}^{I+1} \lambda^j \max_{\zeta' \in \mathcal{Z}(t_{k+1})} \sum_{i=1}^{I} p^j[i] \mathbb{E}_{\{\eta_{i,\tau}^{j\dagger}\}} \left[ g_i\left( X_T^{t_{k+1}, x, \{\eta_{i,\tau}^{j\dagger}\}, \zeta'} \right) + \int_{t_k}^{t_{k+1}} l_i(x(s), u^j, \zeta(u^j)) \, ds \right.$$

$$\left. + \int_{t_{k+1}}^{T} l_i\left( X(s), \{\eta_{i,\tau}^{j\dagger}\}(s), \zeta'(s) \right) ds \right]$$

Combining the above result with (27), we get:

$$J(t_k, x, \{\eta_{i,\tau}^\dagger\}, \zeta, p) \leq \sum_{j=1}^{I+1} \lambda^j \left[ V_\tau(t_{k+1}, x^j, p^j) + C(L - (k+1))\tau^2 \right].$$

Finally, applying (28) and assuming $C_2$-Lipschitz continuity of $V_\tau$, we arrive at:

$$J(t_k, x, \{\eta_{i,\tau}^\dagger\}, \zeta, p) \leq \sum_{j=1}^{I+1} \lambda^j \left[ V_\tau(t_{k+1}, x + \tau w^j, p^j) + C_2|x^j - (x + \tau w^j)| + \right.$$

$$\left. C(L - (k+1))\tau^2 \right]$$

$$\leq \sum_{j=1}^{I+1} \lambda^j \left[ V_\tau(t_{k+1}, x + \tau w^j, p^j) + C_1 C_2 \tau^2 + C(L - (k+1))\tau^2 \right]$$

$$\leq \sum_{j=1}^{I+1} \lambda^j \max_{w \in \mathrm{Co} f(x, u^j, \mathcal{V})} \left[ V_\tau(t_{k+1}, x + \tau w, p^j) + C_1 C_2 \tau^2 + C(L - (k+1))\tau^2 \right]$$

$$\leq V_\tau(t_k, x, p) + (C_1 C_2 + 2)\tau^2 + C(L - (k+1))\tau^2.$$

The term $2\tau^2$ arises from two approximation steps: first $\tau^2$ from selecting an approximately optimal convex decomposition in the definition of $V_\tau(t_k, x, p)$, and the second $\tau^2$ from choosing $u^j$ to be only $\tau^2$-optimal for $\min_u \max_v(\cdot)$ so that they are Borel measurable with respect to $(x, p)$.

Then, choosing $C = (C_1 C_2 + 2)$ satisfies (26).

Since we have, $L = T/\tau$, and $t_k = k\tau$. We get

$$\max_{\zeta \in \mathcal{Z}} J(t_k, x, p; \{\eta_{i,\tau}^\dagger\}, \zeta) \leq V_\tau(t_k, x, p) + C(T - t_k)\tau$$

Note that compared with the proof provided in Cardaliaguet (2009) Theorem 4.1, our proof incorporates instantaneous costs and clarified the stage-wise $\mathcal{O}(\tau)$ error in (27).

$\square$

### C.3 Proof of Theorem 4.2

*Proof.* Our proof focuses on the primal game. The same technique can be applied to the dual game. First, it is easy to see $V(t, x, p) \leq \max_{\zeta \in \mathcal{Z}} J(t, x, p; \{\eta_{i,\tau}^{\dagger}\}, \zeta)$ because $\{\eta_{i,\tau}^{\dagger}\}$ is not necessarily NE in $G$.

Next, we need to show

$$0 \leq \max_{\zeta \in \mathcal{Z}} J(t, x, p; \{\eta_{i,\tau}\}^{\dagger}, \zeta) - V(t, x, p) \leq \epsilon \tag{29}$$

i.e., applying $\{\eta_{i,\tau}^{\dagger}\}$ (which solves $G_\tau$) to $G$ will yield a value not far from the true value of $G$.

Let us fix some $t'$, $x'$, and $p'$, and some $k$ such that $t' \in [t_{k-1}, t_k)$. Then,

$$\max_{\zeta \in \mathcal{Z}_r(t')} J(t', x', p'; \{\eta_{i,\tau}^{\dagger}\}, \zeta) = \max_{\zeta \in \mathcal{Z}(t')} J(t', x', p'; \{\eta_{i,\tau}^{\dagger}\}, \zeta)$$

$$\leq \max_{|x-x'| \leq F\tau} \max_{\zeta \in \mathcal{Z}(t_k)} J(t_k, x, p'; \{\eta_{i,\tau}^{x\dagger}\}, \zeta) + \tau \|l\|_{\infty}$$

where $\{\eta_{i,\tau}^{x\dagger}\}$ is the strategy at $(t_k, x, p')$, $F$ is a bound on $f$, and $\|l\|_{\infty}$ term is due to bounded instantaneous cost. Hence, from Lemma C.4

$$\max_{\zeta \in (t')Z} J(t', x', p', \{\eta_{i,\tau}\}^{\dagger}, \zeta) \leq \max_{|x-x'| \leq F\tau} V_\tau(t_k, x, p') + \mathcal{O}(\tau)$$

Here, $\mathcal{O}(\tau)$ absorbs $C(T - t_k)\tau + \tau \|l\|_{\infty}$. Together with $\tau = T/L \to 0$ and Lemma C.1, $\max_{|x-x'| \leq F\tau} V_\tau(t_k, x, p')$ converges to $V(t', x', p')$ and $\mathcal{O}(\tau) \to 0$

Therefore, for any $\epsilon > 0$ there exists $\tau^* > 0$ such that for all $\tau \in (0, \tau^*]$ inequality (29) holds. $\square$

## D Prediction Error of Value Approximation

Here we show that CAMS shares the same exponential error propagation as in standard approximate value iteration (AVI). The only difference from AVI is that the measurement error in CAMS comes from numerical approximation of the minimax problems rather than randomness in state transition and rewards. To start, let the true value be $V(t, x, p)$. Following Zanette et al. (2019), the prediction error $\epsilon_t^{bias} := \max_{x,p} |\hat{V}_t(x, p) - V(t, x, p)|$ is affected by (1) the prediction error $\epsilon_{t+\tau}^{bias}$ propagated back from $t + \tau$, (2) the minimax error $\epsilon_t^{minmax}$ caused by limited iterations in solving the minimax problem at each collocation point: $\epsilon_t^{minmax} = \max_{(x,p) \in \mathcal{S}_t} |\tilde{V}(t, x, p) - V(t, x, p)|$, and (3) the approximation error due to the fact that $V(t, \cdot, \cdot)$ may not lie in the model hypothesis space $\mathcal{V}_t$ of $\hat{V}_t$: $\epsilon_t^{app} = \max_{x,p} \min_{\hat{V}_t \in \mathcal{V}_t} |\hat{V}_t(x, p) - V(t, x, p)|$.

**Approximation error.** For simplicity, we will abuse the notation by using $x$ in place of $(x, p)$ and omit time dependence of variables when possible. In practice we consider $\hat{V}_t$ as neural networks that share the architecture and the hypothesis space. Note that $\hat{V}_T(\cdot) = V(T, \cdot)$ is analytically defined by the boundary condition and thus $\epsilon_T^{app} = \epsilon_T^{bias} = 0$. To enable the analysis on neural networks, we adopt the assumption that $\hat{V}$ is infinitely wide and that the resultant neural tangent kernel (NTK) is positive definite. Therefore from NTK analysis (Jacot et al., 2018), $\hat{V}$ can be considered a kernel machine equipped with a kernel function $r(x^{(i)}, x^{(j)}) := \langle \phi(x^{(i)}), \phi(x^{(j)}) \rangle$ defined by a feature map $\phi : \mathcal{X} \to \mathbb{R}^{d_\phi}$. Given training data $\mathcal{S} = \{(x^{(i)}, V^{(i)})\}$, let $r(x)[i] := r(x^{(i)}, x)$, $R_{ij} := r(x^{(i)}, x^{(j)})$, $V_{\mathcal{S}} := [V^{(1)}, ..., V^{(N)}]^\top$, $\Phi_{\mathcal{S}} := [\phi(x^{(1)}), ..., \phi(x^{(N)})]$, and $w_{\mathcal{S}} := \Phi_{\mathcal{S}}(\Phi_{\mathcal{S}}^\top \Phi_{\mathcal{S}})^{-1} V_{\mathcal{S}}$ be model parameters learned from $\mathcal{S}$, then

$$\hat{V}(x) = r(x)^\top R^{-1} V_{\mathcal{S}} = \langle \phi(x), w_{\mathcal{S}} \rangle \tag{30}$$

is a linear model in the feature space. Let $\theta^{\phi(x)} := r(x)^\top R^{-1}$ and $C := \max_x \|\theta^{\phi(x)}\|_1$. Further, let $\mathcal{S}^\dagger := \arg\min_\mathcal{S} |\langle \phi(x), w_\mathcal{S}\rangle - V(x)|$ and $w^\dagger := w_{\mathcal{S}^\dagger}$, i.e., $w^\dagger$ represents the best hypothetical model given sample size $N$. Since $N$ is finite, the data-dependent hypothesis space induces an approximation error $\epsilon_t^{app} := \max_x |\langle \phi(x), w^\dagger\rangle - V(x)|$. From standard RKHS analysis, we have $\epsilon_t^{app} \propto N^{-\frac{1}{2}}$.

**Error propagation.** Recall that we approximately solve P$_1$ at each collocation point. Let $z := \{\lambda, p, u, v\}$ be the collection of variables and $\tilde{z}$ be the approximated saddle point resulting from DS-GDA. Let $\tilde{V}(t, x, \tilde{z})$ be the approximate value at $(t, x)$ and let $V(t, x, z^*)$ be the value at the true saddle point $z^*$. Lemma D.1 bounds the error of $\tilde{V}(t, x, \tilde{z})$:

**Lemma D.1.** $\max_x |\tilde{V}(t, x, \tilde{z}) - V(t, x, z^*)| \leq \epsilon_{t+\tau}^{bias} + \epsilon_t^{minmax}$.

*Proof.* Note that $\sum_{k=1}^I \lambda^k = 1$. Then

$$\max_x |\tilde{V}(t, x, \tilde{z}) - V(t, x, z^*)| \leq \max_x |\tilde{V}(t, x, \tilde{z}) - \tilde{V}(t, x, z^*)| + \max_x |\tilde{V}(t, x, z^*) - V(t, x, z^*)|$$

$$\leq \epsilon_t^{minmax} + \max_x |\sum_{k=1}^I \lambda^k (\tilde{V}(t+\tau, x', p^k) - V(t+\tau, x', p^k))|$$

$$\leq \epsilon_t^{minmax} + \epsilon_{t+\tau}^{bias}. \tag{31}$$

$\square$

Now we can combine this measurement error with the inherent approximation error $\epsilon_t^{app}$ to reach the following bound on the prediction error $\epsilon_t^{bias}$:

**Lemma D.2.** $\max_x |\hat{V}_t(x) - V(t, x)| \leq C_t(\epsilon_t^{minmax} + \epsilon_{t+\tau}^{bias} + \epsilon_t^{app}) + \epsilon_t^{app}$.

*Proof.*

$$\max_x |\hat{V}_t(x) - V(t, x)| \leq \max_x |\hat{V}_t(x) - \langle \phi(x), w^\dagger\rangle| + \max_x |\langle \phi(x), w^\dagger\rangle - V(t, x)|$$

$$\leq \max_x |\langle \theta^{\phi(x)}, \tilde{V}(t, x) - V(t, x)\rangle| + \max_x |\langle \theta^{\phi(x)}, V(t, x) - \phi(x)^\top w^\dagger\rangle| + \epsilon_t^{app} \tag{32}$$

$$\leq C(\epsilon_t^{minmax} + \epsilon_{t+\tau}^{bias} + \epsilon_t^{app}) + \epsilon_t^{app}.$$

$\square$

Lem. D.3 characterizes the propagation of error:

**Lemma D.3.** *Let* $\epsilon_t^{app} \leq \epsilon^{app}$, $\epsilon_t^{minmax} \leq \epsilon^{minmax}$, *and* $C_t \leq C$ *for all* $t \in [T]$. *If* $\epsilon_T^{app} = 0$, *then* $\epsilon_0^{bias} \leq TC^T(\epsilon^{app} + C(\epsilon^{minmax} + \epsilon^{app}))$.

*Proof.* Using Lem. D.2 and by induction, we have

$$\epsilon_0^{bias} \leq (\epsilon^{app} + C(\epsilon^{minmax} + \epsilon^{app}))\frac{1 - C^T}{1 - C} \leq TC^T(\epsilon^{app} + C(\epsilon^{minmax} + \epsilon^{app})). \tag{33}$$

$\square$

We can now characterize the computational complexity of the baseline algorithm through Thm. D.4, by taking into account the number of DS-GDA iterations and the per-iteration complexity:

**Theorem D.4.** *For a fixed* $T$ *and some error threshold* $\delta > 0$, *with a computational complexity of* $\mathcal{O}(T^3 C^{2T} I^2 \epsilon^{-4} \delta^{-2})$, *CAMS achieves*

$$\max_{(x,p)\in\mathcal{X}\times\Delta(I)} |\hat{V}_0(x, p) - V(0, x, p)| \leq \delta. \tag{34}$$

*Proof.* From Lem. D.3 and using the fact that $\epsilon^{app} \propto N^{-1/2}$, achieving a prediction error of $\delta$ at $t = 0$ requires $N = \mathcal{O}(C^{2T} T^2 \delta^{-2})$. CAMS solves $TN$ minimax problems, each requires a worst-case $\mathcal{O}(\epsilon^{-4})$ iterations, and each iteration requires computing gradients of dimension $\mathcal{O}(I^2)$, considering the dimensionalities of action spaces as constants. This leads to a total complexity of $\mathcal{O}(T^3 C^{2T} I^2 \epsilon^{-4} \delta^{-2})$. $\square$

# E    MULTIGRID ALGORITHMS AND RESULTS

Fig. 7 shows a typical 2-level multigrid (Full-Approximation Scheme or FAS) approach. As discussed, FAS has four steps, namely: (1) restriction of the fine-grid approximation and its residual into the coarse grid (red arrows in Fig. 7); (2) computation of the coarse-grid solution by incorporating restricted fine-grid residuals; (3) computation of the coarse-grid correction; and finally, (4) prolongation of the coarse-grid correction to the fine-grid (shown by the blue arrows in Fig. 7). This can be further extended to $n$-level multigrid by recursively reducing the coarse-grid size until the desired coarsest grid is reached. Alg. 1 presents the $n$-level multigrid algorithm.

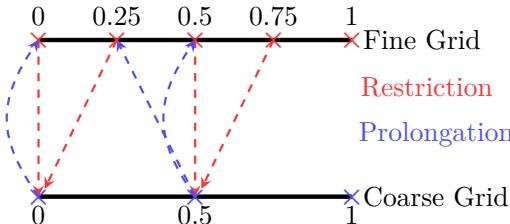

Figure 7: Illustration of 2-level multigrid method.

---

**Algorithm 1:** $n$-Level Multigrid for Value Approximation

---

**Input:** $k_{\max}, k_{\min}, \mathbb{O}$ (minimax solver), $T$ (time horizon), $N$ (number of data points), $\mathcal{R}$ (restriction operator), $\mathcal{P}$ (prolongation operator)

**Initialize:** $\mathcal{T}^l \leftarrow [0, l, 2l, \ldots, T - l], \ \forall\, l \in \{2^{-k_{\max}}, \ldots, 2^{-k_{\min}}\}, L \leftarrow 2^{-k_{\min}}$

**Initialize:** Value networks $\hat{V}_t^l, \ \forall\, t \in \mathcal{T}^l, \ \forall\, l \in \{2^{-k_{\max}}, \ldots, 2^{-k_{\min}}\}$, policy set $\Pi \leftarrow \varnothing$

;

**while** *resources not exhausted or until convergence* **do**

    $R \leftarrow \varnothing, \ E^L \leftarrow \varnothing, \ \mathcal{S} \leftarrow \varnothing$;

    Initialize coarsest-grid correction networks $\varepsilon_t^L, \forall\, t \in \mathcal{T}^L$;

    $\mathcal{S}[t] \leftarrow$ sample $N$ $(t, x, p), \ \forall\, t \in \mathcal{T}^{k_{\max}}$;

    `// down-cycle`

    **for** $k \leftarrow k_{\max}$ ***down to*** $k_{\min} + 1$ **do**

        Compute target via $\mathbb{O}^k$ (init. with $\pi_t$ if $\Pi[k] \neq \varnothing$), and store updated policies $\pi_t$ in $\Pi[k], \ \forall\, t \in \mathcal{T}^k$;

        Compute residuals $r^k[t], \ \forall\, t \in \mathcal{T}^k$;

        **if** $k \neq k_{\max}$ **then**

            $r_t^k \leftarrow r_t^k + \mathcal{R} r_t^{k+1}, \ \forall\, t \in \mathcal{T}^k$;

        Store $r_t^k$ in $R[k]$;

    **for** $t \leftarrow T - L$ **to** $0$ **do**

        `// coarse-solve backwards in time`

        $e_t^L \leftarrow \mathbb{O}^L(\mathcal{R}\hat{V}_{t+L}^l + \varepsilon_{t+L}^L) - \mathbb{O}^L(\mathcal{R}\hat{V}_{t+L}^l) - \mathcal{R}\, r_t^{k_{\min}+1}$ ;        `// ` $e_T^L = 0, \ \varepsilon_T^L = \varnothing$

        Store $e_t^L$ in $E^L$;

        Fit $\varepsilon_L^t$ to $e_L^t$;

    `// up-cycle`

    **for** $k \leftarrow k_{\min} + 1$ ***to*** $k_{\max}$ **do**

        $e_t^k \leftarrow \mathcal{P}(e_t^{k-1}), \ \forall\, t \in \mathcal{T}^k$;

        Update $\hat{V}_t^k \leftarrow \hat{V}_t^k + e_t^k$;

    `// post smoothing (for all t's and l's)`

    `target`, $\pi_t \leftarrow \mathbb{O}^l(\hat{V}_{t+l}^l)$ (initialized with $\pi_t$)

    Fit $\hat{V}_t^l$ to `target` and replace $\pi_t$ in $\Pi[l]$;

---

In Fig. 8 we compare learned trajectories via the multigrid approach against the ground truth. The learned trajectories closely resemble the ground truth as P1 successfully concealing its payoff type until a critical time. In Fig. 8 we visualize the NE trajectories of P2 by solving the dual game.

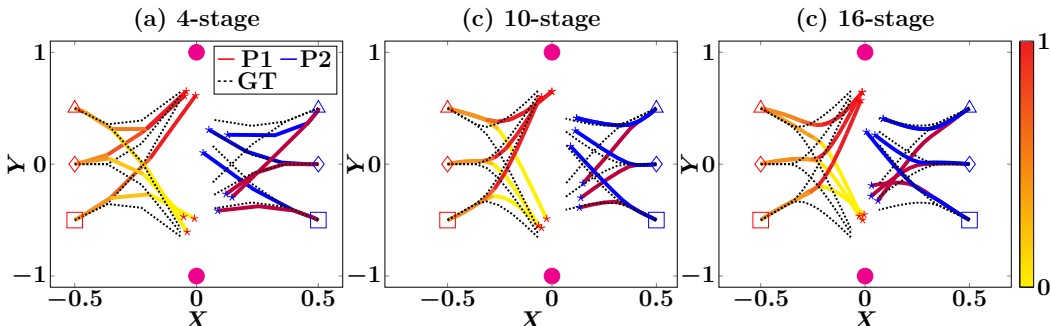

Figure 8: Comparison of trajectories generated using value learned via multigrid method vs the ground truth.

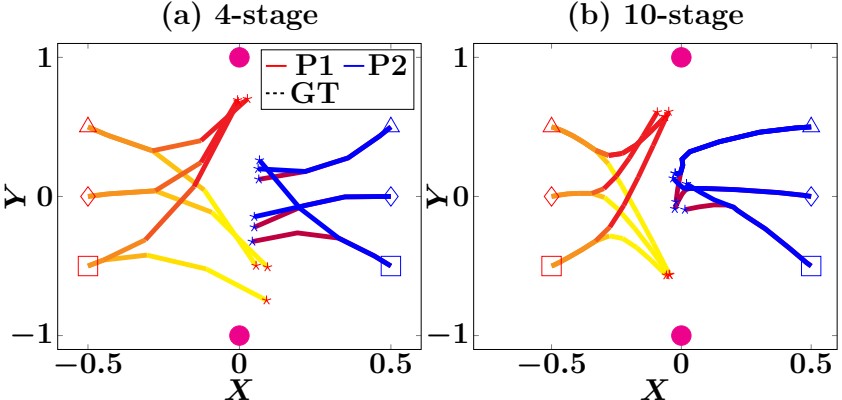

Figure 9: Trajectories when P1 and P2 play their respective primal and dual game. P1's actions are a result of the primal value function whereas P2's actions are a result of the dual value function. Both primal and dual values are learned using multigrid approach.

## F ANALYTICAL EXAMPLES

In this section, we walk through the derivation of analytical NEs for two problems: Hexner's game and a zero-sum variant of the classic beer-quiche game. The former is differential where players take actions simultaneously; the latter is dynamic and turn-based. Both have one-sided payoff information and finite time horizons. These examples are reproduced from Ghimire et al. (2024) with permission.

### F.1 HEXNER'S GAME: ANALYTICAL SOLUTION

Here we discuss the solution to Hexner's game using primal and dual formulations (i.e., $P_1$ and $P_2$) on a differential game as proposed in Hexner (1979). Consider two players with linear dynamics

$$\dot{x}_i = A_i x_i + B_i u_i,$$

for $i = 1, 2$, where $x_i(t) \in \mathbb{R}^{d_x}$ are system states, $u_i(t) \in \mathcal{U}$ are control inputs belonging to the admissible set $\mathcal{U}$, $A_i, B_i \in \mathbb{R}^{d_x \times d_x}$. Let $\theta \in \{-1, 1\}$ be Player 1's type unknown to Player 2. Let $p_\theta$ be Nature's probability distribution of $\theta$. Consider that the game is to be played infinite many times,

the payoff is an expectation over $\theta$:

$$J(u_1, u_2) = \mathbb{E}_\theta \Big[ \int_0^T \Big( \|u_1\|_{R_1}^2 - \|u_2\|_{R_2}^2 \Big) dt +$$
$$\|x_1(T) - z\theta\|_{K_1(T)}^2 - \|x_2(T) - z\theta\|_{K_2(T)}^2 \Big], \tag{35}$$

where, $z \in \mathbb{R}^{d_x}$. $R_1$ and $R_2$ are continuous, positive-definite matrix-valued functions, and $K_1(T)$ and $K_2(T)$ are positive semi-definite matrices. All parameters are publicly known except for $\theta$, which remains private. Player 1's objective is to get closer to the target $z\theta$ than Player 2. However, since Player 2 can deduce $\theta$ indirectly through Player 1's control actions, Player 1 may initially employ a non-revealing strategy. This involves acting as though it only knows about the prior distribution $p_\theta$ (rather than the true $\theta$) to hide the information, before eventually revealing $\theta$.

First, it can be shown that players' control has a 1D representation, denoted by $\tilde{\theta}_i \in \mathbb{R}$, through:

$$u_i = -R_i^{-1} B_i^T K_i x_i + R_i^{-1} B_i^T K_i \Phi_i z \tilde{\theta}_i,$$

for $i = 1, 2$, where $\dot{\Phi}_i = A_i \Phi_i$ with boundary condition $\Phi_i(T) = I$, and

$$\dot{K}_i = -A_i^T K_i - K_i A_i + K_i^T B_i R_i^{-1} B_i^T K_i.$$

Then define a quantity $d_i$ as:

$$d_i = z^T \Phi_i^T K_i B_i R_i^{-1} B_i^T K_i^T \Phi_i z. \tag{36}$$

With these, the game can be reformulated with the following payoff function:

$$J(t, \tilde{\theta}_1, \tilde{\theta}_2) = \mathbb{E}_\theta \left[ \int_{\tau=t}^T (\tilde{\theta}_1(\tau) - \theta)^2 d_1(\tau) - (\tilde{\theta}_2(\tau) - \theta)^2 d_2(\tau) d\tau \right], \tag{37}$$

where $d_1$, $d_2$, $p_\theta$ are common knowledge; $\theta$ is only known to Player 1; the scalar $\tilde{\theta}_1$ (resp. $\tilde{\theta}_2$) is Player 1's (resp. Player 2's) strategy. We consider two player types $\theta \in \{-1, 1\}$ and therefore $p_\theta \in \Delta(2)$.

Then by defining critical time:

$$t_r = \arg\min_t \int_0^t (d_1(s) - d_2(s)) ds,$$

we have the following equilibrium:

$$\tilde{\theta}_1(s) = \tilde{\theta}_2(s) = 0 \quad \forall s \in [0, t_r] \tag{38}$$
$$\tilde{\theta}_1(s) = \tilde{\theta}_2(s) = \theta \quad \forall s \in (t_r, T], \tag{39}$$

To solve the game via primal-dual formulation, we introduce a few quantities. First, introduce time stamps $[T_k]_{k=1}^{2r}$ as roots of the time-dependent function $d_1 - d_2$, with $T_0 = 0$, $T_{2q+1} = t_r$, and $T_{2r+1} = T$. Without loss of generality, assume:

$$d_1 - d_2 < 0 \quad \forall t \in (T_{2k}, T_{2k+1}) \, \forall k = 0, ..., r, \tag{40}$$
$$d_1 - d_2 \geq 0 \quad \forall t \in [T_{2k-1}, T_{2k}] \, \forall k = 1, ..., r. \tag{41}$$

Also introduce $D_k := \int_{T_k}^{T_{k+1}} (d_1 - d_2) ds$ and

$$\tilde{D}_k = \begin{cases} \tilde{D}_{k+1} + D_k, & \text{if } \tilde{D}_{k+1} + D_k < 0 \\ 0, & \text{otherwise} \end{cases}, \tag{42}$$

with $\tilde{D}_{2r+1} = 0$.

The following properties are necessary (see Ghimire et al. (2024) for details):

1. $\int_k^{2q+1} (d_1 - d_2) ds = \sum_k^{2q} D_k < 0, \forall k = 0, ..., 2q$;
2. $\int_{2q+1}^k (d_1 - d_2) ds = \sum_{2q+1}^{k-1} D_k > 0, \forall k = 2q+2, ..., 2r+1$;
3. $\tilde{D}_{2q+2} + D_{2q+1} > 0$;
4. $\tilde{D}_k < 0, \forall k < 2q + 1$.

**Primal game.** We start with $V(T, p) = 0$ where $p := p_\theta[1] = \Pr(\theta = -1)$. The Hamiltonian is as follows:

$$H(p) = \min_{\tilde{\theta}_1} \max_{\tilde{\theta}_2} \mathbb{E}_\theta \left[ (\tilde{\theta}_1 - \theta)^2 d_1 - (\tilde{\theta}_2 - \theta)^2 d_2 \right]$$
$$= 4p(1 - p)(d_1 - d_2).$$

The optimal actions for the Hamiltonian are $\tilde{\theta}_1 = \tilde{\theta}_2 = 1 - 2p$. From Bellman backup, we can get

$$V(T_k, p) = 4p(1 - p)\tilde{D}_k.$$

Therefore, at $T_{2q+1}$, we have

$$V(T_{2q+1}, p) = Vex_p \left( V(T_{2q+2}, p) + 4p(1 - p)D_{2q+1} \right)$$
$$= Vex_p \left( 4p(1 - p)(\tilde{D}_{2q+2} + D_{2q+1}) \right).$$

Notice that $\tilde{D}_{2q+2} + D_{2q+1} > 0$ (property 3) and $\tilde{D}_k < 0$ for all $k < 2q + 1$ (property 4), $T_{2q+1}$ is the first time such that the right-hand side term inside the convexification operator, i.e., $4p(1 - p)(\tilde{D}_{2q+2} + D_{2q+1})$, becomes concave. Therefore, splitting of belief happens at $T_{2q+1}$ with $p^1 = 0$ and $p^2 = 1$. Player 1 plays $\tilde{\theta}_1 = -1$ (resp. $\tilde{\theta}_1 = 1$) with probability 1 if $\theta = -1$ (resp. $\theta = 1$), i.e., Player 1 reveals its type. This result is consistent with Hexner's.

**Dual game.** To find Player 2's strategy, we need to derive the conjugate value which follows

$$V^*(t, \hat{p}) = \begin{cases} \max_{i \in \{1,2\}} \hat{p}[i] & \forall t \geq T_{2q+1} \\ \hat{p}[2] - \tilde{D}_t \left( 1 - \frac{\hat{p}[1] - \hat{p}[2]}{4\tilde{D}_t} \right)^2 & \forall t < T_{2q+1},\ 4\tilde{D}_t \leq \hat{p}[1] - \hat{p}[2] \leq -4\tilde{D}_t \\ \hat{p}[1] & \forall t < T_{2q+1},\ \hat{p}[1] - \hat{p}[2] \geq 4\tilde{D}_t \\ \hat{p}[2] & \forall t < T_{2q+1},\ \hat{p}[1] - \hat{p}[2] < 4\tilde{D}_t \end{cases}$$

Here $\hat{p} \in \nabla_{p_\theta} V(0, p_\theta)$ and $V(0, p_\theta) = 4p[1]p[2]\tilde{D}_0$. For any particular $p_* \in \Delta(2)$, from the definition of subgradient, we have $\hat{p}[1]p_*[1] + \hat{p}[2]p_*[2] = 4p_*[1]p_*[2]\tilde{D}_0$ and $\hat{p}[1] - \hat{p}[2] = 4(p_*[2] - p_*[1])\tilde{D}_0$. Solving these to get $\hat{p} = [4p_*[2]^2\tilde{D}_0, 4p_*[1]^2\tilde{D}_0]^T$. Therefore $\hat{p}[1] - \hat{p}[2] = 4\tilde{D}_0(1 - 2p_*[1]) \in [4\tilde{D}_0, -4\tilde{D}_0]$, and

$$V^*(0, \hat{p}) = \hat{p}[2] - \tilde{D}_0 \left( 1 - \frac{\hat{p}[1] - \hat{p}[2]}{4\tilde{D}_0} \right)^2.$$

Notice that $V^*(t, \hat{p})$ is convex to $\hat{p}$ since $\tilde{D}_0 < 0$ (property 4) for all $t \in [0, T]$. Therefore, there is no splitting of $\hat{p}$ during the dual game, i.e., $\tilde{\theta}_2 = 1 - 2p$. This result is also consistent with results in Hexner (1979).

## F.2 EXAMPLE OF A TURN-BASED GAME

We present a zero-sum variant of the classic beer-quiche game, which is a turn-based incomplete-information game with a perfect Bayesian equilibrium. Unlike in Hexner's game, Player 1 in beer-quiche game wants to maximize its payoff, and Player 2 wants to minimize it; hence, Vex becomes a Cav. We solve the game through backward induction (from $t = 2, 1, 0$) of its primal and dual values (denoted by $V$ and $V^*$ respectively). Players 1 and 2 make their respective decisions at $t = 0$ and $t = 1$, and the game ends at $t = 2$. The state $x$ at a time $t$ encodes the history of actions taken until $t$.

**Primal game:** First, we compute the equilibrium strategy of Player 1 using the primal value. At the terminal time step ($t = 2$), based on Fig. 10, the value for Player 1 is the following:

$$V(2, x, p) = \begin{cases} 4p_T - 2 & \text{if } x = (B, b) \\ p_T & \text{if } x = (B, d) \\ 2p_T - 1 & \text{if } x = (Q, b) \\ 2 - 2p_T & \text{if } x = (Q, d) \end{cases} \tag{43}$$

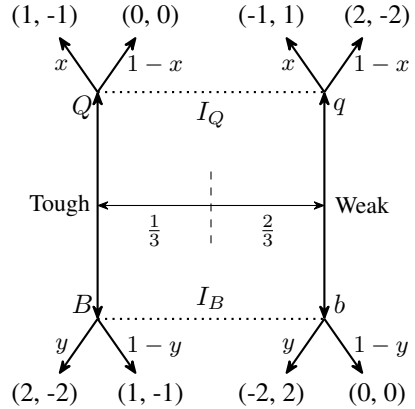

Figure 10: Zero-Sum Beer-Quiche Game

At the intermediate time step ($t = 1$), it is Player 2's turn to take an action. Therefore, the value is a function of Player 1's action at $t = 0$ and Player 2's current action. And for the same reason, the value is not a *concavification* (Cav) over the RHS term.

$$V(1, x, p) = \min_{v \in \{b, d\}} V(2, (x, v), p). \tag{44}$$

We can find the best responses of Player 2 for both actions of Player 1. This leads to

$$V(1, x, p) = \begin{cases} p_T & \text{if } x = B, \ 3p_T - 2 \geq 0 \quad (v^* = d) \\ 4p_T - 2 & \text{if } x = B, \ 3p_T - 2 < 0 \quad (v^* = b) \\ 2 - 2p_T & \text{if } x = Q, \ 4p_T - 3 \geq 0 \quad (v^* = d) \\ 2p_T - 1 & \text{if } x = Q, \ 4p_T - 3 < 0 \quad (v^* = b) \end{cases}. \tag{45}$$

Finally, at the beginning of the game ($t = 0$), we have

$$V(0, \emptyset, p) = \text{Cav} \left( \max_{u \in \{B, Q\}} V(1, u, p) \right). \tag{46}$$

Cav is achieved by taking the concave hull with respect to $p_T$:

$$V(0, \emptyset, p) = \begin{cases} 5p_T/2 - 1 & \text{if } p_T < 2/3 \\ p_T & \text{if } p_T \geq 2/3 \end{cases}. \tag{47}$$

When $p_T \in [0, 2/3)$,

$$V(0, \emptyset, p) = \lambda \max_u V(1, u, p^1) + (1 - \lambda) \max_u V(1, u, p^2),$$

where $p^1 = [0, 1]^T$, $p^2 = [2/3, 1/3]^T$, and $\lambda p^1 + (1 - \lambda) p^2 = p$.

Therefore, when $p_T = 1/3$, $\lambda = 1/2$, Player 1's strategy is:

$$\begin{aligned} \Pr(u = Q|T) = \frac{\lambda p^1[1]}{p[1]} = 0, & \qquad \Pr(u = Q|W) = \frac{\lambda p^1[2]}{p[2]} = 3/4, \\ \Pr(u = B|T) = \frac{(1 - \lambda)p^2[1]}{p[1]} = 1, & \quad \Pr(u = B|W) = \frac{(1 - \lambda)p^2[2]}{p[2]} = 1/4. \end{aligned} \tag{48}$$

**Dual game:** To solve the equilibrium of Player 2, we first derive the dual variable $\hat{p} \in \partial_p V(0, \emptyset, p)$ for $p = [1/3, 2/3]^T$. By definition, $\hat{p}^T p$ defines the concave hull of $V(0, \emptyset, p)$, and therefore we have

$$\begin{aligned} [1/3, 2/3]\hat{p} &= V(0, \emptyset, p) = -1/6 \\ [0, 1]\hat{p} &= V(0, \emptyset, [0, 1]) = -1. \end{aligned} \tag{49}$$

This leads to $\hat{p} = [3/2, -1]^T$.

At the terminal time, we have

$$V^*(2, x, \hat{p}) = \min\{\hat{p}[1] - g_T(x), \hat{p}[2] - g_W(x)\}$$

$$= \begin{cases} \min\{\hat{p}[1] - 2, \hat{p}[2] + 2\} & \text{if } x = (B, b) \\ \min\{\hat{p}[1] - 1, \hat{p}[2]\} & \text{if } x = (B, d) \\ \min\{\hat{p}[1] - 1, \hat{p}[2] + 1\} & \text{if } x = (Q, b) \\ \min\{\hat{p}[1], \hat{p}[2] - 2\} & \text{if } x = (Q, d) \end{cases} \tag{50}$$

At $t = 1$, we have

$$V^*(1, u, \hat{p}) = \text{Cav}_{\hat{p}}\left(\max_v V^*(2, (u, v), \hat{p})\right). \tag{51}$$

When $u = B$, the conjugate value is a concave hull of a piece-wise linear function:

$$V^*(1, B, \hat{p}) = \text{Cav}_{\hat{p}}\left(\begin{cases} \hat{p}[1] - 1 & \text{if } \hat{p}[2] \geq \hat{p}[1] - 1 & (v^* = d) \\ \hat{p}[2] & \text{if } \hat{p}[2] \in [\hat{p}[1] - 2, \hat{p}[1] - 1) & (v^* = b) \\ \hat{p}[1] - 2 & \text{if } \hat{p}[2] \in [\hat{p}[1] - 4, \hat{p}[1] - 2) & (v^* = d) \\ \hat{p}[2] + 2 & \text{if } \hat{p}[2] < \hat{p}[1] - 4 & (v^* = b) \end{cases}\right)$$

$$= \begin{cases} \hat{p}[1] - 1 & \text{if } \hat{p}[2] \geq \hat{p}[1] - 1 & (v^* = d) \\ 2/3\hat{p}[1] + 1/3\hat{p}[2] - 2/3 & \text{if } \hat{p}[2] \in [\hat{p}[1] - 4, \hat{p}[1] - 1) & (\text{mixed strategy}) \\ \hat{p}[2] + 2 & \text{if } \hat{p}[2] < \hat{p}[1] - 4 & (v* = b) \end{cases} \tag{52}$$

For $\hat{p} = [3/2, -1]^T$ which satisfies $\hat{p}[2] \in [\hat{p}[1] - 4, \hat{p}[1] - 1)$, Player 2 follows a mixed strategy determined based on $\{\lambda_1, \lambda_2, \lambda_3\} \in \Delta(3)$ and $\hat{p}^j \in \mathbb{R}^2$ for $j = 1, 2, 3$ such that:

(i) At least one of $\hat{p}^j$ for $j = 1, 2, 3$ should satisfy $\hat{p}[2] = \hat{p}[1] - 1$ and another $\hat{p}[2] = \hat{p}[1] - 4$. These conditions are necessary for $V^*(1, B, \hat{p})$ to be a concave hull:

$$V^*(1, B, \hat{p}) = \sum_{j=1}^{3} \lambda_j \max_v V^*(2, (B, v), \hat{p}^j). \tag{53}$$

(ii) $\sum_{j=1}^{3} \lambda_j \hat{p}^j = \hat{p}$.

These conditions lead to $\lambda_1 = 1/2$ and $\lambda_2 + \lambda_3 = 1/2$. Therefore, when Player 1 picks beer, Player 2 chooses to defer and bully with equal probability.

When $u = Q$, we similarly have

$$V^*(1, Q, \hat{p}) = \begin{cases} \hat{p}[1] & \text{if } \hat{p}[2] \geq \hat{p}[1] + 2 & (v^* = d) \\ ... & \text{if } \hat{p}[2] \in [\hat{p}[1] - 2, \hat{p}[1] + 2) & (\text{mixed strategy}) \\ \hat{p}[2] + 1 & \text{if } \hat{p}[2] < \hat{p}[1] - 2 & (v* = b) \end{cases} \tag{54}$$

The derivation of the concave hull when $\hat{p}[2] \in [\hat{p}[1] - 2, \hat{p}[1] + 2)$ is omitted, because, for $\hat{p} = [3/2, -1]^T$, $V^*(1, Q, \hat{p}) = \hat{p}[2] + 1 = 0$ while $v^* = b$, i.e. if Player 1 picks quiche, Player 2 chooses to bully with a probability of 1.

## G  HEXNER'S GAME SETTINGS, BASELINES, AND GROUND TRUTH

### G.1  GAME SETTINGS

The players move in an arena bounded between $[-1, 1]$ in all directions. All games in the paper follow 2D/3D point dynamics as follows: $\dot{x}_j = Ax_j + Bu_j$, where $x_j$ is a vector of position and velocity and $u_j$ is the action for player $j$. Note that we use $u$ and $v$ in the optimization problems $P_1$ and $P_2$ to represent player 1 and player 2's actions respectively. The type independent effort loss for each player $j$ is defined as $l_j(u_j) = u_j^\top R_j u_j$, where $R_1 = \text{diag}(0.05, 0.025)$ and $R_2 = \text{diag}(0.05, 0.1)$. For the higher dimensional case, $R_1 = \text{diag}(0.05, 0.05, 0.025)$ and $R_2 = \text{diag}(0.05, 0.05, 0.1)$. Note that, in the incomplete information case, P1 is able to get better payoff by hiding the target because P2 incurs higher effort cost, and hence cannot accelerate as fast as P1.

## G.2 GROUND TRUTH FOR HEXNER'S GAME

For the 4-stage and 10-stage Hexner's game, there exists analytical solution to the equilibrium policies via solving the HJB for respective players.

$$u_j = -R_j^{-1} B_j^\top K_j x_j + R_j^{-1} B_j^\top K_j \Phi_j z \tilde{\theta}_j,$$

based on the reformulation outlined below in which players' action $\tilde{\theta}_j \in \mathbb{R}$ become 1D and are decoupled from the state: where $\Phi_j$ is a state-transition matrix that solves $\dot{\Phi}_j = A_j \Phi_j$, with $\Phi_j(T)$ being an identity matrix, and $K_j$ is a solution to a continuous-time differential Ricatti equation:

$$\dot{K}_j = -A_j^\top K_j - K_j A_j + K_j^\top B_j R_j^{-1} B_j^\top K_j, \tag{55}$$

Finally, by defining

$$d_j = z^\top \Phi_j^\top K_j B_j R_j^{-1} B_j^\top K_j^\top \Phi_i z$$

and the critical time

$$t_r = \arg\min_t \int_0^t (d_1(s) - d_2(s)) ds$$

and

$$\tilde{\theta}_j(t) = \begin{cases} 0, & t \in [0, t_r] \\ \theta, & t \in (t_r, T] \end{cases}.$$

As explained in Sec.6, P1 chooses $\theta_1 = 0$ until the critical time $t_r$ and P2 follows.

Note that in order to compute the ground truth when time is discretized with some $\tau$, we need the discrete counterpart of equation 55, namely the discrete-time Ricatti difference equation and compute the matrices $K$ recursively.

## G.3 OPENSPIEL IMPLEMENTATIONS AND HYPERPARAMETERS

We use OpenSpiel (Lanctot et al., 2019), a collection of various environments and algorithms for solving single and multi-agent games. We select OpenSpiel due to its ease of access and availability of wide range of algorithms. The first step is to write the game environment with simultaneous moves for the stage-game and the multi-stage games (with 4 decision nodes). Note that to learn the policy, the algorithms in OpenSpiel require conversion from simultaneous to sequential game, which can be done with a built-in method.

In the single-stage game, P1 has two information states representing its type, and P2 has only one information state (i.e., the starting position of the game which is fixed). In the case of the 4-stage game, the information state (or infostate) is a vector consisting of the P1's type (2-D: [0, 1] for type-1, [1, 0] for type-2), states of the players (8-D) and actions of the players at each time step $(4 \times 2 \times U)$. The 2-D "type" vector for P2 is populated with 0 as it has no access to P1's type. For example, the infostate at the final decision node for a type-1 P1 could be $[0, 1, x^{(8)}, \mathbb{1}_{u_0}^{(U)}, \mathbb{1}_{d_0}^{(U)}, \cdots, \mathbb{1}_{d_2}^{(U)}, \mathbf{0}^{(U)}, \mathbf{0}^{(U)}]$, and $[0, 0, x^{(8)}, \mathbb{1}_{u_0}^{(U)}, \mathbb{1}_{d_0}^{(U)}, \cdots, \mathbb{1}_{d_2}^{(U)}, \mathbf{0}^{(U)}, \mathbf{0}^{(U)}]$ for P2, where $u_k, d_k$ represent the index of the actions at $k^{th}$ decision node, $k = 0, 1, 2, 3$

The hyperparameters for DeepCFR is listed in table 2

## G.4 JOINT-PERTURBATION SIMULTANEOUS PSEUDO-GRADIENT (JPSPG)

The core idea in the JPSPG algorithm is the use of pseudo-gradient instead of computing the actual gradient of the utility to update players' strategies. By perturbing the parameters of a utility function (which consists of the strategy), an unbiased estimator of the gradient of a smoothed version of the original utility function is obtained. Computing pseudo-gradient can often be cheaper than computing exact gradient, and at the same time suitable in scenarios where the utility (or objective) functions are "black-box" or unknown. Building on top of pseudo-gradient, Martin & Sandholm (2024) proposed a method that estimates the pseudo-gradient with respect to all players' strategies simultaneously. The implication of this is that instead of multiple calls to estimate the pseudo-gradient, we can estimate the

Table 2: Hyperparameters for DeepCFR Training

| | |
|---|---|
| Policy Network Layers | (256, 256) |
| Advantage Network Layers | (256, 256) |
| Number of Iterations | 1000 (100, for $U = 16$) |
| Number of Traversals | 5 (10, for $U = 16$) |
| Learning Rate | 1e-3 |
| Advantage Network Batch Size | 1024 |
| Policy Network Batch Size | 10000 (5000 for $U = 16$) |
| Memory Capacity | 1e7 (1e5 for $U = 16$) |
| Advantage Network Train Steps | 1000 |
| Policy Network Train Steps | 5000 |
| Re-initialize Advantage Networks | True |

pseudo-gradient in a single evaluation. More formally, let $\mathbf{f} : \mathbb{R}^d \to \mathbb{R}^n$ be a vector-valued function. Then, its smoothed version is defined as:

$$\mathbf{f}_\sigma(\mathbf{x}) = \mathbb{E}_{\mathbf{z} \sim \mu} \mathbf{f}(\mathbf{x} + \sigma \mathbf{z}), \tag{56}$$

where $\mu$ is a $d$-dimensional standard normal distribution, $\sigma \neq 0 \in \mathbb{R}$ is a scalar. Then, extending the pseudo-gradient of a scalar-valued function to a vector-valued function, we have the following pseudo-Jacobian:

$$\nabla \mathbf{f}_\sigma(\mathbf{x}) = \mathbb{E}_{\mathbf{z} \sim \mu} \frac{1}{\sigma} \mathbf{f}(\mathbf{x} + \sigma \mathbf{z}) \otimes \mathbf{z}, \tag{57}$$

where $\otimes$ is the tensor product.

Typically, in a game, the utility function returns utility for each player given their strategy. Let $\mathbf{u} : \mathbb{R}^{n \times d} \to \mathbb{R}^n$ be the utility function in a game with $n$ players, where each player has a $d$-dimensional strategy. Then, the simultaneous gradient of $\mathbf{u}$ would be a function $\mathbf{v} : \mathbb{R}^{n \times d} \to \mathbb{R}^{n \times d}$. That is, row $i$ of $\mathbf{v}(\mathbf{u})$ is the gradient of the utility of the player $i$ with respect to its strategy, $\mathbf{v}_i = \nabla_i \mathbf{u}_i$. As a result, we can rewrite $\mathbf{v}$ concisely as: $\mathbf{v} = \texttt{diag}(\nabla \mathbf{u})$, where $\nabla$ is the Jacobian. With these we have the following:

$$\begin{aligned} \mathbf{v}_\sigma(\mathbf{x}) &= \texttt{diag}(\nabla \mathbf{u}_\sigma(\mathbf{x})) \\ &= \texttt{diag}\left( \mathbb{E}_{\mathbf{z} \sim \mu} \frac{1}{\sigma} \mathbf{u}_\sigma(\mathbf{x} + \sigma \mathbf{z}) \otimes \mathbf{z} \right) \\ &= \mathbb{E}_{\mathbf{z} \sim \mu} \frac{1}{\sigma} \texttt{diag}\left( \mathbf{u}_\sigma(\mathbf{x} + \sigma \mathbf{z}) \otimes \mathbf{z} \right) \\ &= \mathbb{E}_{\mathbf{z} \sim \mu} \frac{1}{\sigma} \mathbf{u}_\sigma(\mathbf{x} + \sigma \mathbf{z}) \odot \mathbf{z}, \end{aligned} \tag{58}$$

where $\odot$ is element-wise product and a result of the fact that $\texttt{diag}(\mathbf{a} \otimes \mathbf{b}) = \mathbf{a} \odot \mathbf{b}$. Hence, by evaluating Eq. 58 once, we get the pseudo-gradient associated with all players, making the evaluation constant as opposed to linear in number of players.

Once the pseudo-gradients are evaluated, the players update their strategy in the direction of the pseudo-gradient, assuming each player is interested in maximing their respective utility.

**JPSPG Implementation.** In games with discrete-action spaces, where strategy is the probability distribution over the actions, JPSPG can be directly applied to get mixed strategy. However, for continuous-action games, a standard implementation would result in pure strategy solution than mixed. In order to compute a mixed strategy, we can turn into neural network as a strategy with an added randomness that can be learned as described in Martin & Sandholm (2023; 2024). We similarly define two strategy networks for each player, the outputs of which are scaled based on the respective action bounds with the help of hyperbolic tangent (`tanh`) activation on the final layer. The

input to the strategy networks (a single hidden layered neural network with 64 neurons and output neuron of action-space dimension) are the state of the player and a random variable whose mean and variance are trainable parameters. We follow the architecture as outlined by Martin & Sandholm (2024) in their implementation of continuous-action Goofspiel. We would like to thank the authors for providing an example implementation of JPSPG on a normal-form game.

In the normal-form Hexner's game, P1's state $\mathbf{x}_1 = \{x_1, y_1, \texttt{type}\}$, and P2's state $\mathbf{x}_2 = \{x_2, y_2\}$. $x_i$, and $y_i$ denote the x-y coordinates of the player $i$. In 4-stage case, we also include x-y velocities in the state and append the history of actions chosen by both P1 and P2 into the input to the strategy network. As an example, P1's input at the very last decision step a vector $[x_1, y_1, v_{x_1}, v_{y_1}, x_2, y_2, v_{x_2}, v_{y_2}, \texttt{type}, u_{1_x}, u_{1_y}, d_{1_x}, d_{1_y}, u_{2_x}, u_{2_y}, d_{2_x}, d_{2_y}, u_{3_x}, u_{3_y}, d_{3_x}, d_{3_y}] \in \mathbb{R}^{21}$, where $u_j$ and $d_j$ represent actions of P1 and P2, respectively, at $j^{th}$ decision point. P2's input, on the other hand, is the same without the $\texttt{type}$ information making it a vector in $\mathbb{R}^{20}$.

### G.5 SAMPLE TRAJECTORIES

Here we present sample trajectories for three different initial states for each P1 type. The policies learned by CAMS results in trajectories that are significantly close to the ground truth than the other two algorithms.

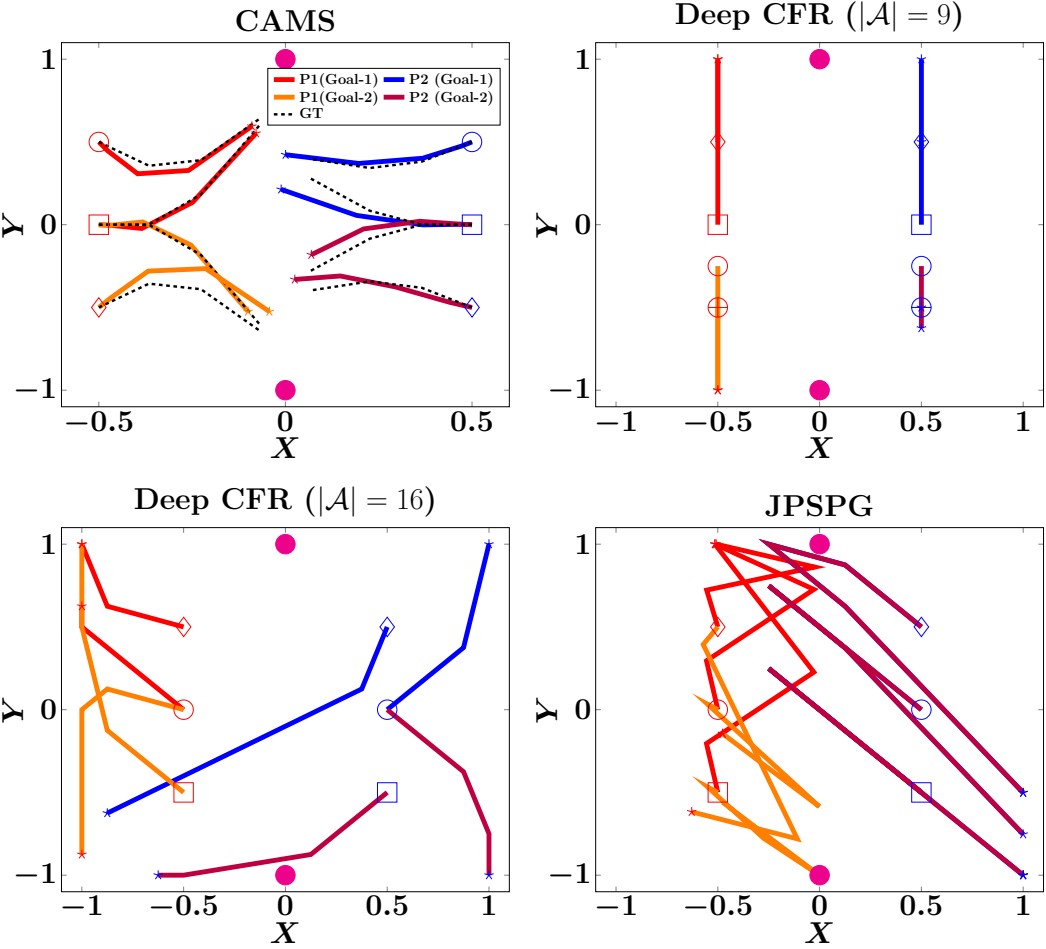

Figure 11: Trajectories generated using CAMS (primal game), DeepCFR, and JPSPG. The initial position pairs are marked with same marker and the final with star. The trajectories from CAMS are close to the ground-truth while those from DeepCFR and JPSPG are not.

### G.6 VALUE NETWORK TRAINING DETAILS

**Data Sampling:** At each time-step, we first collect training data by solving the optimization problem ($P_1$ or $P_2$). Positions are sampled uniformly from [-1, 1] and velocities from $[-\bar{v}_t, \bar{v}_t]$ computed as $\bar{v}_t = t \times u_{max}$, where $u_{max}$ is the maximum acceleration. For the unconstrained game, $u_{max} = 12$ for both P1 and P2. For the constrained case, $u_{x_{max}} = 6$, $u_{y_{max}} = 12$ for P1 and $u_{x_{max}} = 6$, $u_{y_{max}} = 4$ for P2. During training, the velocities are normalized between [-1, 1]. The belief $p$ is then sampled uniformly from $[0, 1]$. For the dual value, we first determine the upper and lower bounds of $\hat{p}$ by computing the sub-gradient $\partial_p V(t_0, \cdot, \cdot)$ and then sample uniformly from $[\hat{p}^-, \hat{p}^+]$.

**Training:** We briefly discuss the training procedure of the value networks. As mentioned in the main paper, both the primal and the dual value functions are convex with respect to $p$ and $\hat{p}$ respectively. As a result, we use Input Convex Neural Networks (ICNN) (Amos et al., 2017) as the neural network architecture. Starting from $T - \tau$, solutions of the optimization problem $P_1$ for sampled $(X, p)$ is saved and the convex value network is fit to the saved training data. The model parameters are saved and are then used in the optimization step at $T - 2\tau$. This is repeated until the value function at $t = 0$ is fit. The inputs to the primal value network are the joint states containing position and velocities of the players $X$ and the belief $p$.

The process for training the dual value is similar to that of the primal value training. The inputs to the dual value network are the joint states containing position and velocities of the players $X$ and the dual variable $\hat{p}$.

## H HYPERPARAMETER SWEEP FOR PG BASELINES

Here we report a sweep of hyperparameters across different learning rates and entropy coefficient for the PG MMD and PPO algorithms. Specifically, we run the algorithms with learning rates of $\{2.5e - 5, 2.5e - 4, 2.5e - 2, 2.5e - 1\}$, and entropy coefficient of $\{0.01, 0.05, 0.1, 0.2\}$. We also run RNaD with all four learning rates. The sweep is reported in Fig. 12.

## I SOLVING HEXNER'S GAME VIA MPC

Here we solve a 2D Hexner's primal game with $I = 2$, $K = 10$, and other settings following App. G. With these settings and using the equilibrium in App. G.2, the true type revelation time is $t_r = 0.5$ second. We directly solve the minimax problem by autodiffing the gradient of the sum of payoffs from the $2^{10}$ paths of the game tree. At each infostate along each path, P1's strategy is modeled by a neural network that takes in $(t, x, p)$ and outputs $I$ action prototypes and an $I$-by-$I$ logit matrix that encodes the type-dependent probabilities of taking each of the action prototypes, and P2's best response is modeled by a separate neural network that takes in $(t, x, p)$ and outputs a single action. With a DS-GDA solver, the search successfully converges to the GT equilibrium. Fig. 13 illustrates the NE trajectories for one particular initial state and the corresponding belief dynamics.

## J A DIFFERENTIABLE 11-VS-11 AMERICAN FOOTBALL GAME

We model a single running/pass play as a 2p0s1 game between the offense (P1) and defense (P2) teams. Each player is a point mass with double-integrator dynamics on a 2D plane. Time is discretised with macro step $\Delta t = \tau$ and $K = T/\tau$ steps, and each macro step is resolved by $n_{\text{sub}}$ semi-implicit Euler substeps for stability.

**State, controls, and bounds.** Let $N = 11$ be players per team. offense positions and velocities are $X^{(1)}, V^{(1)} \in \mathbb{R}^{N \times 2}$; defense $X^{(2)}, V^{(2)} \in \mathbb{R}^{N \times 2}$. We pack them into a state vector $x = [X^{(1)}, V^{(1)}, X^{(2)}, V^{(2)}] \in \mathbb{R}^{8N}$. At each step, the teams apply accelerations $U_1, U_2 \in \mathbb{R}^{N \times 2}$ (stacked later as $u = [u_1; u_2] \in \mathbb{R}^{4N}$). Kinematic saturations enforce a playable box of half-width BOX_POS and box-limited speeds and accelerations BOX_VEL, BOX_ACC by componentwise clamping after each substep.

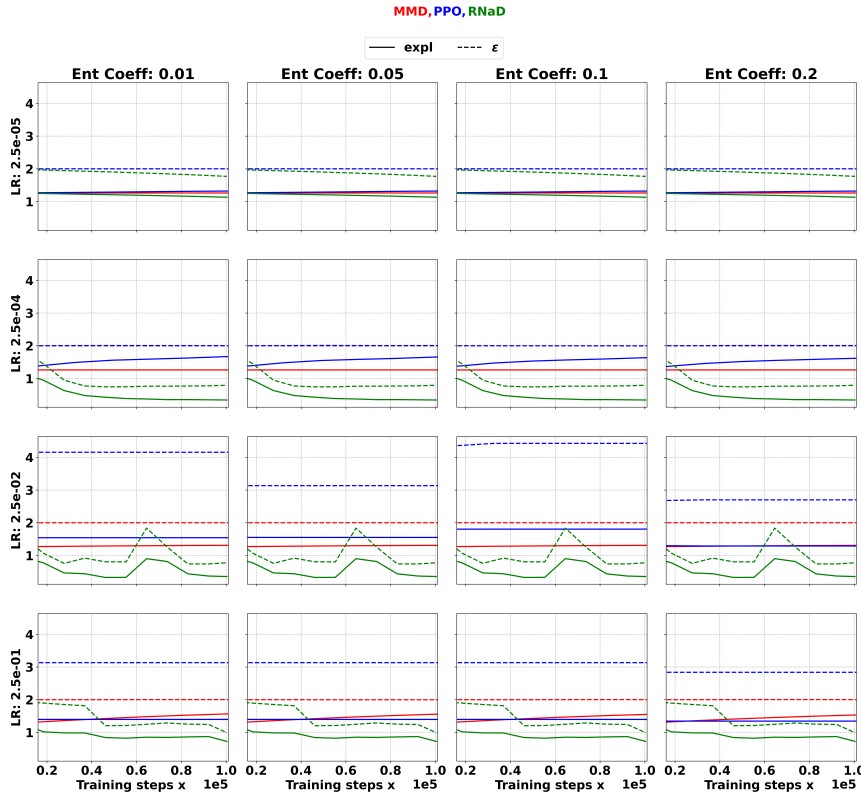

Figure 12: Hyperparameter sweep for the baseline PG algorithms. RNaD, being a non-standard PG algorithm, doesn't use entropy coefficient the way MMD and PPO do; hence, we copy the same plot across different entropy coefficient value for reference.

**Differentiable tackle dynamics (smooth contact and merge).** During a substep with duration $\delta t = \tau/n_{\text{sub}}$, we first compute a soft, pairwise "stickiness" weight between an attacker $i$ and a defender $j$:

$$w_{ij} = \sigma\Big(k_{\text{tackle}}\big(r_{\text{thr}}^2 - \|X_i^{(1)} - X_j^{(2)}\|^2\big)\Big),$$

where $\sigma(z) = 1/(1 + e^{-z})$, $k_{\text{tackle}}$ sets steepness and $r_{\text{thr}}^2 = \text{MERGE\_RADIUS}^2$. These weights form $W \in [0,1]^{N \times N}$. We then compute velocity "sharing" and contact accelerations via convex averaging across opponents:

$$\widehat{V}_i^{(1)} = \frac{V_i^{(1)} + \sum_j w_{ij} V_j^{(2)}}{1 + \sum_j w_{ij}}, \qquad \widehat{V}_j^{(2)} = \frac{V_j^{(2)} + \sum_i w_{ij} V_i^{(1)}}{1 + \sum_i w_{ij}},$$

$$A_{\text{c},i}^{(1)} = \frac{\sum_j w_{ij} A_{\text{c},j}^{(2)}}{1 + \sum_j w_{ij}}, \qquad A_{\text{c},j}^{(2)} = \frac{\sum_i w_{ij} A_{\text{c},i}^{(1)}}{1 + \sum_i w_{ij}},$$

with $A_{\text{c}}^{(\cdot)}$ initialised at zero so the first pass merely defines a contact baseline. This produces smooth, differentiable coupling without hard impulses.

To blend *control* and *contact* accelerations we form state-dependent merge probabilities

$$p_{\text{m},i}^{(1)} = 1 - \exp\Big(-\sum_j w_{ij}\Big), \qquad p_{\text{m},j}^{(2)} = 1 - \exp\Big(-\sum_i w_{ij}\Big),$$

and set

$$A_i^{(1)} = \big(1 - p_{\text{m},i}^{(1)}\big) U_{1,i} + p_{\text{m},i}^{(1)} A_{\text{c},i}^{(1)}, \qquad A_j^{(2)} = \big(1 - p_{\text{m},j}^{(2)}\big) U_{2,j} + p_{\text{m},j}^{(2)} A_{\text{c},j}^{(2)}.$$

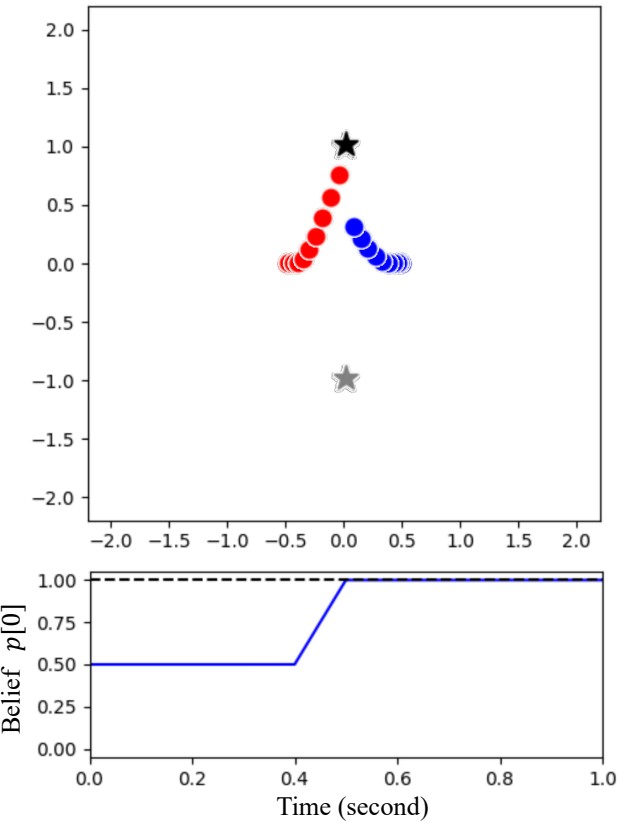

Figure 13: 2D Hexner's game solved by MPC.

We then perform a semi-implicit Euler update with the shared velocities $\widehat{V}$:

$$V^{(1)} \leftarrow \text{clip}\left(\widehat{V}^{(1)} + A^{(1)}\,\delta t,\ \pm\,\text{BOX\_VEL}\right),$$

$$V^{(2)} \leftarrow \text{clip}\left(\widehat{V}^{(2)} + A^{(2)}\,\delta t,\ \pm\,\text{BOX\_VEL}\right),$$

$$X^{(1)} \leftarrow \text{clip}\left(X^{(1)} + V^{(1)}\,\delta t,\ \pm\,\text{BOX\_POS}\right),$$

$$X^{(2)} \leftarrow \text{clip}\left(X^{(2)} + V^{(2)}\,\delta t,\ \pm\,\text{BOX\_POS}\right).$$

**Control-affine analysis** Fix a macro time $k$ and a substep, and treat the current state $(X^{(1)}, V^{(1)}, X^{(2)}, V^{(2)})$ as given. The weights $W$, the merge probabilities $p_{\text{m}}^{(\cdot)}$, the shared velocities $\widehat{V}^{(\cdot)}$, and the contact terms $A_{\text{c}}^{(\cdot)}$ are *functions of the state only* at that substep. Consequently,

$$A^{(1)} = \underbrace{\left(1 - p_{\text{m}}^{(1)}\right)}_{\text{state-only}} \odot U_1 + \underbrace{p_{\text{m}}^{(1)} \odot A_{\text{c}}^{(1)}}_{\text{state-only}}, \qquad A^{(2)} = \left(1 - p_{\text{m}}^{(2)}\right) \odot U_2 + p_{\text{m}}^{(2)} \odot A_{\text{c}}^{(2)}.$$

The semi-implicit update is affine in $(A^{(1)}, A^{(2)})$, hence affine in $(U_1, U_2)$:

$$x_{k+1} = f(x_k) + B_1(x_k)\,u_1 + B_2(x_k)\,u_2,$$

where the "input matrices" $B_1, B_2$ are diagonal masks with entries $(1 - p_{\text{m}}^{(\cdot)})\,\delta t$ in the velocity rows and $(1 - p_{\text{m}}^{(\cdot)})\,\delta t^2$ in the corresponding position rows, all depending only on $x_k$. Thus the map is *control-affine* for any fixed state, and globally *piecewise* control-affine due to the velocity/position clamping at the box limits; the latter introduces non-smooth but almost-everywhere differentiable saturations.

**Tackle probability and running cost.** We summarise the likelihood of a tackle against the ball-carrier (RB) via a differentiable probabilistic OR across all defenders. Let "rb" index the RB on offense, then with the same $W$,

$$p_{\text{tackle}} \;=\; 1 - \prod_{j=1}^{N} \big(1 - w_{\text{rb},j}\big).$$

The running loss at a macro step is

$$\ell_{\text{run}} \;=\; \frac{0.1}{2}\, \tau \big(\text{vec}(U_1)^\top R_1 \,\text{vec}(U_1) \;-\; \text{vec}(U_2)^\top R_2 \,\text{vec}(U_2)\big) \;+\; \lambda_{\text{tackle}}\, p_{\text{tackle}},$$

with $R_1 = R_2 = I_{4N}$ in our defaults, a small control weight to encourage purposeful motion, and $\lambda_{\text{tackle}}$ is the penalty weight for RB being tackled.

**Terminal payoffs: power-push (RB) vs. QB throw** The hidden type $i^\star \in \{0, 1\}$ selects the objective. For the power-push run ($i^\star = 0$), let $(x_{\text{rb}}, y_{\text{rb}})$ denote the RB coordinates and $\alpha_{\text{in}} = -0.8$. The terminal loss is

$$L_{\text{term}}^{\text{run}} \;=\; -\Big(x_{\text{rb}} + \alpha_{\text{in}}|y_{\text{rb}}|\Big),$$

which rewards downfield progress while softly encouraging an inside lane. For the QB throw ($i^\star = 1$), we reward the deepest downfield offensive player, regardless of role:

$$L_{\text{term}}^{\text{throw}} \;=\; -\max_{i \in \{1,\dots,N\}} X_{i,x}^{(1)}.$$

The implemented terminal function is

$$L_{\text{term}} \;=\; \begin{cases} L_{\text{term}}^{\text{run}}, & i^\star = 0, \\ L_{\text{term}}^{\text{throw}}, & i^\star = 1. \end{cases}$$

The overall zero-sum loss is the sum of running losses over $k = 0, \dots, K-1$ plus $L_{\text{term}}$.

**Initial lineup.** For $N = 11$ we instantiate a realistic I-formation offense against a 4–3 base defense in a normalized field window. Coordinates use $x$ as downfield (increasing towards the defense) and $y$ as lateral. offense aligns its line on the line of scrimmage at $x = \text{LINEUP\_OFF\_X}$ with O-line $y$ coordinates $\{-0.80, -0.40, 0.00, 0.40, 0.80\}$ labelled LT, LG, C, RG, RT, a tight end at $y = 1.10$ (right), wide receivers at $y = \pm 1.45$ at the same $x$, a quarterback at $x = \text{LINEUP\_OFF\_X} - 0.20$, a fullback at $x = \text{LINEUP\_OFF\_X} - 0.30$, $y = 0.20$, and the running back at $x = \text{LINEUP\_OFF\_X} - 0.40$, $y = 0.00$. defense places a four-man line at $x = \text{LINEUP\_DEF\_X}$ with $y \in \{-0.60, -0.20, 0.20, 0.60\}$, three linebackers at $x = \text{LINEUP\_DEF\_X} - 0.15$, $y \in \{-0.80, 0.00, 0.80\}$, cornerbacks slightly pressed at $x = \text{LINEUP\_DEF\_X} + 0.05$, $y = \pm 1.45$, and two safeties deep at $x = \text{LINEUP\_DEF\_X} - 0.45$, $y \in \{-0.90, 0.90\}$.

## J.1 Making Sense of Exploitability

While exploitability provides a good indication of whether or not the learned policy is converging to the equilibrium, without a well known baseline outcome, it is often difficult to make sense of how drastic the true nash equilibrium is compared to the resulting policy with non-zero $\epsilon$ exploitability. Hence, here we compare the resulting trajectories when each player switch to their best-response policy to get a sense of how "exploitable" are the strategies, qualitatively. We plot the comparison in Fig. 14. Note that unlike in Fig. 1(b), in Fig. 14 P2 plays a dual game, which results in slight change in P1's policy.

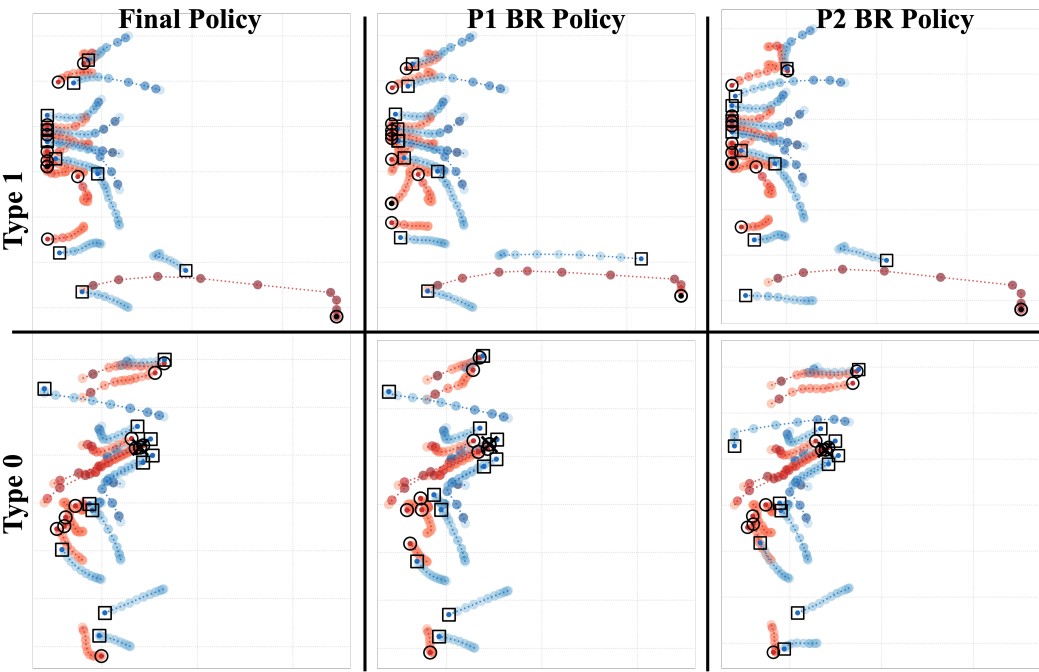

Figure 14: Comparison of resulting behavior from the final P1 and P2's policies vs their respective best-response (BR) policies. Both players' behavior do not shift qualitatively when switching to their BR policies highlighting that the resulting policy with non-zero exploitability is indeed close to the equilibrium.

