# OpenReview forum: "Solving Football by Exploiting Equilibrium Structure of 2p0s Differential Games with One-Sided Information"
_ICLR.cc/2026/Conference — ICLR 2026 Poster_

### Official Review · Reviewer_wDjH · 2025-10-21

**Soundness:** 2
**Presentation:** 1
**Contribution:** 2
**Rating:** 2
**Confidence:** 4

**Summary:**

This paper tackles the problem of solving two-player zero-sum differential games with continuous state/action spaces and one-sided incomplete information (where P1 knows the payoff type, and P2 only has a prior belief). The core theoretical contribution is proving that the Nash Equilibrium (NE) strategies for both players have an "atomic" structure: P1's behavioral strategy concentrates on at most I actions, and P2's on at most I+1 actions, where I is the number of possible payoff types.

**Strengths:**

The proposed CAMS framework effectively utilizes the theoretical structural guarantee to bypass the curse of dimensionality associated with continuous action spaces.

The idea of solving a smaller, reformulated nonconvex-nonconcave problem independent of the original action space size is powerful and clever.

The work bridges differential game theory and computational game theory, offering a viable path to solve realistic problems.

**Weaknesses:**

1. The goal of the paper is to solve an IIEFG with a very large action space. I noticed that the authors claim that there is currently no insightful action-state-time abstraction method, but in my opinion there are lots of works on action abstraction and reducing the size of the game tree. The authors do not fully discuss the differences between this work and existing works [1-20] in the paper. The author should discuss why the previous method of dealing with continuous action space in IIEFGs fails in the 2p0s1 incomplete information game.

2. The introduction is not very well written. The author abruptly raises the 2p0s1 problem, but I don't think this problem is very important because the assumptions are too idealistic. The author's description in the contribution is also not convincing enough. There seems to be no theoretical breakthrough, and I don't see enough innovation in the algorithm.

3. In line 113, the author claims that “convergence guarantee only exists if the NE lies in the interior of the simplex ∆(U)”, which requires citing relevant literature. There are already some works [4, 21] that give bounds for infinite U.

4. The authors use a gridded value approximation algorithm, but do not discuss or use common value approximation methods in IIGs [22-27]. This is because the value of the information set in IIGs changes with strategy, and simple approximations may not fully reflect the true value. While the multigrid method is used to accelerate the value approximation, the paper lacks a compelling explanation or theoretical guarantee for why it is particularly effective or necessary for solving the underlying Hamilton-Jacobi-Isaacs PDEs in this context. The benefits remain empirically observed but not deeply justified.

5. The writing is often dense and confusing, with terminology and notation that are not well-aligned with the broader computational game theory literature. Key concepts, proofs, and the overall narrative are not self-contained enough for a general audience, making it difficult to assess and appreciate the contributions fully.

6. The paper claims to optimize the complexity from U^{2K} to I^K. In my opinion, optimizing the metric from 2K to K is similar to the approach used in CFR. The optimized CFR [28, 29] can enumerate information sets during iterations, rather than states. The complexity reported in this paper is not fundamentally different from CFR, and the claim that the CFR algorithm is U^{2K} requires further discussion.

7. There is already work that studies the Nash equilibrium strategies of both players when one player reveals her strategy [30, 31]. This work needs to further elaborate on its novelty.

8. The experimental benchmarks used in the paper, including the selection of games, metrics, and the fairness of method comparisons, need to be verified. Achieving outstanding performance on widely recognized games is urgently needed (e.g. StarCraft).

[1] David Schnizlein, Michael H. Bowling, and Duane Szafron. Probabilistic state translation in extensive games with large action sets. IJCAI 2009

[2] John Alexander Hawkin, Robert Holte, and Duane Szafron. Automated action abstraction of imperfect information extensive-form games. AAAI 2011

[3] John Alexander Hawkin, Robert Holte, and Duane Szafron. Using sliding windows to generate action abstractions in extensive-form games. AAAI 2012

[4] Sam Ganzfried and Tuomas Sandholm. Action translation in extensive-form games with large action spaces: Axioms, paradoxes, and the pseudo-harmonic mapping. IJCAI 2013

[5] Noam Brown and Tuomas Sandholm. Regret transfer and parameter optimization. AAAI 2014

[6] Noam Brown and Tuomas Sandholm. Simultaneous abstraction and equilibrium finding in games. IJCAI 2015

[7] Shuxin Li, Youzhi Zhang, Xinrun Wang, Wanqi Xue, and Bo An. CFR-MIX: solving imperfect information extensive-form games with combinatorial action space. IJCAI 2021

[8] Carlos Martin and Tuomas Sandholm. Finding mixed-strategy equilibria of continuous-action games without gradients using randomized policy networks. IJCAI 2023

[9] Boning Li, Zhixuan Fang, and Longbo Huang. RL-CFR: Improving action abstraction for imperfect information extensive-form games with reinforcement learning. ICML 2024

[10] Linjie Xu, Diego Perez Liebana, andAlexander Dockhorn. Strategy Game-Playing with Size-Constrained State Abstraction. CoG 2024

[11] Carlos Martin and Tuomas Sandholm. Joint-Perturbation Simultaneous Pseudo-Gradient. IJCAI 2025

[12] Boning Li and Longbo Huang. Efficient online pruning and abstraction for imperfect information extensive-form games. ICLR 2025

[13] Carlos Martin and Tuomas Sandholm. Solving Infinite-Player Games with Player-to-Strategy Networks. arxiv 2025

[14] Weijun Zeng, Yinghao Li, Xiaosi Chen, Zijie Chang, Fei Ge. GNN-ReBeL: Enhancing Neural Belief Representation for Imperfect-Information Games. IEEE SMC 2025

[15] Weijun Zeng, Yinghao Li, Xiaosi Chen, Zijie Chang, Fei Ge. An Investigation of Subgame Depth in ReBeL: Impact on Convergence and Performance in Imperfect-Information Games. IEEE SMC 2025

[16] David Abel, Nate Umbanhowar, Khimya Khetarpal, Dilip Arumugam, Doina Precup and Michael L. Littman. Value Preserving State-Action Abstractions. AISTATS 2020

[17] Noam Brown, Tuomas Sandholm and Brandon Amos. Depth-Limited Solving for Imperfect-Information Games. NeurIPS 2018

[18] Trevor Davis, Kevin Waugh and Michael Bowling. Solving Large Extensive-Form Games with Strategy Constraints. AAAI 2019

[19] Samuel Sokota, Gabriele Farina, David J. Wu, Hengyuan Hu, Kevin A. Wang, J. Zico Kolter and Noam Brown. The Update-Equivalence Framework for Decision-Time Planning. ICLR 2024

[20] Sam Ganzfried. Algorithm for Computing Approximate Nash Equilibrium in Continuous Games with Application to Continuous Blotto. Games 2021

[21] Christian Kroer and Tuomas Sandholm. Discretization of Continuous Action Spaces in Extensive-Form Games. AAMAS 2015

[22] Rosemary Emery-Montemerlo, Geoffrey J. Gordon, Jeff G. Schneider and Sebastian Thrun. Approximate Solutions for Partially Observable Stochastic Games with Common Payoffs. AAMAS 2004

[23] Anne Souquière. Approximation and representation of the value for some differential games with asymmetric information. Int. J. Game Theory 2010

[24] Auke J. Wiggers and Frans A. Oliehoek and Diederik M. Roijers. Structure in the Value Function of Two-Player Zero-Sum Games of Incomplete Information. ECAI 2016

[25] Gabriele Farina and Tuomas Sandholm. Fast Payoff Matrix Sparsification Techniques for Structured Extensive-Form Games. AAAI 2022

[26] Vojtech Kovarík, Dominik Seitz and Viliam Lisý, Jan Rudolf, Shuo Sun and Karel Ha. Value functions for depth-limited solving in zero-sum imperfect-information games. Artif. Intell. 2023

[27] Ratip Emin Berker, Emanuel Tewolde, Ioannis Anagnostides, Tuomas Sandholm and Vincent Conitzer. The Value of Recall in Extensive-Form Games. AAAI 2025

[28] Michael Johanson, Kevin Waugh, Michael H. Bowling and Martin Zinkevich. Accelerating Best Response Calculation in Large Extensive Games. IJCAI 2011

[29] Michael Johanson, Nolan Bard, Marc Lanctot, Richard Gibson, and Michael Bowling. Efficient nash equilibrium approximation through monte carlo counterfactual regret minimization. AAMAS 2012

[30] Samuel Sokota, Ryan D'Orazio, Chun Kai Ling, David J. Wu, J. Zico Kolter and Noam Brown. Abstracting Imperfect Information Away from Two-Player Zero-Sum Games. ICML 2023

[31] Weiming Liu, Haobo Fu, Qiang Fu and Wei Yang. Opponent-Limited Online Search for Imperfect Information Games. ICML 2023

**Questions:**

1. In the problem setting of this paper, does P2 know that P1 knows its belief distribution? If P2 knows that P1 knows its belief distribution, why does P2 let P1 know its belief distribution and do not modify the distribution? I think the problem setting is not very clear.

2. What are the specific steps in the LLM proof? How do you ensure the correctness of the proof? Since I am not good at theoretical analysis, I cannot judge whether the proof is valid.

3. Are the value estimates used only for the CAMS algorithm, or for all comparison methods (CFR, etc.)? If only the CAMS algorithm is used, it will be an unfair comparison.

4. The atomic structure is proven under the Isaacs' condition, which guarantees the existence of pure-strategy NEs in the "non-revealing" games. Could you discuss scenarios or game classes where this condition might fail and what the implications would be for the atomic structure of the overall NE?

5. Algorithms like CFR can be parallelized. Can CAMS be parallelized? Is your experiment parallelized?

6. What does Action Error mean in an experiment?

7. How is I generated from U? What is the size of I?

**Details Of Ethics Concerns:**

The paper mentions "his" a lot of times.

---

> ### Author Response · Authors · 2025-11-22
> **Response to Reviewer wDjH (1/4)**
>
> Thank you for taking the time to review our work, and for recognizing our contribution towards enabling solution to realistic differential game problems.
>
> **Regarding Discrimination/bias/fairness concern**: We appreciate the reviewer’s concern and apologize for any unintended offense. Our use of “he” for player 1 and “she” for player 2 was purely a notational device intended to help the reader keep track of the two roles, in the spirit of conventions sometimes used in the game-theory literature. For example, Osborne and Rubinstein (1994), in the preface “A Note on Personal Pronouns”, explicitly discuss that there is no fully satisfactory convention and adopt fixed genders for players in order to improve readability rather than to express any normative view.
>
> **Reference**
> *Osborne, M. J., & Rubinstein, A. (1994). A course in game theory. MIT press.*
> ***
>
> ### Response to Weaknesses
> ***
> 1. **The authors do not fully discuss the differences between this work and existing works [1-20] in the paper.**
>
>     The key theoretical contribution of this paper is to explain that for 2p0s1 games, NE strategies are **naturally atomic**, and therefore action abstraction is **not necessary**. This is fundamentally because when information is one-sided, the informed player (P1) gains full control of the belief dynamics, which then controls the evolution of players’ strategies. Therefore, this paper is not directly competing with those from the reviewer’s list for better abstraction, but rather just pointing out that a special NE structure exists for this focused set of games and exploiting this structure enables scalable solving of some important games (represented by football as a civil and relatable example).
>
>     We would argue that the novelty of our paper is supported by the fact that this simple and useful structure exploitation technique is not mentioned in any of the listed papers. In fact, there is no evidence that combining the knowledge and methods from this list would enable tractable solve of the football game we presented and solved (which contains a 22-dimensional continuous action space for each player, an 88-dimensional continuous joint state space, a 2D simplex for belief, and 10 time steps).
>
>     We agree with the reviewer that our statement on line 41: “Directly applying existing solvers to differential games would require either insightful action-state-time abstraction or enormous compute. Neither are readily available” was misleading. We meant to say that abstractions for particular games, e.g., football, are not always available, yet we do agree that abstraction **techniques** are widely studied for IIEFGs. In the revision, we change this statement to:
>
>    “Applying existing solvers to differential games would require either game-specific insights or extra computational overhead for automated abstraction [1-20].” We note that [14,15] are not yet available online or in proceedings at the time of this response, but we will make sure to follow up.

---

> ### Author Response · Authors · 2025-11-22
> **Response to Reviewer wDjH (2/4)**
>
> 2. **The authors use a gridded value approximation algorithm, but do not discuss or use common value approximation methods in IIGs [22-27]...While the multigrid method is used to accelerate the value approximation, the paper lacks a compelling explanation or theoretical guarantee for why it is particularly effective or necessary for solving the underlying Hamilton-Jacobi-Isaacs PDEs in this context.**
>
>     - On value approximation: Our case studies demonstrated that the proposed atomic NE structure is useful for common learning modes, including value approximation, MARL (proximal gradient), and MPC (where known differential dynamics is assumed). It is not our intention to compete with other learning techniques (such as [22], an early attempt at solving POSG, and [26], a discussion on depth-limited solving of POSG) that are tangential to the incorporation of the atomic structure.
>
>         [23-25] and [27] are not related to value approximation, which we discuss below:
>         - [23] is the only paper among the list related to differential games. Yet this paper only provides the existence proof and characterization of value for 2p0s differential games rather than proposing methods for value approximation.
>         - [24] discussed the convex-concave structure of POSG but does not propose value approximation algorithms. To the authors’ best knowledge, this convex-concave structure (extended from POMDP) still yields exponentially growing computational complexity backward in time.
>         - [25] discusses payoff matrix sparsification, which is specific to IIEFGs and Poker endgames in particular. For the football game we presented, if we discretize each action dimension by 10 values, we would have a public game tree complexity of $10^{440}$, leading to a payoff matrix of the size $10^{440}$-by-$10{^440}$. There is no evidence that this matrix possesses the sparse structure as studied in [25], and we do not think sparse decomposition is tractable on matrices of this size.
>         - [27] discusses games with imperfect recall. It is not about value approximation.
>
>     - On multigrid: Multigrid has been widely used to accelerate the numerical solution of HJB/HJI-type PDEs on grids, with convergence guarantees in optimal control (Akian et al., 1995) and empirical convergence for differential games (Han & Wan, 2013; Akian & Detournay, 2012). In our setting, it reduces the cost of each PDE solve, allowing us to employ smaller time-step $\tau$ at comparable runtime and thereby reducing the time-discretization error term (1) provided by Theorem 4.2.
>
>         Furthermore, multigrid works by addressing both high and low frequency errors, and hence provides a better quality solution compared to the solution obtained on a single (fine) grid under the same or lower computational budget. Our experiments confirm that multigrid approach is effective in practice: multigrid yields substantial improvements in wall-clock solution time (Table 1), and the policies derived from the resulting value function closely track the ground-truth policies (Fig. 6 and Fig. 7 in the Appendix).
>
>         Note: We only claim empirical benefits, as there currently are no theoretical guarantees of using multigrid method for HJI PDEs, and has been only proven in the case of linear (Braess & Hackbusch, 1983) and certain classes of elliptical non-linear PDEs (Reusken, 1988).
>
>         **References**
>
>         *Olson, Luke. Multigrid Methods. https://lukeo.cs.illinois.edu/files/2015_Ol_encmg.pdf, 2015.*
>
>         *Akian, Marianne, P. Séquier, and Agnks Sulem. "A finite horizon multidimensional portfolio selection problem with singular transactions." Proceedings of 1995 34th IEEE Conference on Decision and Control. Vol. 3. IEEE, 1995.*
>
>         *Han, Dong, and Justin WL Wan. "Multigrid Methods for Second Order Hamilton--Jacobi--Bellman and Hamilton--Jacobi--Bellman--Isaacs Equations." SIAM Journal on Scientific Computing 35.5 (2013): S323-S344.*
>
>         *Akian, Marianne, and Sylvie Detournay. "Multigrid methods for two‐player zero‐sum stochastic games." Numerical Linear Algebra with Applications 19.2 (2012): 313-342.*
>
>         *Braess, D., and W. Hackbusch. “A New Convergence Proof for the Multigrid Method Including the V-Cycle.” SIAM Journal on Numerical Analysis, vol. 20, no. 5, 1983, pp. 967–75. JSTOR, http://www.jstor.org/stable/2157109. Accessed 31 Mar. 2025.*
>
>         *Reusken, A. (1988). Convergence of the multilevel Full Approximation Scheme including the V-cycle. Numerische Mathematik, 53(6), 663–686. doi:10.1007/bf01397135.*

---

> ### Author Response · Authors · 2025-11-22
> **Response to Reviewer wDjH (3/4)**
>
> 3. **In line 113, the author claims that “convergence guarantee only exists if the NE lies in the interior of the simplex ∆(U)”, which requires citing relevant literature.**
>
>     Perloat et al. (2021) provides a proof that if the equilibrium policy is interior, R-NaD converges to this equilibrium. However, convergence for non-interior equilibrium cases is yet to be proved. We would also like to note that the convergence guarantees for interior equilibrium is only for sequence-form games (which is not scalable to large games) but not behavioral-form ones (which is commonly modeled for large games such as Poker).
>
>     **Reference**
>
>     *Julien Perolat, Remi Munos, Jean-Baptiste Lespiau, Shayegan Omidshafiei, Mark Rowland, Pedro Ortega, Neil Burch, Thomas Anthony, David Balduzzi, Bart De Vylder, et al. From poincaré recurrence to convergence in imperfect information games: Finding equilibrium via regularization. In the International Conference on Machine Learning, pp. 8525–8535. PMLR, 2021.*
>
> 4. **The writing is often dense and confusing,...**
>
>     We have put considerable effort into making the paper readable. We would be more than happy to clarify **any specific confusion** that the reviewer might have.
>
> 5. **The paper claims to optimize the complexity from U^{2K} to I^K. In my opinion, optimizing the metric from 2K to K is similar to the approach used in CFR.**
>
>     In our setting, if one discretizes each player’s continuous control at each of the K stages into U grid points and allows stagewise mixed strategies, as is the case for CFR and its variants (including [29]), then the resulting extensive-form game has a game tree complexity of $U^{2K}$.
>
>     Our results show that (1) NE strategies for 2p0s1 differential games are mixtures over I (I+1) atomic action prototypes for P1 (P2); and (2) the NE strategy of each player can be solved with the other player playing deterministic best responses (Thm. 4.1). This leads to the reduced public tree complexity of I^K for P1 and (I+1)^K for P2. As a result, when I is much smaller than U (in particular, U = \infty when the action space is continuous), our result reveals the much smaller intrinsic game tree complexity, i.e., $I^{K} \ll U^{2K}$.
>
>     Finally, [28] is concerned with accelerating best response computation, which is **not a part of computing strategy in CFR** but rather a **tool to evaluate the computed strategies**.
>
> 6. **There is already work that studies the Nash equilibrium strategies of both players when one player reveals her strategy [30, 31].**
>
>     It is not clear to us what “...strategies of both players when one player reveals her strategy” mean and how this is related to [30, 31].
>
>     Nonetheless, we believe that the challenges and solutions discussed in these papers are **tangential** to those of ours. Specifically:
>     - [30] addresses the non-correspondence issue in 2p0s IIEFGs: briefly, if P2 plays in response to P1 in each turn, then P2 would play best yet exploitable responses rather than an NE. In our paper, this issue is addressed by the introduction of the primal and dual games: In the primal game, P1 plays NE against P2’s best response; and in the dual game, P2 plays NE against P1’s best response.
>
>     - [31] proposes to use opponent-limited online search to avoid full common-knowledge closure enumeration during online subgame solving. For the 2p0s1 games we focused on where private information is only related to game types, the common-knowledge closure has a fixed size of $I$ throughout the game until information revelation by the informed player, from which point on the game becomes complete-information. Therefore, such games are not bottlenecked by the recomputation of the common-knowledge closure.

---

> ### Author Response · Authors · 2025-11-22
> **Response to Reviewer wDjH (4/4)**
>
> 7. **The experimental benchmarks used in the paper, including the selection of games, metrics, and the fairness of method comparisons, need to be verified. Achieving outstanding performance on widely recognized games is urgently needed (e.g. StarCraft).**
>
>     Our paper explains the atomic NE structure for 2p0s1 (2p0s differential games with one-sided incomplete information) and demonstrates the significant value of exploiting this structure in existing game solving modes. Practically, our method reduced the computational cost of solving large-scale 2p0s1 games such as football set-pieces from intractable to within an hour. We believe the choice of game settings used in our case studies are necessary (Hexner's game with analytical ground truth) and sufficient (football game with large action space, currently intractable for SOTA solvers, widely recognized).
>
>     It is common for game-theoretic papers to focus on solving one type of games with significant value, e.g., **existing papers**. For our paper, we focus on games with one-sided information, which is necessary for defense, * see previous responses.
>
>     Lastly, setting up a testbed for imperfect-information games with both large action/state spaces and long time horizons would have **its own merit and is of great value**. However, we would like to humbly argue that this infrastructure building task is out of scope of our paper. In fact, **NONE of the 31 relevant papers cited by the reviewer tested their methods on StarCraft**.
>
> ### Response to Questions
> ***
>
> 1. **Does P2 know that P1 knows its belief distribution? If P2 knows that P1 knows its belief distribution, why does P2 let P1 know its belief distribution and do not modify the distribution?**
>
>     We assume that P2 knows $p_0$, which is the distribution P1 uses to draw their payoff type. We also assume that P1 knows P2 knows $p_0$. It is possible in practice that P2 does not have the correct knowledge about $p_0$. However, this does not affect P1 as long as they play their NE strategy, i.e., P1's payoff will not become worse if P2 plays a strategy according to their wrong knowledge about $p_0$.
>
> 2. **What are the specific steps in the LLM proof? How do you ensure the correctness of the proof?**
>
>     We used LLM to complete details for the proof that $V_\tau$ as defined in Eq. (12) is a convexification (please see Page 15). This is proof-read by the authors.
>
> 3. **Are the value estimates used only for CAMS?**
>
>     The value estimates are not unique to CAMS. Our DeepCFR baseline learns an advantage function at each information state, which is central to the notion of regret in CFR. Concretely, DeepCFR trains an advantage (Q-value) network that approximates counterfactual advantages A(I,a), and these learned advantages are exactly the quantities used in the regret-matching updates. Furthermore, both PPO and PG-version of the MMD use value estimates, since they are actor-critic methods.
>
>     CAMS uses a continuous state-value critic, but this does not make the comparison unfair. The goal of our experiments is to compare complete learning schemes, not to force all algorithms to share the same internal value parameterization. This is analogous to standard RL benchmarks that compare value-based methods such as DQN to actor-critic methods such as PPO: different architectures may use values in different ways, but they are still valid baselines against each other.
>
>
> 4. **When Isaacs may fail & implications**
>
>     Isaacs condition fails when $\inf_u sup_v f(x, u, v) \xi \neq \sup_v \inf_u f(x, u, v) \xi$.  This occurs when the minimax theorem cannot be applied, for example when the Hamiltonian is not convex in the minimizing control or not concave in the maximizing control. When it fails, the game doesn't have a value, i.e., the lower and upper values do not coincide.
>
>     For studies on continuous-time dynamical systems, a pair of widely accepted sufficient conditions for Isaacs’ condition to hold are (1) the dynamics is control-affine, and (2) the running loss is convex for P1 and concave for P2. These conditions hold for both Hexner’s and the football games we studied.
>
> 5. **Can CAMS be parallelized?**
>
>     Sub-games can also be parallelized using our method.
>
> 6. **What does “action-error” mean?**
>
>     We have defined action-error(s) in line 401-402. To reiterate, $\epsilon$ computes the expected $l_2$ distance between the chosen action and the ground-truth solution. $\bar{\epsilon}$ is an estimation of the expected error from a 100 different sample trajectories of a game.
>
> 7. **How is I generated from U? What is the size of I?**
>
>     $I$ is the number of payoff types, and is independent from the size of the action space $U$. In both Hexner’s game and football, we used $|I| = 2$. Since both games have continuous action spaces, $U = \infty$.

---

> ### Comment · Reviewer_wDjH · 2025-11-26
>
> Thank you for your detailed response. I have reviewed the revised manuscript and your rebuttal. While I appreciate the clarifications, several of my core concerns regarding the framing, methodology, and evaluation remain unresolved. I have outlined these points below for further consideration.
>
> Regarding Discrimination/bias/fairness concern: The manuscript uses gendered pronouns ("he", "his") for both P1 and P2. To ensure inclusivity and align with modern academic writing standards, I strongly recommend replacing these with neutral terms (e.g., "they", "their", or "it" for the player role) throughout the paper.
>
> Weakness 1. The paper positions itself against tabular CFR to highlight the challenge of large action spaces. However, the specific game structure studied (simultaneous-move over K steps) is a specialized subclass of IIEFGs. Several existing methods [9, 13, 17, 19] are designed to handle large or continuous action spaces and would be more appropriate and competitive baselines than vanilla CFR. The current comparison may not fully demonstrate the advantage over the state-of-the-art.
>
> Weakness 2. The author do not reply me for this weakness.
>
> Weakness 3. Please cite the paper.
>
> Weakness 4. A critical clarification is needed on whether the value estimation is based on the actual state or the information set. The method appears to be state-based. If so, relying on a single value estimate can be unsafe in imperfect information games, as it does not account for the uncertainty in the opponent's private information. The technique in [17] is specifically designed for such scenarios for safe solving. Could the authors please clarify this point?
>
> If the value is state-based, doesn't computing the terminal payoffs require enumerating all possible terminal state pairs, leading to a complexity of $O(I^{2K})$? If this is the case, the significance of reducing the game tree to $O(I^K)$ is unclear, as the payoff computation would become the bottleneck. This seems related to the point raised in [28, 29], which compresses payoff matrices.
>
> Weakness 5 & 6. The definitions of U, I, and K and their role in the value function caused significant confusion. Specifically, the parameter K is critical. In standard IIEFGs, K often represents the total number of moves in the game. Here, it seems to represent the number of steps per player in a simultaneous-move setting. This must be explicitly defined in the manuscript.
> The claimed complexity reduction from $O(U^{2K})$ to $O(I^K)$ is the paper's core contribution. However, if the payoff calculation at the leaves remains $O(I^{2K})$, the overall benefit is diminished. The paper must clearly explain how the proposed method avoids this complexity.
>
> Weakness 7. The approach of restricting P2's belief to I+1 branches resembles a form of opponent modeling. The paper would be strengthened by a explicit discussion of how the proposed primal-dual method and the resulting atomic structure differ from and advance upon prior work in opponent modeling for extensive-form games.
>
> Weakness 8. While the 22-player football game is an impressive demonstration, its use as a primary benchmark presents challenges for evaluation. The true Nash Equilibrium is unknown, making it impossible to measure exploitability or action error meaningfully. The ultimate metric for a game-solving algorithm is often its performance against competitive baselines. It would be more convincing to include results against known bots in established environments, or to use the football game as a qualitative case study alongside quantitative benchmarks with known equilibria. Reporting win rates against a set of scripted or learned strategies would provide a more intuitive and compelling performance measure. If you can do the experiments in well-recognized game such as StarCraft (or other common games), the experiments will be really strong.
>
> In summary, while the theoretical idea of exploiting the equilibrium structure is interesting, the paper in its current form has not fully addressed these fundamental concerns regarding its positioning, methodological safety, computational claims, and experimental validation. I believe addressing these points is essential for establishing the paper's contribution and ensuring its clarity and rigor.

---

> > ### Author Response · Authors · 2025-11-28
> >
> > Thank you for the follow-up.
> >
> > We revised the paper using gender-neutral pronouns wherever necessary. We also added the citation to line 114.
> >
> > Here we address the remaining concerns:
> >
> > 1. **Weakness 1. The paper positions itself against tabular CFR to highlight the challenge of large action spaces. However, the specific game structure studied (simultaneous-move over K steps) is a specialized subclass of IIEFGs. Several existing methods [9, 13, 17, 19] are designed to handle large or continuous action spaces and would be more appropriate and competitive baselines than vanilla CFR. The current comparison may not fully demonstrate the advantage over the state-of-the-art.**
> >
> >     Below we explain (1) why [9] is not used as a baseline, (2) the technique underlying [13] for continuous actions is exactly what we implemented as a baseline, and (3) why [17] and [19] do not address continuous actions.
> >     >[9] Boning Li, Zhixuan Fang, and Longbo Huang. RL-CFR: Improving action abstraction for imperfect information extensive-form games with reinforcement learning. ICML 2024
> >
> >     This paper proposes an outer RL loop in which an action-abstraction policy is learned to dynamically construct abstractions of the original (potentially large or continuous) action space. The algorithm, however, fundamentally relies on a baseline abstraction to define its RL reward, which is feasible for their sole demonstration on HUNL. In many real-world domains, such as football, such a discrete baseline is not available. We also reiterate that our paper provides a theoretically grounded answer to the question “How many discrete actions should we include in an abstraction?” for the specific class of games “2p0s1”.
> >
> >     >[13] Carlos Martin and Tuomas Sandholm. Solving Infinite-Player Games with Player-to-Strategy Networks. arxiv 2025.
> >
> >     This paper studies infinite-player general-sum games, and proposes a Player-to-Strategy Network that maps a continuum of players to strategies and is trained with Shared-Parameter Simultaneous Gradient (SPSG). We first note that their algorithm is designed for **general n-player games** and is evaluated only on **static** infinite-player game benchmarks, while our setting is two-player, zero-sum, and with continuous-time dynamics.
> >
> >     More importantly, [13] closely resembles an earlier paper of the same authors: “Joint-Perturbation Simultaneous Pseudo-Gradient (JPSPG) (Martin & Sandholm, 2024)” where the focus is to address continuous action spaces. Both JPSPG and [13] leverage randomized policy networks (Martin & Sandholm, 2023) to learn the mixed strategy in continuous action games. JPSPG is indeed the algorithm we included as a baseline (please see Fig. 3 for results and G.4 for implementation details). We obtained a version of JPSPG for normal-form game directly from the authors of [13] since the implementation is not yet open sourced.
> >
> >     **References**
> >
> >     *Martin, Carlos, and Tuomas Sandholm. Joint-perturbation simultaneous pseudo-gradient. arXiv preprint arXiv:2408.09306 (2024).*
> >
> >     *Martin, Carlos, and Tuomas Sandholm. Finding mixedstrategy equilibria of continuous-action games without gradients using randomized policy networks. In Proceedings of the International Joint Conference on Artificial Intelligence (IJCAI), 2023.*
> >
> >     >[17] Noam Brown, Tuomas Sandholm and Brandon Amos. Depth-Limited Solving for Imperfect-Information Games. NeurIPS 2018.
> >     [19]  Samuel Sokota, Gabriele Farina, David J. Wu, Hengyuan Hu, Kevin A. Wang, J. Zico Kolter and Noam Brown. The Update-Equivalence Framework for Decision-Time Planning. ICLR 2024.
> >
> >     Both [17] and [19] are decision-time planning techniques for **finite-action** IIEFGs. They do not address continuous action spaces.
> >
> >     [17] proposed online depth-limited solving of sub-games to handle and refine play when the opponent plays actions outside of the abstraction set. Our method does not suffer from this out-of-abstraction issue since in both primal and dual game solving, we have one player playing atomic mixed strategy against the deterministic best response from their opponent. Both the mixed strategy and the best response are defined on the continuous action spaces without abstraction.
> >
> >     [19] reframes decision-time search as last-iterate updates. It is specifically designed to address scalability to non-public information, which is the case for games like Dark Hex. Their contribution neither addresses complexity due to large or continuous action space, nor provides a pathway that avoids discretization of continuous action spaces.

---

> > > ### Author Response · Authors · 2025-11-28
> > >
> > > 2. **Weakness 2. The author do not reply me for this weakness.**
> > >
> > >     >The introduction is not very well written. The author abruptly raises the 2p0s1 problem, but I don't think this problem is very important because the assumptions are too idealistic. The author's description in the contribution is also not convincing enough. There seems to be no theoretical breakthrough, and I don't see enough innovation in the algorithm.
> > >
> > >       - **“The assumptions are too idealistic”**: We refer to “2p0s1” as 2p0s differential games with one-sided incomplete information, continuous action spaces, and continuous state spaces. Examples include (1) sports games where the offense team chooses to execute one of the attack plans, yet often needs to conceal the plan through coordinated team play to keep the defense in dark (Grabowski, 2020); and (2) defense games, e.g., missile/drone swarm offense-defense (Garcia et al., 2019; 2021; Liang et al., 2019) where deceptive and anti-deception strategies are necessary.
> > >
> > >       - **“contribution is also not convincing…no theoretical breakthrough…don't see enough innovation in the algorithm”**: We would like to reiterate that the key contribution of this paper is showing that for 2p0s1 games, NE strategies are naturally atomic, and therefore action discretization to conform the game to SOTA solvers is not necessary. We also show how this atomic structure can be leveraged in various settings (e.g., MARL, MPC) to compute the Nash equilibrium policy in the original continuous action space with much reduced complexity as opposed to solving them by discretizing the action space.
> > >
> > >     **References**
> > >
> > >     *Keith Grabowski. Wake forest’s slow mesh rpo for explosive plays, Oct 2020. URL https://coachandcoordinator.com/2020/10/wake-forests-slow-mesh-rpo-for-explosive-plays/*
> > >
> > >     *Eloy Garcia, David W. Casbeer, and Meir Pachter. Design and analysis of state-feedback optimal strategies for the differential game of active defense. IEEE Transactions on Automatic Control, 64(2):553–568, 2019.*
> > >
> > >     *Eloy Garcia, David W. Casbeer, and Meir Pachter. The complete differential game of active target defense. Journal of Optimization Theory and Applications, 191:675–699, 2021.*
> > >
> > >     *Li Liang, Fang Deng, Zhihong Peng, Xinxing Li, and Wenzhong Zha. A differential game for cooperative target defense. Automatica, 102:58–71, 2019.*
> > >
> > > 3. **Weakness 3. Please cite the paper.**
> > >
> > >     Thank you for reminding. We added the citation to the paper.
> > >
> > > 4. **Weakness 4. A critical clarification is needed on whether the value estimation is based on the actual state or the information set. The method appears...**
> > >
> > >     - **On value estimation**: From Eq. (3), our value function is defined on the **information set** represented by the tuple (time t, public state x, public belief p), because beliefs are sufficient statistics under perfect recall (this public belief representation is also used in ReBeL (Brown et al., 2020)). From Eq. (4), the dual value is defined on (time t, public state x, dual variable $\hat{p}$).
> > >     So from the perspective of the informed player (P1), their strategy does take into account P2’s uncertainty (belief $p$) about P1’s payoff type. From P2’s perspective, however, the dual game reveals that their strategy should be to minimize the maximum risk (see discussion in “the dual game” on Page 4) without considering belief (thus robust against P1’s potential belief manipulation). This primal-dual perspective is unique to games with one-sided information.
> > >
> > >
> > >     - **On game-tree complexity**: For 2p0s IIEFG, each public tree node (info-state) has a branching factor of $U^2$ where $U$ is the size of each player’s action set, because we consider both players playing mixed behavioral strategies. With $K$ time steps, this leads to the game tree complexity of $U^{2K}$. For 2p0s1 in this paper, however, we showed through Thm. 4.1 and 4.2 that we can solve P1's and P2's NE **sequentially** through the primal and dual games. For each game, the opponent only needs to play a deterministic best response, e.g., P2 plays a specific action in response to each of the $I$ actions P1 potentially takes at any information set. This leads to a branching factor of $I$ for P1 ($I+1$ for P2) and tree complexity of $I^K$ ($(I+1)^K$). When $I \ll U$, our method leads to a much reduced game tree without abstraction. This allows us to perform exact gradient descent-ascent via complete game tree traversal.
> > >
> > >
> > >     **Reference**
> > >
> > >     *Brown, N., Bakhtin, A., Lerer, A., & Gong, Q. (2020). Combining deep reinforcement learning and search for imperfect-information games. Advances in neural information processing systems, 33, 17057-17069.*

---

> > > > ### Author Response · Authors · 2025-11-28
> > > >
> > > > 5. **Weaknesses 5 & 6.  The definitions of U, I, and K and their role in the value function caused significant confusion. Specifically, the parameter...**
> > > >
> > > >     By $K$ time-steps, we mean $K$ simultaneous player moves in the game.
> > > >
> > > >     **Regarding payoff calculation at leaves**: As we responded to the last question, at every **simultaneous** decision node of the primal game, P1 solves $I$ actions and their respective best-response actions from P2. Since there are only $I$ action pairs, the branching factor is $I$, rendering the overall complexity as $O(I^{K})$ rather than $O(I^{2K})$.
> > > >
> > > >     Our claim that the atomic NE structure reduces game-tree complexity ($O(U^{2K}) > O(I^{K})$) when $U > I$ is quite straight forward, and is especially useful when $U = \infty$ in the case of continuous action spaces. Should the reviewer remain concerned about this claim, we will do our best to further clarify. Thank you.
> > > >
> > > > 6. **Weakness 7. The approach of restricting P2's belief to I+1 branches resembles a form of opponent modeling...**
> > > >
> > > >     We would like to clarify that P2’s strategy being (I+1)-atomic is a consequence of the primal-dual reformulation of the game and naturally arises (as we proved in Thm. 4.1) in the context of 2p0s1 games we study. To recap, while the primal value is a convexification over the belief space which is a **I-dimensional simplex**, the dual value is a convexification over the dual variable space which is the **I-dimensional real space**. This difference in the support of the convexification explains why P1’s strategies are I-atomic while P2’s are (I+1)-atomic. Therefore, P2’s strategy being (I+1)-atomic is **not** a modeling choice.
> > > >
> > > > 7. **Weakness 8. While the 22-player football game is an impressive demonstration, its use as a primary benchmark presents challenges for evaluation...**
> > > >
> > > >     We agree that the true Nash Equilibrium of the football game is unknown. This is precisely why we presented extensive results on Hexner’s game which does have an analytical Nash Equilibrium. Therefore, as the reviewer suggested, we did actually **use football as a qualitative case study alongside quantitative benchmark (i.e., Hexner’s game) with known analytical benchmark.**
> > > >
> > > > 8. **In summary, while the theoretical idea of exploiting the equilibrium structure is interesting, the paper in its current form has not fully addressed these fundamental concerns regarding its positioning, methodological safety, computational claims, and experimental validation. I believe addressing these points is essential for establishing the paper's contribution and ensuring its clarity and rigor.**
> > > >
> > > >     We thank the reviewer for their comments and support, and welcome further questions regarding remaining concerns on the paper’s positioning, methodological safety, computational claims, and experimental validation.

---

### Official Review · Reviewer_4YnV · 2025-10-31

**Soundness:** 3
**Presentation:** 3
**Contribution:** 3
**Rating:** 6
**Confidence:** 2

**Summary:**

This paper formulates the football control problem as a two-player zero-sum differential game under asymmetric information, where the attacking agent has full state observability while the defending agent has only partial observations. The authors derive the equilibrium structure of such 2P0S games and exploit it to design robust, feedback-based strategies for both agents. By leveraging this game-theoretic formulation, the proposed method computes strategies that are provably robust against worst-case responses. Empirical results in the Google Research Football Environment demonstrate the effectiveness of this approach across multiple offensive-defensive scenarios, showcasing improved coordination, defensive anticipation, and sample efficiency compared to traditional reinforcement learning baselines.

**Strengths:**

- Theory–structure: Clear, well-motivated primal–dual formulation with rigorous atomic-support results (P1 $I$-atomic, P2 $(I{+}1)$-atomic) and a consistency theorem, explaining why equilibria lie on the simplex boundary for 2p0s1.
- Practical impact: Simple but powerful architectural change (action prototypes + logits) that plugs into value approximation, MARL, and MPC and delivers sizable accuracy/efficiency gains; compelling football demonstration otherwise out of reach for standard IIEFG solvers.

**Weaknesses:**

- Empirics emphasize action error and qualitative behavior. However, exploitability or best-response value gaps to certify closeness to NE are not systematically reported.
- The theoretical framework hinges on the 2P0S game structure, where only one player lacks full observability. However, in realistic football environments, both agents typically face partial observability.

**Questions:**

- How does the atomic structure and the primal–dual DP extend when dynamics or observations are stochastic so P1 cannot precisely steer beliefs?
- Can the authors elaborate on how feedback Nash equilibria are computed in practice for the 2P0S differential game setting? Is the computation exact, or does it involve discretization, function approximation, or numerical solvers? If approximations are used, how is the error controlled or bounded?
- The paper seems to focus on relatively isolated one-vs-one or two-agent scenarios. Can the framework be extended to team-based settings with multiple attackers and defenders? How would the equilibrium computation scale or need to change?

---

> ### Author Response · Authors · 2025-11-22
> **Response to Reviewer 4YnV (1/2)**
>
> Thank you for reviewing our work and for acknowledging our contributions.
> ### Response to Weaknesses
> ***
> 1. **However, exploitability or best-response value gaps to certify closeness to NE are not systematically reported.**
>     We empathize with the reviewer’s concern. Because Hexner’s game has analytical NE, computing the error directly to the optimal solution (in continuous action space) would provide a clear signal in assessing the advantage of our method. Exploitability as defined for games with discretized action sets will not be a truthful metric of convergence because it is computed for discretized best responses only, while the true best responses now live in the continuous action spaces. For completeness, we will add [these exploitability comparison plots](https://postimg.cc/67KS4NSV) (anonymous link to the image) as well.
> 2. **However, in realistic football environments, both agents typically face partial observability.**
>
>     We agree that many two-player interactions have hidden information on both sides. Yet we would argue that for many real-world games, solving their one-sided variants, i.e., fully informed attackers vs. partially informed defenders, is of critical importance especially for the defenders, when **solution speed and robustness of strategies are of priority**. Examples of such games include safety-critical scenarios in defense (drone swarm defense with partially known attack targets) (Garcia et al., 2019; 2021; Liang et al., 2019), cybersecurity (defense against data exfiltration) (Durkota et al., 2017; Mc Carthy et al., 2016; Horak et al., 2016; van Dijk et al., 2013), and finance (surveillance against insider trading) (Kyle, 1985; Back, 1992; Wang et al., 2020). All these games have meaningful one-sided information and continuous-time continuous-action settings.
>     **References**
>     *Eloy Garcia, David W. Casbeer, and Meir Pachter. Design and analysis of state-feedback optimal strategies for the differential game of active defense. IEEE Transactions on Automatic Control, 64(2):553–568, 2019.*
>     *Eloy Garcia, David W. Casbeer, and Meir Pachter. The complete differential game of active target defense. Journal of Optimization Theory and Applications, 191:675–699, 2021.*
>     *Li Liang, Fang Deng, Zhihong Peng, Xinxing Li, and Wenzhong Zha. A differential game for cooperative target defense. Automatica, 102:58–71, 2019.*
>
>      *Karel Durkota, Viliam Lisy, Christopher Kiekintveld, Karel Horak, Branislav Bosansky, and Tomas Pevny. Optimal strategies for detecting data exfiltration by internal and external attackers. In Decision and Game Theory for Security (GameSec 2017), volume 10575 of Lecture Notes in Computer Science, pages 171–192. Springer, 2017.*
>
>     *Sara Marie Mc Carthy, Arunesh Sinha, Milind Tambe, and Pratyusa Manadhata. Data exfiltration detection and prevention: Virtually distributed POMDPs for practically safer networks. In Decision and Game Theory for Security (GameSec 2016), volume 9996 of Lecture Notes in Computer Science, pages 39–61. Springer, 2016.*
>
>     *Karel Horak and Branislav Bosansky. A point-based approximate algorithm for one-sided partially observable pursuit-evasion games. In Decision and Game Theory for Security (GameSec 2016), volume 9996 of Lecture Notes in Computer Science, pages 435–454. Springer, 2016.*
>
>     *Marten van Dijk, Ari Juels, Alina Oprea, and Ronald L. Rivest. Flipit: The game of “stealthy takeover”. Journal of Cryptology, 26(4):655–713, 2013.*
>
>     *Albert S. Kyle. Continuous auctions and insider trading. Econometrica, 53(6):1315–1335, 1985.*
>
>     *Kerry Back. Insider trading in continuous time. The Review of Financial Studies, 5(3):387– 409, 1992.*
>
>     *Xintong Wang and Michael P. Wellman. Market manipulation: An adversarial learning framework for detection and evasion. In Proceedings of the Twenty-Ninth International Joint Conference on Artificial Intelligence (IJCAI-20), pages 4626–4632, 2020. Special Track on AI in FinTech.*
>
> ### Response to Questions
> ***
> 1. **How does the atomic structure and the primal–dual DP extend when dynamics or observations are stochastic so P1 cannot precisely steer beliefs?**
>
>     We are actively working on extending the theory to include stochastic dynamics.  Our ongoing study shows the current value, which would be a lower bound, can actually be exact because, from information onesidedness, P1 remains *the only player with control over the belief* (and thus also controls the strategy evolution of both players throughout the game). More formally, the RHS of the Bellman operator can still be written as *a linear functional of the posterior belief distribution*: e.g., for P1, $V(t,x,p) = \int_{\Delta(I)} \tilde{V}(t,x,p’) \nu(p’)$. The difference is that now $\tilde{V}(t,x,p’)$ has to take into account the randomness in future dynamics. That being said, a value gap statement similar to Thm. 4.2 is yet to be developed for games with stochastic and **continuous-time** dynamics.

---

> > ### Author Response · Authors · 2025-11-22
> > **Response to Reviewer 4YnV (2/2)**
> >
> > 2. **Can the authors elaborate on how feedback Nash equilibria are computed in practice for the 2P0S differential game setting? Is the computation exact, or does it involve discretization, function approximation, or numerical solvers? If approximations are used, how is the error controlled or bounded?**
> >
> >     We presented three ways to compute feedback Nash in the paper to provide a holistic evaluation of the effectiveness of the discovered atomic NE structure: (1) In value approximation, we solve the DPs (($\text{P}_1$) and ($\text{P}_2$)) at discretized time steps and sampled states and beliefs in their corresponding continuous spaces. The approximation starts at the terminal time and rolls backward in time. This provides value and strategy approximation for all time, states, and beliefs. (2) In RL, we solve the game for a fixed initial state and belief, and sample (discrete time steps, continuous states, continuous belief) using rollouts. We do not assume knowing the system dynamics. (3) In MPC, we solve the game for a fixed initial state and belief as a single minimax problem, by assuming knowledge about the differentiable system dynamics.
> >
> >     To summarize and answer your questions: Time is always discretized. Value and policy (strategy) networks are used to generalize across continuous state and belief spaces. A numerical descent-ascent solver is used to solve either the minimax game at each infostate (for value approximation and RL), or the entire game (for MPC).
> >
> >     Theorem 4.2 provides the error due to time-discretization, and Theorem 5.1 provides the error associated with minimax solver, and neural network approximation of the value. For value approximation, Thm. 5.1 motivated the use of multigrid, which is a matured numerical method for error control in PDE solve (note that value approximation essentially solves an HJI PDE).
> >
> >
> > 3. **Can the framework be extended to team-based settings with multiple attackers and defenders? How would the equilibrium computation scale or need to change?**
> >
> >     Yes, solving team-based games is exactly where our method has a significant advantage against existing IIEFG solvers, because our method has a complexity **independent** of the number of players within a team. In comparison, IIEFGs have complexities scaling exponentially with respect to the number of players. Consider the 2D football example in Fig.1 with 11 players on each team. The dimensionality of the action space scales linearly with the number of players: 22 for each team. With a coarse discretization of 10 values per action dimension, this leads to an action set of size $U = 10^{22}$ and with 10 time steps, the game tree complexity becomes $U^{2K} = 10^{440}$.

---

### Official Review · Reviewer_f6Dd · 2025-10-31

**Soundness:** 3
**Presentation:** 3
**Contribution:** 3
**Rating:** 6
**Confidence:** 2

**Summary:**

This paper studies the specific case of zero-sum differential games where the only form of imperfect information is player 1's type (with a common prior).

The key contribution is the proof that even though the action space (at every state) can be large or continuous, the support in the NE (behavioral strategy) is at most I (or I+1), where I is the number of types. This characterization of the equilibria can be utilized to solve problems in MARL and MPC.

**Strengths:**

- I generally enjoyed reading the paper. Most papers regarding game solving suffer from an overwhelming complexity in notation, and this paper is no exception; however, the authors partially alleviated this by having text that explains most of the detail well.
- Experiments are generally well done, with a reasonably objective explanation of the results involved (e.g., Sec 6.2).
- The literature on solving continuous time, continuous action games with imperfect information is relatively limited; even though this is a very limited case, the empirical ability to scale looks like it will be a good addition to the literature.

Note: This is slightly outside my field of expertise, so I cannot confidently comment on novelty nor quality.

**Weaknesses:**

- There are several terms which I felt were not well defined, and a simple search did not did yield satisfactory definitions, e.g., I-"atomic".
- There are some technical questions I have/comparisons that I felt were unfair (see below)
- I found the introduction on the multigrid speedup difficult to understand and perhaps difficult to follow for the average reader. Given that it is an important component of getting good performance, perhaps a figure or summary would be more effective, at least in the main paper.

**Questions:**

- I am not convinced about the example given in Figure 1 and line 81. I agree that the branching factor is I instead of the (discretized) U. However, I do not think it is fair to compare the size of a IIEFG and the "primal and dual games" introduced by the author; specifically the way the depth of the search tree is K instead of 2K.
- Is "action error" an appropriate measure of performance (Figure 5)? I would have thought the regular notion of exploitability is more appropriate in a 2p0s setting.
- How does the policy network of P2 look like (e.g., section 5.2). I think it is nice (and consistent with the argument that CAMS-DRL works better than alternatives when NE do not lie in the strict interior) that the architecture allows for this by having action prototypes. Could the authors comment on how the I+1 action prototypes were learnt? I would think there may be a brittleness problem here. For sxample, if due to say, local minima, not all action protypes were learnt (e.g., if two action prototypes turned out to be near identical), then the restriction of outputting only a logit of size I+1 can lead to very a distribution that is very suboptimal since one prototype was "wasted".
-Typos: Line 45 (uniformed vs uninformed)

---

> ### Author Response · Authors · 2025-11-22
> **Response to Reviewer f6Dd (1/2)**
>
> We thank the reviewer for their thorough review and are grateful that they enjoyed reading our paper. Also, thank you for spotting the typo.
>
> ### Response to Weaknesses
> ***
> 1. **There are several terms which I felt were not well defined, and a simple search did not did yield satisfactory definitions, e.g., I-"atomic".**
>
>     Thank you for pointing this out. We provide the definition of I-atomic on paper (line 166): $G_\tau$ is I-atomic, i.e., $\eta_{i, \tau}^\dagger$ concentrates on at most $I$ actions in $\mathcal{U}$. We will push this definition earlier in the paper so that there is no confusion.
>
> 2. **I found the introduction on the multigrid speedup difficult to understand and perhaps difficult to follow for the average reader. Given that it is an important component of getting good performance, perhaps a figure or summary would be more effective, at least in the main paper.**
>
>     We will add [this visual description](https://postimg.cc/ThBqd9mJ) (an anonymous link to the image) of multigrid in the appendix to explain the concepts visually: A standard multigrid scheme consists of four steps (1) Restricting the fine-grid approximation and its residual; (2) solving the coarse-grid problem using the fine-grid residual; (3) computing the coarse-grid correction; and (4) prolonging the coarse-grid correction to fine-grid and add the correction to the fine-grid approximation. The red arrows in the figure represent the restriction process, in which the high-frequency errors in the existing fine-grid approximation is restricted to the coarse grid; and the blue arrows represent the prolongation process, in which the correction obtained by solving the problem in coarse grid is “passed onto” the fine grids.
>
>     Multigrid is especially applicable in our case due to the fact that we are solving continuous-time (differential) games where time discretization introduces approximation error (Thm. 4.2). IIEFGs, in contrast, are discrete-time (dynamic) games, with a fixed time discretization, and hence multigrid is not applicable. Instead, the studies on IIEFGs focus on action-discretization/abstraction if some subset of the nodes in the game-tree contain continuous action (such as bet amount in poker).
>
>
> ### Response to Questions
> ***
> 1. **I am not convinced about the example given in Figure 1 and line 81...**
>
>     For standard IIEFG, each public tree node (infostate) has a branching factor of $U^2$ because we consider both players playing mixed behavioral strategies. With a game tree depth of $K$, this leads to the game tree complexity of $U^{2K}$ and the requirement of converging paired mixed strategies at all reachable tree nodes. For the focused games in this paper, however, we showed through Thm. 4.1 that we can solve P1's and P2's NE **sequentially** through the primal and dual games. For each game, the opponent only needs to play a deterministic best response. Together with the atomic structure of the NE, this leads to a branching factor of $I$ for P1 ($I+1$ for P2) and tree complexity of $I^K$ ($(I+1)^K$).
>
> 2. **Is "action error" an appropriate measure of performance (Figure 5)? I would have thought the regular notion of exploitability is more appropriate in a 2p0s setting.**
>
>     We tracked exploitability during our experiment as a stopping criterion. It should be noted that this exploitability metric, when applied to games with discretized action sets, is not a truthful goodness measure of the learned strategies because the associated best responses are only searched from the corresponding discrete action sets rather than the original continuous action spaces.
>
>     For this reason and for the fact that Hexner’s game has known NE strategies, we decided to report the action error instead because it directly measures how close the learned strategies are to the true NEs across methods.
>
>     That being said, we provide the exploitability vs iterations plots [here](https://postimg.cc/67KS4NSV) (an anonymous link to the image) for the normal-form Hexner’s game below for comparisons between CFR+, MMD, and the proposed CAMS.

---

> > ### Author Response · Authors · 2025-11-22
> > **Response to Reviewer f6Dd (2/2)**
> >
> > 3. **Could the authors comment on how the I+1 action prototypes were learnt?**
> >
> >     The (I+1)-action prototypes for P2 were learned using the dual game.
> >
> >     - For the value approximation mode: Each minimax problem ($\text{P}_2$) is solved using a convergent gradient descent-ascent algorithm DS-GDA (Zheng et al., 2023), similar to ($\text{P}_1$).
> >
> >     - For MARL: The policy network consists of two output heads, one providing the logits (a vector with I-by-I entries for P1 or I+1 entries for P2) and another the action prototypes with dimension I-by-action dimension for P1 or (I+1)-by-action dimension for P2. We then use standard MARL algorithms (MMD, PPO) to solve for the pair of policy networks.
> >     - For MPC: We solve the primal and dual game sequentially. For the primal game, the policy network of P1 is the same as for MARL, and that of P2 outputs a deterministic best-response action. For the dual game, P2 uses a network as for MARL and P1 outputs deterministic best responses.
> >
> >     **Reference**
> >     *Taoli Zheng, Linglingzhi Zhu, Anthony Man-Cho So, José Blanchet, and Jiajin Li. Universal gradient descent ascent method for nonconvex-nonconcave minimax optimization. Advances in Neural Information Processing Systems, 36:54075–54110, 2023.*
> >
> > 4. **I would think there may be a brittleness problem here. For example, if due to say, local minima, not all action prototypes were learnt (e.g., if two action prototypes turned out to be near identical)**
> >
> >     This may well be the case at the true NE. Taking Hexner's game as an example, the ground truth NE is for P1 to take a non-revealing strategy up to a critical time. The non-revealing strategy is deterministic (e.g., P1 moves towards the center of the two potential targets, thus concealing their true target), and thus P1's policy network should indeed converge to outputting two identical action prototypes (with arbitrary conditional probabilities). In our experiments, we do observe this result using CAMS.
> >
> >     The reviewer's concern might be: "would it be possible for the learning dynamics to **falsely** collapse to a smaller number of action prototypes than the ground truth and cannot recover?" This is a great question because the minimax problems with respect to behavioral strategies are in general nonconvex-nonconcave, and our algorithms only guarantee convergence to local saddles. In fact, to our best knowledge, last-iterate IIEFG solvers such as RNaD, MMD, and PPO do not have a convergence proof for behavioral-form strategies either (Sokota et al., 2022; Perolat et al., 2021). (Note: Convergence proofs exist in sequence form, which is not scalable to large games, and also for average-iterate solvers such as CFR variants)
> >
> >     **References**
> >     *Samuel Sokota, Ryan D’Orazio, J Zico Kolter, Nicolas Loizou, Marc Lanctot, Ioannis Mitliagkas, Noam Brown, and Christian Kroer. A unified approach to reinforcement learning, quantal response equilibria, and two-player zero-sum games. arXiv preprint arXiv:2206.05825, 2022.*
> >
> >
> >     *Julien Perolat, Remi Munos, Jean-Baptiste Lespiau, Shayegan Omidshafiei, Mark Rowland, Pedro Ortega, Neil Burch, Thomas Anthony, David Balduzzi, Bart De Vylder, et al. From poincaré recurrence to convergence in imperfect information games: Finding equilibrium via regularization. In International Conference on Machine Learning, pp. 8525–8535. PMLR, 2021.*

---

> > > ### Comment · Reviewer_f6Dd · 2025-11-26
> > > **Acknowledgement**
> > >
> > > I thank the authors for addressing my concerns, as well as for the diagrams. My score remains unchanged.
> > >
> > > I would suggest the authors specify somewhere, even if its the appendix, that they are looking for Nash in behavioral strategies; this wasn't clear to me immediately, and to the best of my knowledge the bulk of the work in the ML community focuses on approximating equilibria in strategic form (either implicitly or via the sequence form).

---

### Official Review · Reviewer_9LSw · 2025-11-01

**Soundness:** 3
**Presentation:** 3
**Contribution:** 4
**Rating:** 6
**Confidence:** 3

**Summary:**

This paper studies two-player zero-sum differential games with one-sided incomplete information. They identify an atomic equilibrium structure, which dramastically reduces the game tree complexity. Base on this, they propose a primal-dual DP reformulation and establishes convergence guarantees. They also leverage this atomic property to develop several scalable solvers. Experiments  shows large reductions in computational cost and improved solution accuracy compared to SOTA solvers.

**Strengths:**

- The paper is generally well organized.
- The paper identifies an atomic Nash structure that theoretically reduces the exponential complexity of imperfect-information differential games. The atomic-NE theorem is clean, leverages convexification geometry, and directly leads to algorithmic simplifications exploited throughout the paper.
- It provides a unifying primal–dual reformulation, which bridges classical differential-game theory and RL/Control.
- The paper validates using realistic case study. The football experiment is an impressive  demo that this theoretical structure can yield large practical gains when assumptions hold.

**Weaknesses:**

- Assumption A5 is kind of strong. It requires full knowledge of dynamics and perfect recall, but many interesting real-world problems might violate these. The atomic results assumes the ability of P1 to precisely control the public belief, and the paper notes the stochastic case only yields lower bounds. These limitations should be emphasized when claiming broad applicability.
- The CAMS value-approximation procedure requires solving many small minimax problems and training value networks.
- Thm 5.1 gives a high-level complexity bound but constants and practical solver behaviour  are sensitive and not fully characterized.

**Questions:**

- Provide a short, explicit toy example, e.g. one or two time steps in the main text to illustrate how the convexification produces the I atoms.
- The paper mentions that P1’s inability to precisely control belief in stochastic settings and that the convexified Bellman operator becomes a lower bound. Is it possible to extend CAMS to stochastic settings?

---

> ### Author Response · Authors · 2025-11-22
> **Response to Reviewer 9LSw (1/2)**
>
> We are thankful to the reviewer for the positive feedback and for recognizing our contribution.
>
> ### Response to Weaknesses
> ***
> 1. **Assumption A5 is kind of strong....**
>
>     Thank you for your suggestion. We will emphasize these limitations in the revision, in addition to highlighting them in the problem statement as we currently do. We have two remarks regarding the assumptions.
>
>     - “Full knowledge of dynamics” is not required from the perspective of someone who solves the game using policy gradients (e.g., PPO variants).
>     - It is likely that stochastic dynamics can be incorporated but we will have to discuss this in a follow-up paper. Very briefly, due to information one-sidedness, P1 remains *the only player with control over the belief* (and thus also controls the strategy evolution of both players throughout the game). In other words, the RHS of the Bellman operators in P_1 and P_2 can still be written as *a linear functional of the primal/dual variable distribution*: e.g., for P1, $V(t,x,p) = \int_{\Delta(I)} \tilde{V}(t,x,p’) \nu(p’)$ where the RHS is linear to the posterior belief distribution. The difference is that now the continuation value $\tilde{V}(t,x,p’)$ has to take into account the randomness in future dynamics. That being said, a value gap statement similar to Thm. 4.2 is yet to be developed for games with stochastic and **continuous-time** dynamics.
>
> 2. **The CAMS value-approximation procedure requires solving many small minimax problems and training value networks.**
>
>     Thank you and this is true. We should note that subgame solves are parallelized following the practice of existing IIEFG solvers such as ReBeL (Brown et al., 2020) and SoG (Schmid et al., 2023).
>
>     **References**
>
>     *Brown, N., Bakhtin, A., Lerer, A., & Gong, Q. (2020). Combining deep reinforcement learning and search for imperfect-information games. Advances in neural information processing systems, 33, 17057-17069.*
>
>     *Schmid, M., Moravčík, M., Burch, N., Kadlec, R., Davidson, J., Waugh, K., ... & Bowling, M. (2023). Student of games: A unified learning algorithm for both perfect and imperfect information games. Science Advances, 9(46), eadg3256.*
>
> 3. **Thm 5.1 gives a high-level complexity bound but constants and practical solver behaviour are sensitive and not fully characterized.**
>
>     This is also true. We used Thm 5.1 to explain that even with the discovery of the atomic NE structure, games with large time horizons are still **intrinsically** hard to solve. This hardness comes from the fact that value approximation is essentially about solving a Hamilton-Jacobi (nonlinear) PDE with a terminal boundary, where the value landscape has to be approximated backward in time from the terminal. Thm 5.1 thus motivated the use of multigrid, which has matured and successful applications to this type of problems.
>
> ### Response to Questions
> ***
> 1. **Explicit toy example showing how convexification produces I atoms.**
>
>     A simple example is explained in Fig. 2 of the paper. An improved illustration can be found [here](https://postimg.cc/Y4VWypH0).
>
>     To reiterate, consider a game with $I=2$ payoff types and we arrive at a particular infostate at time t, state x, and belief p. We call  $\tilde{V}(t,x,p)$ the nonrevealing value at (t,x,p), which is resulted from if players play a deterministic NE (the existence and uniqueness is ensured by Isaacs’ condition). We can see from the figure that $\tilde{V}$ is not necessarily convex in p, and thereby it is possible for P1 (the informed player) to achieve a lower value by convexifying $\tilde{V}$ through a mixed strategy. Specifically, this can be achieved by first identifying two “splitting” points in the belief simplex, $p^a$ and $p^b$ , and the corresponding weights $\lambda^a$ and $\lambda^b$, respectively, such that $\lambda^a p^a + \lambda^b p^b = p$. The nonrevealing NE strategy profiles $(u^a, v^a)$ and $(u^b, v^b)$ are then computed at infostates $(t,x,p^a)$ and $(t,x,p^b)$, respectively. Lastly, we can construct P1’s mixed strategy as to play action $u^k$ (for $k \in$ {$a$,$b$}) with probability $\alpha^k = p^k[i]\lambda^k/p[i]$ if P1 is of type $i$. By playing this strategy, P1 controls the belief to shift to $p^k$ if action $u^k$ is taken, and thereby obtains a convexified value $V(p) = \lambda^a \cdot  \tilde{V}(p^a) + \lambda^b \cdot \tilde{V}(p^b)$.
>
>     For a numerical example, we walked through a two-player zero-sum beer-quiche game in Appendix F.2 with step-by-step derivation of the NE strategy for P1 and P2 using the primal $(\text{P}_1)$ and dual games $(\text{P}_2)$.
>
>     For example, for P1 who maximizes in this case, their value function at the initial time of the game $V(0,p)$ is defined as the concavation in Eq. (43), leading to its final form in Eq. (44). One can use Eq. (42) to plot out $max_{u\in\{B,Q\}} V(1,u,p)$ and see that at any $p$, the concavation of $V(0,p)$ involves $I=2$ actions.

---

> > ### Author Response · Authors · 2025-11-22
> > **Response to Reviewer 9LSw (2/2)**
> >
> > 2. **Is it possible to extend CAMS to stochastic settings?**
> >
> >     Thank you for bringing this up. This is an important limitation we are working on. Our ongoing study shows that the lower bound can actually be exact because due to information one-sidedness, P1 remains *the only player with control over the belief* (and thus also controls the strategy evolution of both players throughout the game). In other words, the RHS of the Bellman operators in P_1 and P_2 can still be written as *a linear functional of the primal/dual variable distribution*: e.g., for P1, $V(t,x,p) = \int_{\Delta(I)} \tilde{V}(t,x,p’) \nu(p’)$ where the RHS is linear to the posterior belief distribution. The difference is that now the continuation value $\tilde{V}(t,x,p’)$ has to take into account the randomness in future dynamics. That being said, a value gap statement similar to Thm. 4.2 is yet to be developed for games with stochastic and **continuous-time** dynamics.

---

> > > ### Comment · Reviewer_9LSw · 2025-11-24
> > >
> > > Thank you for your detailed response and clarifications, which address my concerns regarding the CAMS value-approximation procedure. It would be great if these revisions could be incorporated during the discussion. I'm also reading the other reviewers' comments. To me, I still have a few further questions.
> > >
> > > When you say that “full knowledge of the dynamics is not required from the perspective of someone who solves the game using policy gradients (e.g., PPO variants),” do you mean solving the game using stochastic gradient methods rather than exact gradient-based methods? If so, is the theoretical analysis based on  the exact gradient, or stochastic one?

---

> > > > ### Author Response · Authors · 2025-11-24
> > > >
> > > > Thank you for the follow-up. We updated the paper with the new visualization for Fig. 2. We will highlight all the changes made so far.
> > > >
> > > > To clarify, we do require that **the players themselves** know the dynamics of the game, because this assumption is necessary for the game to have a value, which is a starting point for analyzing the structure of the underlying Nash Equilibrium. Regarding how we solve the game as solvers, the algorithmic choice, e.g., stochastic vs exact gradient-based methods, doesn't affect the conclusions about the Nash Equilibrium structure.

---

### Official Review · Reviewer_52aL · 2025-11-01

**Soundness:** 3
**Presentation:** 3
**Contribution:** 3
**Rating:** 6
**Confidence:** 3

**Summary:**

This paper studies a class of two-player zero-sum differential games with one-sided information (2p0s1) and proves that equilibrium strategies are atomic in the number of payoff types $I$. P1 needs at most $I$ action prototypes and P2 at most $I+1$. This collapses game-tree complexity by reducing the effective branching factor from $U$ (all actions) to $I$ or $I+1$. Building on this, the authors implement three solvers to exploit this problem structure: a model-based pipeline (CAMS), a model-free RL version (CAMS-DRL using PPO/MMD), and a control variant (CAMS-MPC). The authors show learning speed and performance improvements vs methods that use large discrete action spaces. They also demonstrate behavior on an 11v11 American football domain.

**Strengths:**

- The paper is well written with concepts presented clearly.
- The proofs of $I$ and $(I+1)$-atomicity are useful and relevant to solving large 2p0s games with asymmetric information.
- The authors show empirical gains by leveraging this problem structure across a range of algorithmic approaches.
- OpenSpiel-compatible implementations are provided.

**Weaknesses:**

1) PPO/MMD hyperparameters and any relevant sweeps are not listed in the paper. It's possible that not enough hyperparameter tuning was done (for both the proposed method and baselines).


Rudolph et al. (2025) describe that (well-regularized) PPO and MMD are quite sensitive to the entropy coefficient. While they generally recommend quite high coefficients like 0.1 or 0.05, if PPO and MMD have issues converging to non-interior solutions, perhaps there are lower entropy coefficients (or different learning rates) that are sufficient for this domain, which is quite different from dark hex/PTTT?

It would be great to see a minor sweep on these baselines to make sure that PPO/MMD isn't flatlining just because, e.g. the entropy regularization is too high.

Rudolph et al. Reevaluating policy gradient methods for imperfect-information games. 2025.

--

2) Adding exploitability or approximate exploitability comparisons could improve this paper to help measure convergence to NE. This is less important as the authors provide other indicators of convergence.

**Questions:**

Your algorithms parameterize each player’s policy as a mixture over $I$ or $I+1$ action prototypes, relying on the paper’s existence result for atomic equilibria. In practice, how robust are the learned policies to an unconstrained best response that is not parameterized atomically (and instead operates over the raw action space)? How large is the risk that restricting your learning process to atomic mixtures can allow non-atomic opponents to induce out-of-distribution sequences and inputs in agents trained with the proposed method?

---

> ### Author Response · Authors · 2025-11-22
> **Response to Reviewer 52aL (1/2)**
>
> Thank you for taking the time to review our work. Here we address the concerns and questions raised by the reviewer.
>
> ### Response to Weaknesses
> ***
> 1. **PPO/MMD hyperparameters and any relevant sweeps are not listed in the paper. It's possible that not enough hyperparameter tuning was done (for both the proposed method and baselines)....It would be great to see a minor sweep on these baselines to make sure that PPO/MMD isn't flatlining just because, e.g. the entropy regularization is too high.**
>
>     Following the reviewer’s suggestion, we conducted a hyperparameter sweep for the normal-form Hexner’s game with action size of $U=100$, using learning rates (lr) in $[2.5e-5, 2.5e-4, 2.5e-2, 2.5e-1]$ and entropy coefficients in $[0.01, 0.05, 0.1, 0.2]$ for PPO, MMD, and RNaD. All three are standard OpenSpiel-compatible algorithms that approximate gradients using randomly sampled rollouts (256 samples per gradient). Exploitability and action convergence results are reported [here](https://postimg.cc/5Xqc3mZq) (an anonymous link to the image) and will be added to the revision.
>
>     *Remarks on the results:*
>     - PPO and MMD fail to converge in either exploitability or action error across the hyperparameter sweep. This suggests that these algorithms are sensitive to hyperparameter settings, even though there may exist certain convergent settings outside of the tested range.
>     - We report RNaD, which uses the same game and learning setup as PPO and MMD, and achieves better convergence under some learning rates (note that RNaD does not use an entropy coefficient). However, compared to the CAMS version of MMD and PPO, RNaD performance is still limited by its action-space discretization, see Fig. 5 in the paper.
>
>     To rule out the possibility of implementation issues, we note that the implementation of PPO, MMD, and RNaD directly comes from the open source code of Rudolph et al.(2025).
>
>     **Reference**
>
>     *Rudolph et al. Reevaluating policy gradient methods for imperfect-information games. 2025.*
>
>
> 2. **Adding exploitability or approximate exploitability comparisons could improve this paper to help measure convergence to NE. This is less important as the authors provide other indicators of convergence.**
>
>     Thank you for your suggestion. We will add exploitability convergence results to the revision.
>     We tracked exploitability during experiments as a stopping criterion as is standard in OpenSpiel. It should be noted, however, that this exploitability metric is defined on the discrete action sets, and thus is not a direct goodness measure of the learned strategies when applied to games that originally have continuous action spaces, because the exploiting best responses are only searched from the discrete action sets rather than the continuous ones.
>
>     For this reason and for the fact that Hexner’s game has known NE strategies, we decided to report the action error instead since the error directly measures how close the learned strategies are to the true NEs across methods.
>
>
>     That being said, we provide [here](https://postimg.cc/67KS4NSV) (anonymous link) and in the revision the exploitability convergence comparisons for the normal-form Hexner’s game between CFR+, tabular MMD, and the proposed CAMS. Note that tabular MMD uses exact gradients but is only applicable to small game trees.
>
> ## Response to Questions
> ***
>
> 1. **In practice, how robust are the learned policies to an unconstrained best response that is not parameterized atomically (and instead operates over the raw action space)?**
>
>     We first note that there always exist best responses in the deterministic strategy space. This is because if a mixed strategy (over the entire action space) is exploitive, then there exists a deterministic strategy that exploits more. Therefore we used deterministic best responses in all exploitability computation (as is standard in OpenSpiel).
>
>
>     We would also like to highlight that the proposed method (CAMS) achieves better robustness than baselines (DeepCFR, JPSPG, MMD, PPO, and R-NaD). Specifically, for 4-stage Hexner’s game (Fig. 3(d) for the value approximation mode and Fig. 5(b) for the MARL mode), we see that while CAMS has non-zero errors, indicating non-zero exploitability, its strategies are much closer to the ground truth than those from the baselines under the same or less compute (see wall time comparison in Fig. 3(a)). The advantage of CAMS is also visualized in Fig. 9 for the same game and learning setup as for Fig. 3(d): Due to action discretization, DeepCFR cannot successfully converge to anything even remotely close to the true NE strategies. JPSPG, while designed for continuous action spaces, also fails potentially due to its reliance on pseudo-gradients.

---

> ### Author Response · Authors · 2025-11-22
> **Response to Reviewer 52aL (2/2)**
>
> 2.  **How large is the risk that restricting your learning process to atomic mixtures can allow non-atomic opponents to induce out-of-distribution sequences and inputs in agents trained with the proposed method?**
>
>
>     Risks (non-zero exploitability) against out-of-distribution action sequences exist because of intrinsic limitations of learning methods rather than the introduction of the atomic structure: In theory, there is zero exploitability for a player playing their Nash equilibrium strategy against any opponent’s mixed strategies. And the key insight from this paper is that for 2p0s1 games, equilibrium strategies are always atomic (Thm. 4.1 and 4.2). Therefore, exploitability does not come from modeling strategies as atomic.
>
>     Let us now explain potential risks for different learning modes, and emphasize that existing mitigation methods can be implemented without interfering with the atomic NE structure.
>     - **Value approximation mode**: We approximate the value function over the entire (time, state, belief) space, and as a result, players can always solve a minimax problem (as in $(\text{P}_1)$ and $(\text{P}_2)$) to approximate their NE behavior strategies at any infostate, i.e., their strategies are robust to opponents playing non-optimally (“out of distribution”). That being said, there are two sources of risks: (1) The NE strategy derived from **perfectly** solving ($\text{P}_1$) or ($\text{P}_2$) at all game tree nodes has a bounded value gap as stated in Thm. 4.2. Reducing this gap requires a long time horizon. (2) Since ($\text{P}_1$) and ($\text{P}_2$) are only approximately solved through gradient descent-ascent, and value/strategy approximation has to be implemented to accommodate for the continuous state space, there are additional value approximation errors. Thm. 5.1 provides the computational complexity for mitigating these risks. The fact that this complexity is exponential to the time horizon motivated the use of multigrid.
>
>     - **MARL**: Due to the nature of random sampling for proximal gradients, there are usually only a subset of (time, state, belief) nodes being visited before the learning converges/terminates. Therefore, it is possible that the value and strategy approximations are incorrect for (time, state, belief) nodes that are not visited during the learning. This intrinsic issue of MARL can be empirically addressed by the entropy bonus that promotes game tree exploration during the learning.
>
>     - **Model predictive control (MPC)**: Our MPC mode is different from MARL in that it uses known differentiable dynamics to compute the exact gradients during descent-ascent (which is enabled by the reduced tree complexity due to its atomic NE structure). In addition, for the case studies we discussed, the subgames after complete information revelation (belief becomes one-hot) are strongly convex-concave minimax problems due to control-affine dynamics and convex running losses, and therefore have unique NEs. That being said, the use of neural networks for modeling behavioral strategies may cause convergence to **local** saddles and thus non-zero exploitability against “out-of-distribution” strategies. This issue can be alleviated by (1) introducing parallel policy network initializations; and (2) online warm-start of MPCs to update strategies once out of distribution (which, again, becomes possible due to the tree complexity reduction).

---

### Author Response · Authors · 2025-11-24
**Revision**

Based on the reviewers' suggestions, we've made the following revisions:

1. We revised the sentence in line 41 to: "Applying existing solvers to differential games would require game-specific insight or extra computational overhead for automated abstraction (Kroer & Sandholm, 2015; Hawkin et al., 2011; Brown & Sandholm, 2015)."
2. Updated Fig. 2 with better visualization.
3. Fixed typo in line 45.
4. Added a visual description of multigrid method in Appendix E.
5. Added exploitability comparison plots in Appendix G.5.
6. Added hyperparameter sweep results for PG baselines in Appendix H.
7. Added citation to line 114.
8. Replaced all gendered pronouns with gender-neutral pronoun "it".

---

### Meta-Review · Area_Chair_geL1 · 2026-01-16

**Summary:**

The paper tackles the complexity of continuous games by proving that in one-sided information scenarios, optimal strategies actually boil down to just a few specific actions. By exploiting this structure to drastically simplify the game tree, the authors managed to solve a realistic 22-player football game (a task that effectively breaks standard solvers).

**Reviewer Concerns:**

There were few concerns of the reviewers claiming that the authors fail to discuss existing action abstraction methods. Additionally, there were concerns about the assumptions and also benchmarks were viewed as insufficient. Nevertheless, most of the reviewers were positive (one is negative) about this work and the authors put a lot of effort in addressing the concerns of the negative reviewer.

**Reviewer Scores:**

It is not clear to the AC whether the negative reviewer would be in favor of accepting the paper after the rebuttal. In the AC's opinion, the paper has some interesting results and recommends acceptance with reservations.

---

### Decision · Program_Chairs · 2026-01-26

Accept (Poster)